# Autophagy-regulated mitochondrial inheritance controls early CD8+ T cell fate commitment

Mariana Borsa [1,2] ✉, Ana Victoria Lechuga-Vieco [1,3], Amir H. Kayvanjoo [2], Edward Jenkins[1], Yavuz Yazicioglu[1], Ewoud B. Compeer [1], Felix C. Richter[1], Simon Rapp [1,2], Robert Mitchell[1], Tom Youdale[4], Hien Bui[5,6], Emilia Kuuluvainen[5,6], Michael L. Dustin [1], Linda V. Sinclair [4], Pekka Katajisto [5,6,7] & Anna Katharina Simon [1,2] ✉

T cell immunity deteriorates with age, accompanied by a decline in autophagy and asymmetric cell division. Here we show that autophagy regulates mitochondrial inheritance in CD8+ T cells. Using a mouse model that enables sequential tagging of mitochondria in mother and daughter cells, we demonstrate that autophagy-deficient T cells fail to clear premitotic old mitochondria and inherit them symmetrically. By contrast, autophagy-competent cells that partition mitochondria asymmetrically produce daughter cells with distinct fates: those retaining old mitochondria exhibit reduced memory potential, whereas those that have not inherited old mitochondria and exhibit higher mitochondrial turnover are long-lived and expand upon cognate-antigen challenge. Multiomics analyses suggest that early fate divergence is driven by distinct metabolic programmes, with one-carbon metabolism activated in cells retaining premitotic mitochondria. These findings advance our understanding of how T cell diversity is imprinted early during division and support the development of strategies to modulate T cell function.

Efficient immune responses depend on coordination amongst different immune cells and on generating diversity within the same cell type. In CD8+ T cells, one single cell can differentiate into progeny with heterogeneous fates upon activation[1,2]. Activation of a naive T cell produces both short-lived effector cells that exert cytotoxic effector functions and long-lived memory cells that self-renew and differentiate upon antigenic rechallenge and are central to vaccination efficacy. Despite increased understanding of mechanisms that contribute to fate decision, there is still no consensus on when these decisions occur and how long-lived memory T cells form[3–5]. Moreover, T cell memory is severely impaired during ageing[6,7], and senescent CD8+ T cell subsets that exhibit DNA damage, cell cycle arrest, mitochondrial dysfunction[8] and global poor effector function accumulate[9–11]. Amongst the processes supporting CD8+ T cell memory formation and maintenance, two highly conserved mechanisms stand out as being negatively impacted by ageing: macroautophagy (hereafter autophagy) and asymmetric cell division (ACD)[6,12,13].

Autophagy involves the recycling and degradation of cellular cargoes through the engulfment of cellular components and organelles by autophagosomes, followed by their fusion with lysosomes.

[1]Kennedy Institute of Rheumatology, University of Oxford, Oxford, UK. [2]Max-Delbrück-Center for Molecular Medicine, Berlin, Germany. [3]Institute for Research in Biomedicine, Barcelona, Spain. [4]Division of Cell Signalling and Immunology, School of Life Sciences, University of Dundee, Dundee, UK. [5]Faculty of Biological and Environmental Sciences, University of Helsinki, Helsinki, Finland. [6]Institute of Biotechnology, HiLIFE, University of Helsinki, Helsinki, Finland. [7]Department of Cell and Molecular Biology, Karolinska Institutet, Stockholm, Sweden. ✉e-mail: mariana.borsa@kennedy.ox.ac.uk; katja.simon@mdc-berlin.de

The role of autophagy in immune cell fate decision is cell- and context-dependent[14]. In CD8[+] T cells, loss of autophagy results in impaired memory responses[6,15,16], which is at least partly caused by accumulation of damaged organelles[17–19]. ACD has been well characterized in model organisms[20], but in primary mammalian cells, evidence of its impact on fate decisions remains correlative and inconclusive[21]. In cells from the haematopoietic lineage, this arises from technical limitations, as identification of sibling cells has relied on cell cargoes not directly tied to fate decisions or expressed in patterns that prevent conclusive inheritance assignment from a mother cell. Thus, two critical issues remain unresolved: (1) Is there a cargo that can drive fate divergence? (2) Is inherited material synthesized post cell division, or is it inherited asymmetrically? Here, we address these questions in CD8[+] T cells. ACD in CD8[+] T cells generates effector-like and memory-like daughter cells[22] from the very first mitosis following naive T cell activation by high-affinity T cell receptor (TCR) stimulation[23,24]. These cells inherit several layers of asymmetry, including surface markers, transcription factors, centrosomes, divergent metabolic activity and translation profiles[25–29]. Despite this, direct causal evidence linking asymmetric inheritance of premitotic (preM) T cell cargo, and the future fate of emerging daughter cells in vivo is lacking not only in T cells but across primary immune cells in general.

Because it is unclear whether cell cargo degradation can contribute to cell division asymmetries, we performed an integrated functional analysis of the contribution of autophagy and ACD to CD8[+] T cell differentiation. We identified mitochondria producing superoxide as asymmetrically inherited cargo, which was amplified by mitophagy. Using a novel mouse model that allows specific and sequential mitochondria labelling, we insured unequivocal tracking between preM and postmitotic (postM) cargo and analyses of segregation, degradation and biogenesis events. This novel system allowed us to follow the presence of preM mitochondria by imaging and flow cytometry and evaluate the impact of mitochondrial inheritance by single-cell RNA sequencing (scRNA-seq) proteomics, metabolomics and in vivo transfer of daughter cells. Our results suggest that autophagy contributes to the generation of early fate divergency by promoting both clearance and asymmetric partitioning of heterogeneous mitochondria populations. Furthermore, we establish the first causal link between the inheritance of cell cargo and fate commitment in immune cells, as low mitochondrial turnover caused poor memory potential in CD8[+] T cell immune responses. Our findings offer new insight into how T cell diversity is early imprinted and how organelle inheritance regulates metabolism and function, opening paths for refined therapeutic approaches to modulate T cell function.

## Results

### Divergent proteome and mitochondrial inheritance in CD8[+] T cell mitosis relies on autophagy

ACD in CD8[+] T cells results in unequal inheritance of cell cargoes that culminates in divergent transcriptomes between daughter cells[28,30–32]. To broaden our understanding of early events of asymmetric segregation, we examined the global proteome of first-daughter CD8[+] T cells using CD8 expression as surrogate marker of effector-like (CD8[hi]) and memory-like (CD8[lo]) progenies[30] (Fig. 1a). More than 6,000 proteins were identified, and the proteomic ruler method was used to calculate both protein mass and copy numbers of each protein per cell[33]. Although protein mass was comparable between CD8[hi] and CD8[lo] daughter cells (Extended Data Fig. 1a), we identified several proteins that were enriched in one of these two populations, many of them with roles associated with metabolism and mitochondrial function and biogenesis (Fig. 1b). A comprehensive analysis of mitochondria-related proteins further emphasized this divergence (Fig. 1c). Given the known role of mitochondria in T cell fate, we focused on these organelles[34,35]. Mitochondrial inheritance in mitotic T cells has been previously investigated with conflicting results[26,27,36,37]. By electron microscopy, we could neither observe any differences in mitochondrial content (Fig. 1d) nor architecture. However, by confocal fluorescence microscopy, we observed that CD8[hi] (effector-like) daughter cells inherited more mitochondria producing reactive oxygen species (ROS, a readout of mitochondrial superoxide) (Fig. 1e). Because ROS-producing mitochondria are targets of autophagy, a mechanism essential for T cell memory and known to decline with age, we interrogated whether mitophagy contributes to this unequal inheritance. To address whether priming would be impacted by constitutive loss of autophagy, we stimulated wild-type (WT) and Atg7[fl/fl] Cd4[Cre] CD8[+] T cells on supported lipid bilayers (SLB) and observed that the immune synapse area and TCR clustering were distinct between autophagy-sufficient and autophagy-deficient CD8[+] T cells (Extended Data Fig. 1b,c). As immune synapse formation and TCR affinity and signalling strength are crucial for ACD[22–24], we moved to an inducible model of autophagy deletion (Atg16l1[fl/fl] ROSA26(R26)-Ert2[Cre]), allowing T cell activation in presence of autophagy (Extended Data Fig. 1d). Loss of autophagy abolished the asymmetric inheritance of mitochondria producing superoxide (Fig. 1e). We also observed that the autophagic machinery itself is polarized during cell

**Fig. 1 | Autophagy regulates asymmetries in CD8[+] T cell mitosis. a**, The experimental layout: CTV-labelled naive CD8[+] T cells were activated on anti-CD3-, anti-CD28- and Fc-ICAM-1-coated plates for 36–40 h. The cells were collected and stained with anti-CD8 antibodies. First-daughter cells were identified as the first peak of CTV dilution (in reference to undivided cells). CD8[hi] and CD8[lo] cells were sorted as populations expressing 20% highest or lowest CD8, respectively, as previously described[30]. The cell pellets were frozen and stored until being processed for proteomics analysis. **b**, A volcano plot showing differentially inherited proteins by CD8[hi] and CD8[lo] daughter cells. The data are pooled from four samples done in two independent experiments. Each sample had cells originally collected from two to three mice. A statistical analysis was performed using an unpaired two-tailed Student's t-test. Encoding genes for proteins among the top 50 differentially expressed in CD8[lo] and CD8[hi] daughter cells are highlighted. The genes in bold have their function linked to mitochondrial metabolism and function. **c**, A heat map from proteomics data (made on GraphPad) listing genes encoding mitochondrial proteins. The black squares represent proteins that were absent in one of the groups. The scale bar represents the coefficient of variation. **d**, Left: representative transmission electron microscopy images from CD8[hi] and CD8[lo] daughter cells emerging from the first mitosis following naive CD8[+] T cell activation. The number of mitochondria per image (per slice) was calculated (right, n = 35). The experiment was performed twice with pooled cells from ten mice. **e**, Representative images of WT (Atg16l1[fl/+] R26-Ert2[Cre], n = 26) and autophagy KO (Atg16l1[fl/fl] R26-Ert2[Cre],

n = 26) mitotic CD8[+] T cells 36–40 h post activation. Autophagy depletion was achieved by culturing cells in presence of 500 nM 4OHT. Inheritance of MitoSOX was calculated as previously described[30]. Any values above or below the grey area in the graph were considered asymmetric. The data are from two independent experiments (each with cells from two to three mice) are represented as the mean ± s.e.m. A statistical analysis was performed using an unpaired two-tailed Student's t-test. The exact P values are depicted in the figure. **f**, Representative images of mitotic CD8[+] T cells from young (8–16 weeks old, n = 95) and old (>100 weeks old, n = 40) mice 36–40 h post activation. Inheritance of LC3B, a marker of autophagosomes, was calculated in each group. The data from more than three independent experiments (each with pooled cells from two mice) are represented as the mean ± s.e.m. A statistical analysis was performed using an unpaired two-tailed Student's t-test. The exact P values are depicted. **g**, A volcano plot showing differentially inherited proteins by CD8[hi] and CD8[lo] daughter cells from Atg16l1[fl/+] R26-Ert2[Cre] (post-tamoxifen inducible depletion of autophagy) and old (>100 weeks) mice. The data are pooled from two (old) or four (Atg16l1-KO) samples. Each sample had cells originally collected from two to three mice. Statistical analysis was performed using an unpaired two-tailed Student's t-test. Encoding genes for proteins among the top 50 differentially expressed in CD8[lo] and CD8[hi] daughter cells are highlighted in green and magenta, respectively, for each type of sample. The genes are ordered from top to bottom in decreasing fold-change values. The genes in bold have their function linked to mitochondrial metabolism and function. Proteomics volcano plots were done using Tableau.

division, as membrane-bound LC3B-II (microtubule-associated protein 1A/1B light-chain 3), a marker of autophagosomes[14], was preferentially inherited by CD8[hi] effector-like cells, suggesting reduced autophagic flux and accumulation of damaged cargo in this subset. In CD8[+] T cells from aged mice, LC3B-II displayed symmetric distribution, consistent with impaired autophagy and weakened ACD potential[12,15,16] (Fig. 1f). Finally, proteomic comparisons reinforced these observations (Fig. 1g). In CD8[+] T cells from both autophagy-deficient and aged mice the number of differentially inherited proteins was reduced compared with WT cells (Fig. 1b). Moreover, proteins linked to mitochondrial function were less represented amongst differentially inherited cargoes.

Interestingly, the pool of enriched proteins found in CD8[hi] and CD8[lo] progenies from old mice was very small, reflecting the dual decline of ACD and autophagy with age. Together these findings highlight the relevance of autophagy not only to support cellular homeostasis but also to establish asymmetric inheritance patterns.

### Segregation of old mitochondria is autophagy-dependent

To functionally address whether organelle inheritance influences fate determination during ACD, we used a knock-in murine model in which the mitochondrial outer membrane protein 25 (OMP25) is fused to a SnapTag upon Cre recombination (*Omp25*-SnapTag)[38]. We crossed

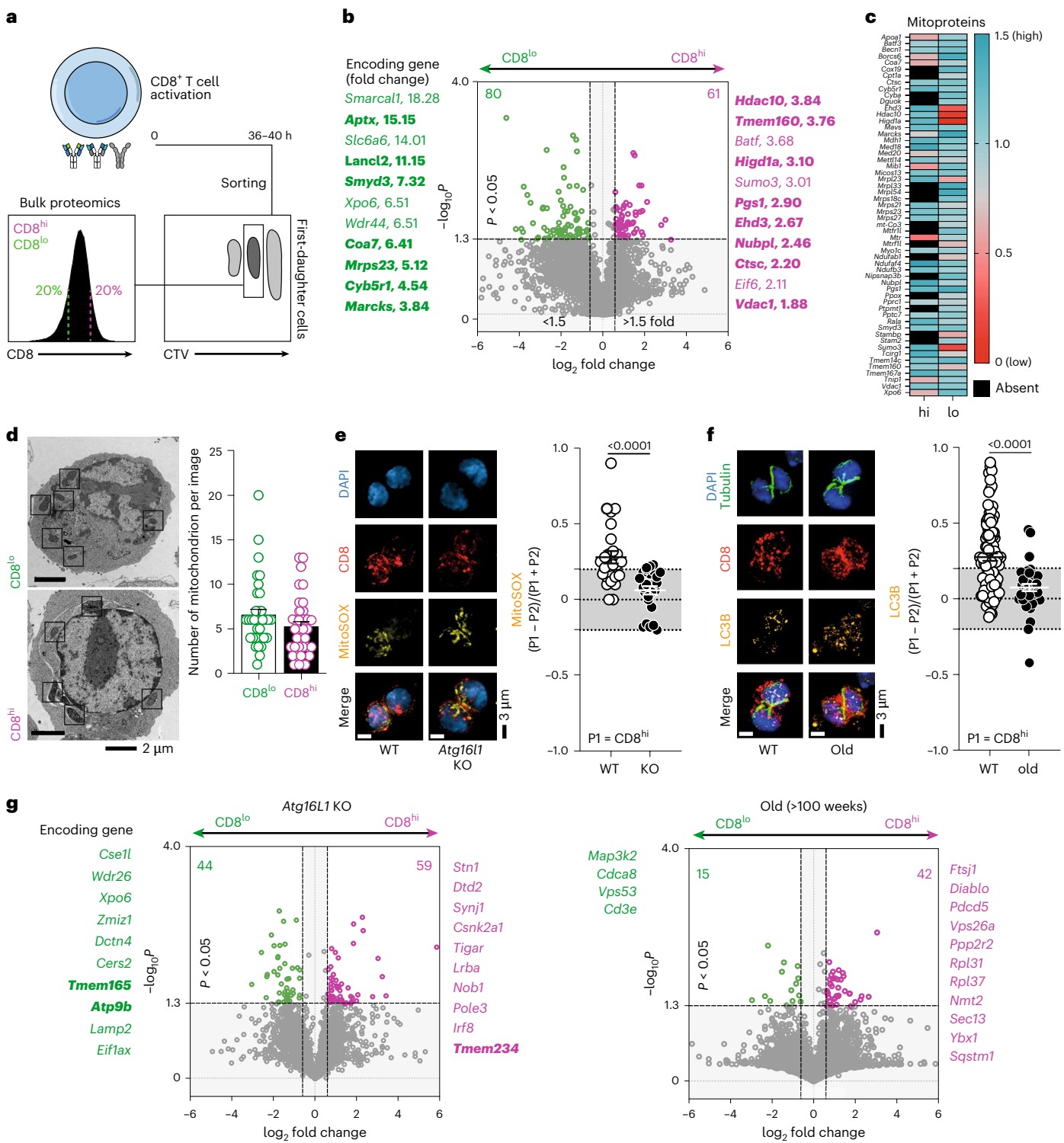

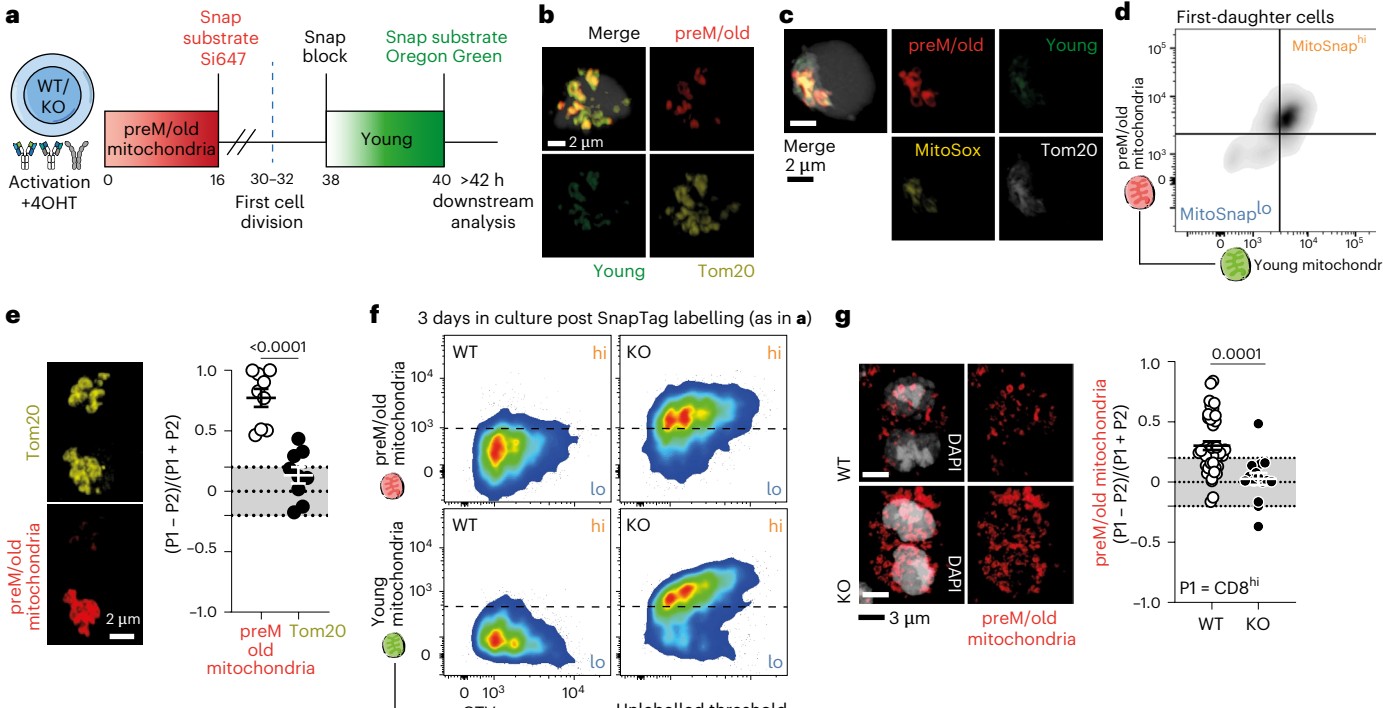

**Fig. 2 | Segregation of old mitochondria is autophagy-dependent. a**, The experimental layout: CTV-labelled naive MitoSnap CD8⁺ T cells (WT-*Atg16l1*[fl/+] *Omp25*[fl/+] *Ert2*[Cre] or KO-*Atg16l1*[fl/fl] *Omp25*[fl/+] *Ert2*[Cre]) were activated on anti-CD3-, anti-CD28- and Fc-ICAM-1-coated plates for 40 h. The cells were cultured in T cell medium containing 500 nM 4OHT. A total of 16 h post activation, the cells were collected and labelled with Snap-Cell 647-SiR to tag preM/old mitochondria and cultured for a further 24 h, when Snap-Cell Block and Snap-Cell Oregon Green incubations allowed young organelle labelling. A downstream analysis was done >2 h after cell resting in complete T cell medium at 37 °C. **b**, Representative confocal microscopy images showing specificity of SnapTag labelling (staining overlaps with anti-Tom20 antibody labelling) in WT MitoSnap CD8⁺ T cells 36 h post activation. The experiment was performed independently twice. **c**, Representative confocal microscopy images showing overlap between MitoSOX staining and preM/old mitochondria labelling in WT MitoSnap CD8⁺ T cells 36 h post activation. The experiment was performed independently twice. **d**, Representative flow cytometry plot of preM/old and postM/young mitochondria inheritance amongst activated MitoSnap CD8⁺ T cells following

first cell division. **e**, Representative confocal microscopy images of mitotic MitoSnap CD8⁺ T cells 36–40 h post activation (*n* = 9). Asymmetric inheritance of Tom20 (total mitochondria) and SnapTag preM/old mitochondria were calculated. The data from one experiment, with three biological replicates, are represented as the mean ± s.e.m. A statistical analysis was performed using an unpaired two-tailed Student's *t*-test. The exact *P* values are depicted in the figure. **f**, Representative flow cytometry plots showing inheritance of young and old mitochondria during several cell division cycles in both autophagy-sufficient (WT) and autophagy-deficient (KO) cells. The experiment was performed independently twice (pooled cells from two biological replicates). **g**, Representative confocal microscopy images of mitotic WT (*n* = 44) and KO (*n* = 12) MitoSnap CD8⁺ T cells 36–40 h post activation. The asymmetric inheritance of preM/old mitochondria was calculated in each group. The data derived from two independent experiments (cells from two or three biological replicates) are represented as the mean ± s.e.m. A statistical analysis was performed using an unpaired two-tailed Student's *t*-test. The exact *P* values are depicted.

these mice to *R26-Ert2*[Cre] (hereafter *Ert2*[Cre]), which allowed inducible SnapTag expression upon tamoxifen treatment. SnapTag can covalently bind to cell-permeable substrates linked to different fluorophores, enabling sequential labelling of mitochondria and accurate discrimination between old (preM) and young (postM) organelles[39,40] (Fig. 2a). With the MitoSnap system we can unequivocally link the inheritance of labelled (preM/old) mitochondria to an event of asymmetric segregation of a cell cargo that was present minimum 14 h before cell division. Furthermore, this approach is unaffected by postmitotic transcriptional, translational or anabolic events of biogenesis. Fluorescence microscopy confirmed SnapTag specificity and showed colocalization with Tom20⁺ structures (Fig. 2b). Importantly, MitoSnap[lo] cells result from segregation of old mitochondria (preM labelling) and/or degradation of old (preM) and young (postM) mitochondria, as labelling efficiency is close to 100% (Extended Data Fig. 2a). We also confirmed that old mitochondria are MitoSOX⁺ (Fig. 2c and Extended Data Fig. 2b).

Flow cytometry of first-daughter CD8⁺ T cells revealed two main populations: one inheriting both preM/old and postM/young mitochondria and another with low SnapTag labelling (Fig. 2d). Fluorescence microscopy of dividing cells demonstrated that while total mitochondrial mass was evenly distributed, old mitochondria were

asymmetrically partitioned, preferentially segregating into CD8[hi] progenies (Fig. 2e). To further verify the contribution of autophagy to the generation of MitoSnap[hi] and MitoSnap[lo] populations, we generated autophagy-deficient MitoSnap mice (*Atg16l1*[fl/fl] *Omp25*-SnapTag[fl/+] *Ert2*[Cre]). Tracking mitochondria over several division cycles showed that loss of autophagy led to retention of both old and young mitochondria, indicating impaired turnover (Fig. 2f). Finally, confocal microscopy of mitotic CD8⁺ T cells confirmed that autophagy loss abolished asymmetric segregation of old mitochondria (Fig. 2g), placing autophagy as a critical mechanism coupling organelle quality control to asymmetric inheritance of cell fate determinants during ACD.

### Mitophagy drives the emergence of cell progenies inheriting heterogeneous mitochondrial populations

To dissect the role of autophagy in generating MitoSnap[lo] progenies, we compared the fate of MitoSnap[hi] and MitoSnap[lo] cells following first CD8⁺ T cell division (Fig. 3a). We observed that WT cells that were originally MitoSnap[hi] became MitoSnap[lo], whereas most MitoSnap[hi] autophagy-deficient CD8⁺ T cells retained their labelling (Fig. 3b). To confirm whether OMP25 staining loss is caused by mitochondrial degradation, we used bafilomycin A (BafA) to inhibit autophagy flux

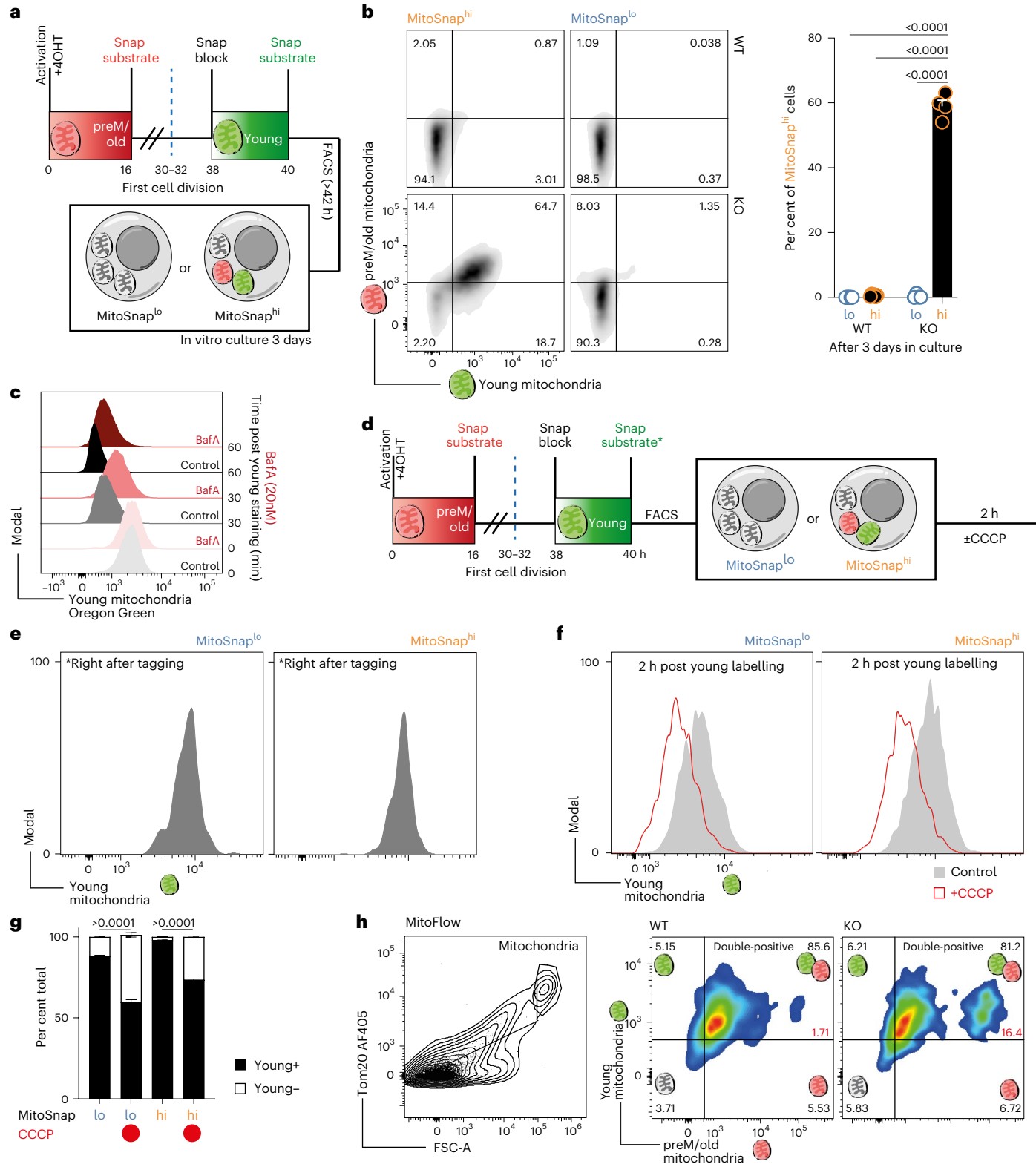

right after young OMP25/mitochondria labelling, which slowed their decay and provided further evidence that autophagy is required for the generation of MitoSnap$^{lo}$ cells (Fig. 3c). When mitophagy was induced using carbonyl cyanide 3-chlorophenylhydrazone (CCCP) (Fig. 3d), both MitoSnap$^{hi}$ and MitoSnap$^{lo}$ first-daughter cells rapidly lost young mitochondrial labelling (Fig. 3e–g), which allowed us to confirm that all live cells underwent mitochondrial biogenesis and to link any SnapTag-labelling loss to mitophagy. These results

corroborated our previous observation that MitoSnap$^{hi}$ cells are still capable of degrading mitochondria if autophagy-sufficient (Fig. 3b). At the organelle level, analysis of mitochondrial fractions by flow cytometry further highlighted differences between autophagy-deficient and autophagy-sufficient cells. In *Atg16l1*-deficient cells mitochondria were on average larger and less heterogeneous (Extended Data Fig. 2c), with a strikingly higher proportion of preM/old brightly labelled structures (16.4%) in comparison with their WT counterparts (1.71%), which

**Fig. 3 | Mitophagy drives the emergence of cell progenies inheriting heterogeneous mitochondrial populations. a**, The experimental layout: CTV-labelled MitoSnap CD8[+] T cells (WT-*Atg16l1*[fl/+] *Omp25*[fl/+] *Ert2*[Cre] or KO-*Atg16l1*[fl/fl] *Omp25*[fl/+] *Ert2*[Cre]) were activated and SnapTag labelled as in Fig. 2a. MitoSnap[lo] cells and MitoSnap[hi] cells were sorted as depicted. The sorted cells were cultured for 3 days in T cell medium supplemented with IL-2, IL-7 and IL-15. **b**, Representative plots from MitoSnap CD8[+] T cells 3 days post sorting. Sorted MitoSnap[lo] cells were used to set up gating strategy. The frequency of double-positive MitoSnap[+] cells (for both preM/old and postM/young) was calculated. The experiment was performed twice with two or three biological replicates ($n_{WT} = 6$; $n_{KO} = 4$). The data are represented as the mean + s.e.m. A statistical analysis was performed using a two-way ANOVA and Tukey's multiple comparisons test. **c**, After young OMP25 staining with Snap-Cell Oregon Green, a pH-sensitive substrate, the cells were cultured in presence or not of BafA (20 nM), an inhibitor of lysosomal function that prevents autophagy flux. The loss of SnapSubstrate Oregon Green was measured at 0, 30 and 60 min post staining. **d**, The experimental layout: CTV-labelled naive MitoSnap CD8[+] T cells (WT-*Atg16l1*[fl/+] *Omp25*[fl/+] *Ert2*[Cre] or KO-*Atg16l1*[fl/fl] *Omp25*[fl/+] *Ert2*[Cre]) were activated on anti-CD3-, anti-CD28- and Fc-ICAM-1-coated plates for 40 h. The cells were cultured in T cell medium containing 500 nM 4OHT. A total of 16 h post

activation, the cells were collected and labelled with Snap-Cell 647-SiR to tag preM/old mitochondria and cultured for a further 24 h, when Snap-Cell Block and Snap-Cell Oregon Green incubations allowed postM/young organelle labelling. The first-daughter cells were sorted and put back in culture for 2 h in presence or not of CCCP. **e**, The efficiency of young labelling to confirm recent mitochondrial biogenesis was performed. **f**, Representative histograms exhibiting young MitoSnap labelling in MitoSnap[hi] and MitoSnap[lo] cells subjected or not mitophagy induction by CCCP. **g**, The frequencies of cells keeping or not young OMP25 labelling after 2 h were quantified. The representative data from one out of two experiments are shown. Each experiment had cells isolated from two or three mice and three technical replicates per condition. The data are represented as the mean ± s.e.m. A statistical analysis was performed using a two-way ANOVA and Tukey's multiple comparisons test. **h**, MitoSnap CD8[+] T cells (WT-*Atg16l1*[fl/+] *Omp25*[fl/+] *Ert2*[Cre] or KO-*Atg16l1*[fl/fl] *Omp25*[fl/+] *Ert2*[Cre]) from three mice were activated and SnapTag-labelled as in Fig. 2a. Mitochondria were purified and phenotyped by flow cytometry. Left: mitochondrial gating was determined based on size and Tom20 expression (also refer to Extended Data Fig. 2C). SnapTag labelling was preserved, and maintenance of old and young organelle staining was evaluated in autophagy-sufficient and autophagy-deficient cells.

---

largely cleared them (Fig. 3h). This important observation reveals that the co-inheritance of young and old OMP25-SnapTag-labelled mitochondria reflects how mitochondrial biogenesis occurs in recently activated CD8[+] T cells, where new structures are mostly added onto the existing mitochondrial network. Furthermore, the absence of bright young organelle labelling suggests that biogenesis occurs at comparable levels in autophagy-sufficient and autophagy-deficient cells post activation, and that differences arise from clearance rather than synthesis. Given that preM/old and postM/young mitochondrial structures are co-inherited at cellular and organelle level, we concluded that cells either maintaining preM/old mitochondria or not clearing postM/young mitochondria in a short timeframe are the same population and from this point were considered MitoSnap[hi] regardless whether young staining was performed or not. Taken together, these results suggest that the emergence of MitoSnap[lo] cells relies both on segregation and degradation events and that autophagy plays a role in both mechanisms.

### Faster mitochondrial turnover predicts memory CD8[+] T cell differentiation

We next asked whether the inheritance of aged mitochondria influences CD8[+] T cell fate in vivo. We generated OT-I CD45.1 MitoSnap mice, in which CD8[+] T cells express a transgenic TCR specific for the OVA[257-264]

SIINFEKL peptide[41], allowing specific activation and tracking of these cells in a host mouse. First, to exclude that mitochondrial inheritance could have been influenced by the nature of TCR-stimulation, we performed experiments using SIINFEKL peptide-loaded dendritic cells[30] and again observed that the emergence of MitoSnap[lo] progenies derives from both degradation and segregation events (Fig. 4a). Phenotypical analysis of resulting first-daughter cells also allowed us to identify early features of fate divergence, such as significantly higher frequencies of CD44[+]CD62L[+] cells (memory fate) within MitoSnap[lo] cells and higher frequencies of CD44[+]CD25[+] cells (effector fate) amongst MitoSnap[hi] cells (Fig. 4a, right). Adoptive transfer of naive OT-I MitoSnap cells followed by *Listeria monocytogenes* expressing OVA (LM-OVA) (Fig. 4b) infection demonstrated that heterogeneous inheritance of preM/old mitochondria also occurs in vivo: MitoSnap[lo] cells being more frequent amongst memory precursors (CD127[+]) (Fig. 4c) and MitoSnap[hi] cells exhibiting higher granzyme B expression (Fig. 4d). To further evaluate long-term memory potential, MitoSnap[hi] and MitoSnap[lo] first-daughter cells were sorted (Extended Data Fig. 3a), transferred into naive hosts and challenged with LM-OVA >4 weeks later, after which OT-I blood kinetics and maintenance were monitored (Fig. 4e). Progenies of MitoSnap[lo] cells consistently showed superior ability to survive and stronger re-expansion potential (Fig. 4f). At the memory-phase MitoSnap[lo] cells were present in higher frequencies,

---

**Fig. 4 | Faster mitochondrial turnover predicts memory CD8[+] T cell differentiation. a**, Left: isolated OT-I CD8[+] T cells were cocultured with SIINFEKL-loaded DC1940 dendritic cells for 40 h. Middle: coculture of OT-I cells with unloaded dendritic cells (DCs) was used as a control to determine undivided CD8[+] T cells, identifed as first division gates. Right: frequencies of CD44[+]CD62L[+] (memory-like) and CD44[+]CD25[+] (effector-like) cells. The data are from two independent experiments (*n* = 10; each data point represents cells from one mouse) are represented as the mean ± s.e.m. **b**, FACS-purified naive MitoSnap[+] OT-I CD8[+] T cells (5 × 10[5]) were transferred to new hosts, followed by infection with *Listeria monocytogenes* expressing SIINFEKL (ovalbumin peptide for which OT-I cells are specific for.) **c**, First-daughter cells were identified based on CTV expression. CD127[+] populations were quantified in MitoSnap[hi] and MitoSnap[lo] progenies (based on preM/old mitochondria expression). A statistical analysis was performed using an unpaired two-tailed Student's *t*-test. The data from two independent experiments, each with three biological replicates, are represented as the mean ± s.e.m. **d**, The frequencies of cells expressing granzyme B (ex vivo) in MitoSnap[hi] and MitoSnap[lo] progenies. A statistical analysis was performed using an unpaired two-tailed Student's *t*-test. The data from two independent experiments, each with two or three biological replicates, are represented as the mean ± s.e.m. **e**, The experimental layout: CTV-labelled naive OT-I MitoSnap CD8[+] T cells (*Atg16l1*[fl/+] *Omp25*[fl/+] *Ert2*[Cre]) were activated on anti-CD3-, anti-CD28- and

Fc-ICAM-1-coated plates for 36–40 h. The cells were cultured in T cell medium containing 500 nM 4OHT. A total of 16 h post activation, the cells were collected and labelled with Snap-Cell 647-SiR to tag preM/old mitochondria and put back in culture. A total of 24 h later, the cells were sorted into MitoSnap[hi] and MitoSnap[lo] cells, and 5 × 10[3] cells were transferred to new hosts (CD45.1 and CD45.2 congenic markers were used to trace transferred cells). More than 30 days following adoptive cell transfer, the host mice were infected with 2,000 colony forming units (CFU) of *Listeria monocytogenes* expressing ovalbumin (OVA) (LM-OVA). Immune responses were evaluated in the blood and spleen. **f**, The frequency of adoptively transferred OT-I cells within CD8[+] T cells in the blood of recipient mice. **g**, The frequency and numbers of adoptively transferred OT-I cells within CD8[+] T cells in the spleens of recipient mice. The representative data are from one (same as in **f**) out of eight experiments, each containing three to five biological replicates per group (please refer to Extended Data Fig. 3b). The data are represented as the mean + s.e.m. A statistical analysis was performed using an an unpaired two-tailed Student's *t*-test. The exact *P* values are depicted. **h**, Left: the frequency of splenic IFN-γ and TNF OT-I producing cells. Right: representative flow cytometry plots of MitoSnap[hi] and MitoSnap[lo] cytokine producing cells. The data from one (*n* = 3 biological replicates) out of two experiments are represented as the mean + s.e.m. A statistical analysis was performed using an unpaired two-tailed Student's *t*-test. The exact *P* values are depicted.

which were also predictive of larger absolute numbers (Fig. 4g and Extended Data Fig. 3b). Although no differences were observed in classical surface marker signatures of memory precursors (KLRG1⁻CD127⁺) or short-lived effector cells (KLRG1⁺CD127) (Extended Data Fig. 3c), functionally MitoSnap^lo progenies produced significantly more IFN-γ and TNF than their MitoSnap^hi counterparts (Fig. 4h). Independent validation using in vitro Tat-Cre driven recombination resulted in similar outcomes (Extended Data Fig. 3d). Together, these findings demonstrate that phenotype and function of MitoSnap^lo CD8⁺ T cells grant them superior memory potential, further emphasizing that asymmetric mitochondrial inheritance provides a mechanistic basis for early fate divergence in CD8⁺ T cells.

## Mitochondrial inheritance patterns control T cell fate commitment

Effector CD8⁺ T cells are highly proliferative, whereas memory CD8⁺ T cells divide slower and remain quiescent. This fate is established early after T cell activation, with cell cycle speed predictive of CD8⁺ T cell fate[23,42]. CD8⁺ T cells also differ metabolically: effector cells rely on glycolysis, while long-lived naive and memory cells perform mitochondrial oxidative phosphorylation (OXPHOS) and fatty acid oxidation to produce adenosine triphosphate (ATP). Mitochondrial architecture influences these processes as fission supports glycolysis, whereas mitochondrial fusion promotes fatty acid oxidation[34,35]. We examined whether MitoSnap^hi and MitoSnap^lo CD8⁺ T cells

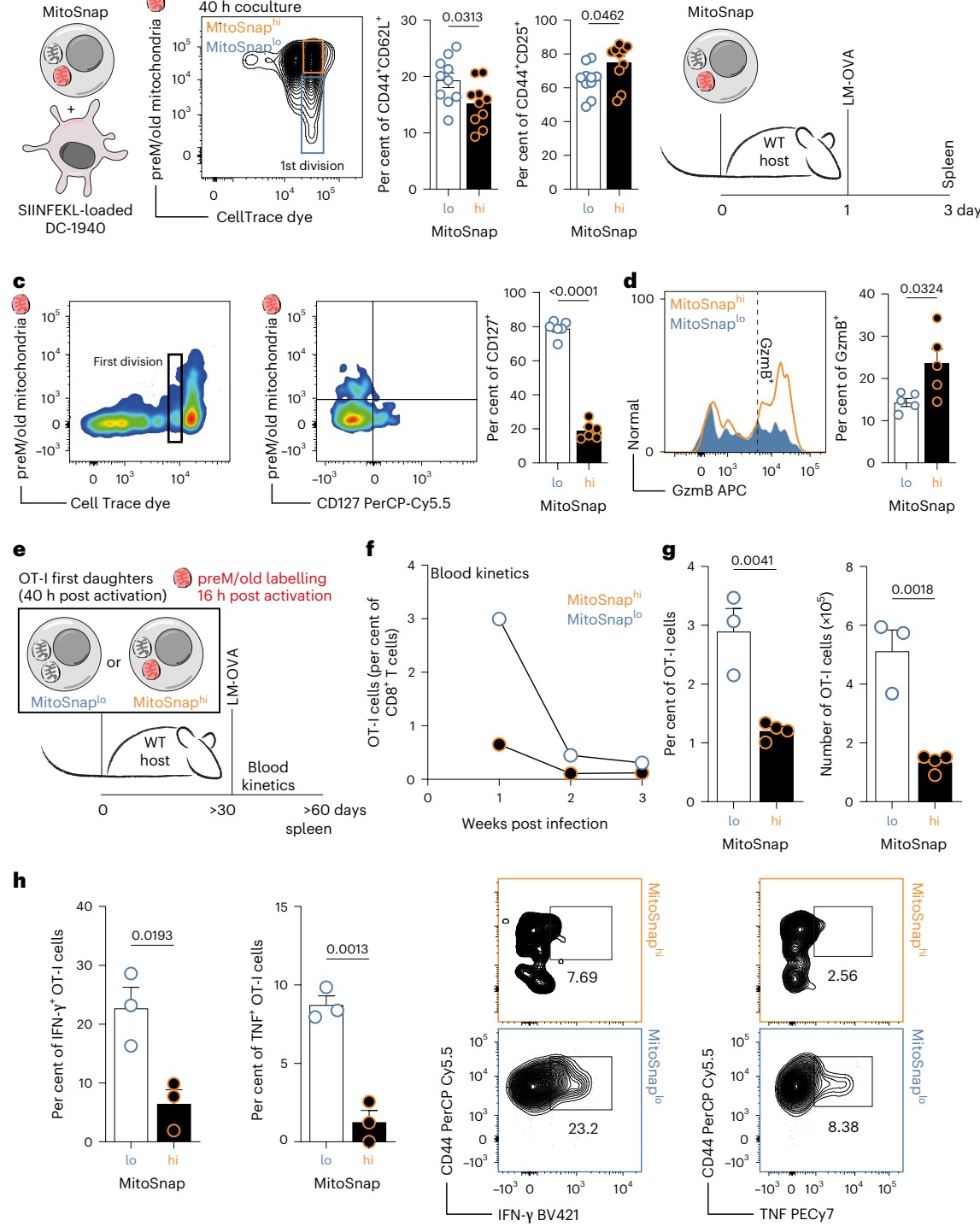

(sorted as in Extended Data Fig. 3a) display distinct mitochondrial dynamics by confocal microscopy. Mitochondrial networks were rendered from three-dimensional reconstructions (Z-stacks) and geometric features from individual mitochondria were quantified (Fig. 5a and Extended Data Fig. 4). Mitochondria from MitoSnap[hi] cells exhibited heterogeneous SnapTag labelling intensity, which we accounted for in our analysis (Fig. 5b). SnapTag[hi] mitochondria from MitoSnap[hi] CD8[+] T cells were significantly larger, less compact and more complex than other mitochondria labelled with lower levels of SnapTag in both cell types (Fig. 5c). These are features of tubular, fused mitochondria, which prevent their degradation by autophagy, while mitigating stress and preserving OXPHOS capacity upon mitochondrial damage[43,44]. Similar patterns were observed for young-OMP25 organelles, confirming co-inheritance of postM/young and preM/old OMP25-SnapTag mitochondria (Extended Data Fig. 5). We then used a modified version of the Scenith assay[45], which measures translation via O-propargyl-puromycin (OPP, a puromycin analogue) incorporation as a proxy for ATP production to perform metabolic dependency measurements. MitoSnap[hi] CD8[+] T cells exhibited higher global translation rates, which were significantly reduced by both glycolysis and OXPHOS inhibitors, while quiescent MitoSnap[lo] cells remained largely unaffected by metabolic modulators (Fig. 5d). When directly measuring oxygen consumption rates (OCR) in autophagy-sufficient and autophagy-deficient MitoSnap[hi] and MitoSnap[lo] cells, we observed no significant differences (Fig. 5e and Extended Data Fig. 6a). Direct measurements of ATP levels further confirmed increased production in cells inheriting preM/old mitochondria, consistent with the conclusions of the Scenith assay (Fig. 5f). To evaluate the functional consequences of inheriting distinct mitochondrial pools, MitoSnap[hi] and MitoSnap[lo] cells were cultured with IL-2, IL-7 and IL-15 in the absence of T cell activation. MitoSnap[hi] cells exhibited faster, more homogeneous proliferation and lower frequencies of slow-dividing cells than MitoSnap[lo] cells, corroborating their less quiescent status that might contribute to premature cell death (Fig. 5g). Autophagy-deficient cells proliferated slower regardless of mitochondrial inheritance, suggesting

a role for autophagy in cell cycle regulation, consistent with reports of autophagy-dependent degradation of cyclin-dependent kinase inhibitor 1B (CDKN1B) in T cells[46]. In cytokine-limiting conditions, as expected for the effector-like population, autophagy-sufficient MitoSnap[hi] cells showed reduced survival (Fig. 5h), while MitoSnap[lo] cells were enriched for CD44[+]CD62L[+] memory-like cells[23,42] (Fig. 5i). Autophagy-deficient cells were poor survivors and did not exhibit differences in phenotype linked to early mitochondrial inheritance (Fig. 5h and Extended Data Fig. 6b,c). Because in WT cells aged mitochondria are cleared after 3 days even in MitoSnap[hi] cells that inherited their mitochondria from the mother cell, our results suggest that organelles inherited at first division influence the decision between cellular quiescence (memory fate) and differentiation (effector fate). Our results provide the first unequivocal data linking organelle inheritance—here mitochondria—in immune cells to changes in cell function that culminate in fate commitment of cells in vivo.

## Unequal inheritance of mitochondrial populations drives changes in the transcriptome and proteome of CD8[+] T cells
To further investigate the metabolism and fate-divergency drivers in daughter cells inheriting distinct mitochondrial pools, we labelled preM/old mitochondria in activated CD8[+] T cells, sorted MitoSnap[hi] and MitoSnap[lo] first-daughter cells (as in Extended Data Fig. 3a) and performed single-cell transcriptomics (scRNA-seq) and bulk proteomics analysis (Fig. 6a). Proteomics allowed us to identify a restricted set of differentially inherited proteins (Fig. 6b). MitoSnap[lo] cells expressed higher levels of Werner protein (WRN), an enzyme important for genome stability[47], and NADH dehydrogenase 4 (mt-ND4), a component of the mitochondrial respiratory chain complex I. MitoSnap[hi] cells were enriched in hypoxia inducible factor 1 subunit alpha (HIF1a) and late endosomal/lysosomal adaptor 2 (LAMTOR2), proteins involved in mammalian target of rapamycin (mTOR) metabolism, which is known to boost effector CD8[+] T cell differentiation, as well as proteins linked to cell cycle progression[48,49]. In previous studies, including our own, transcriptional profiling of first-daughter CD8[+] T cell cells, using

**Fig. 5 | Mitochondrial inheritance patterns control T cell fate commitment.**
**a**, FACS-purified MitoSnap[lo] and MitoSnap[hi] CD8[+] T cells were imaged by confocal microscopy and their mitochondrial architecture was analysed (based on Tom20 labelling to identify organelles). **b**, MitoSnap labelling intensity per individual mitochondria was used to cluster them into four groups. MitoSnap[lo] cells exhibited only mitochondria with low intensity of SnapTag labelling, whereas mitochondria of MitoSnap[hi] cells were arbitrarily stratified into low (lo), intermediate (mid) and high (hi) based on StapTag brightness. The data are represented as the mean ± s.e.m. A total of 1,961 mitochondria were used for further analysis. The pooled data from two independent experiments, each containing sorted cells from two to three biological replicates, are shown. **c**, The geometric parameters were extracted and used to generate uniform manifold approximation and projections (UMAPs) of mitochondria found in MitoSnap[lo] and MitoSnap[hi] cells. The squares in the UMAPs highlight regions with high compactness, area (A)/ volume (V) ratio and complexity. Right: the quantifications of mitochondrial complexity (high values, round organelles), compactness (high values, organelles with tight architecture), volume (V) and sphericity (lower values, high roundness/sphericity) are represented. The data are represented as the mean ± s.e.m. A statistical analysis was performed using a non-parametric Dunn test with Bonferroni correction. The exact P values are depicted (MitoSnap[lo], 972 mitochondria; MitoSnap[hi], 989 mitochondria). **d**, MitoSnap CD8[+] T cells were collected 36–40 h post activation and prepared for the Scenith assay to evaluate their metabolic reliance. OPP incorporation was used as a readout of translation. 2-Deoxy-D-glucose was used to inhibit glycolysis and oligomycin was used to inhibit mitochondrial respiration. A combination of both inhibitors was used to supress both metabolic pathways and obtain an OPP baseline. An analysis was done by flow cytometry, which allowed the discrimination of MitoSnap[hi] and MitoSnap[lo] cells. The data are represented as the mean ± s.e.m. A statistical analysis was performed using a two-way ANOVA with a Tukey's post hoc test. The exact P values are depicted. Representative data

of one out of two experiments are shown. **e**, The OCR of sorted MitoSnap[hi] and MitoSnap[lo] first-daughter CD8[+] T cells was measured under basal conditions and in response to indicated drugs. The data are represented as the mean ± s.e.m. The data points represent two to four biological replicates from independent sorting experiments. **f**, ATP production by sorted MitoSnap[hi] and MitoSnap[lo] first-daughter CD8[+] T cells originally isolated from WT (Atg16l1[fl/+] Omp25[fl/+] Ert2[Cre]) or KO (Atg16l1[fl/fl] Omp25[fl/+] Ert2[Cre]) mice. The data are represented as the mean ± s.e.m. A statistical analysis was performed using an unpaired two-tailed Student's t-test. The exact P values are depicted. The data points represent four biological replicates from independent sorting experiments. **g**, WT and KO MitoSnap[hi] and MitoSnap[lo] cells were sorted as represented in Fig. 2g and cultured for 3 days in T cell medium supplemented with IL-2, IL-7 and IL-15. The frequency of slow-dividing cells (one or two divisions) was calculated. The data are represented as the mean + s.e.m. A statistical analysis was performed using a one-way ANOVA. The exact P values are depicted. The data points represent technical replicates from two pooled biological samples per group ($n_{WT}$ = 6; $n_{KO}$ = 4). The representative data from one out of two experiments are shown. **h**, WT and KO MitoSnap[hi] and MitoSnap[lo] cells were sorted as represented in Fig. 2g and cultured for 7 days in T cell medium supplemented with IL-2, IL-7 and IL-15. The frequency of viable cells was calculated. The data are represented as the mean + s.e.m. A statistical analysis was performed using a one-way ANOVA. The exact P values are depicted. The data points represent four to five technical replicates from three pooled experiments ($n_{WT}$ = 15; $n_{KO}$ = 17). **i**, The frequency of CD44[+]CD62L[+] cells within surviving cells from **g** was calculated. Right: the gating strategy is depicted. The data are represented as the mean + s.e.m. A statistical analysis was performed using an unpaired two-tailed Student's t-test. The exact P values are depicted. The data points represent three to five technical replicates from four biological samples per group (n = 15). The pooled data from three independent experiments are shown.

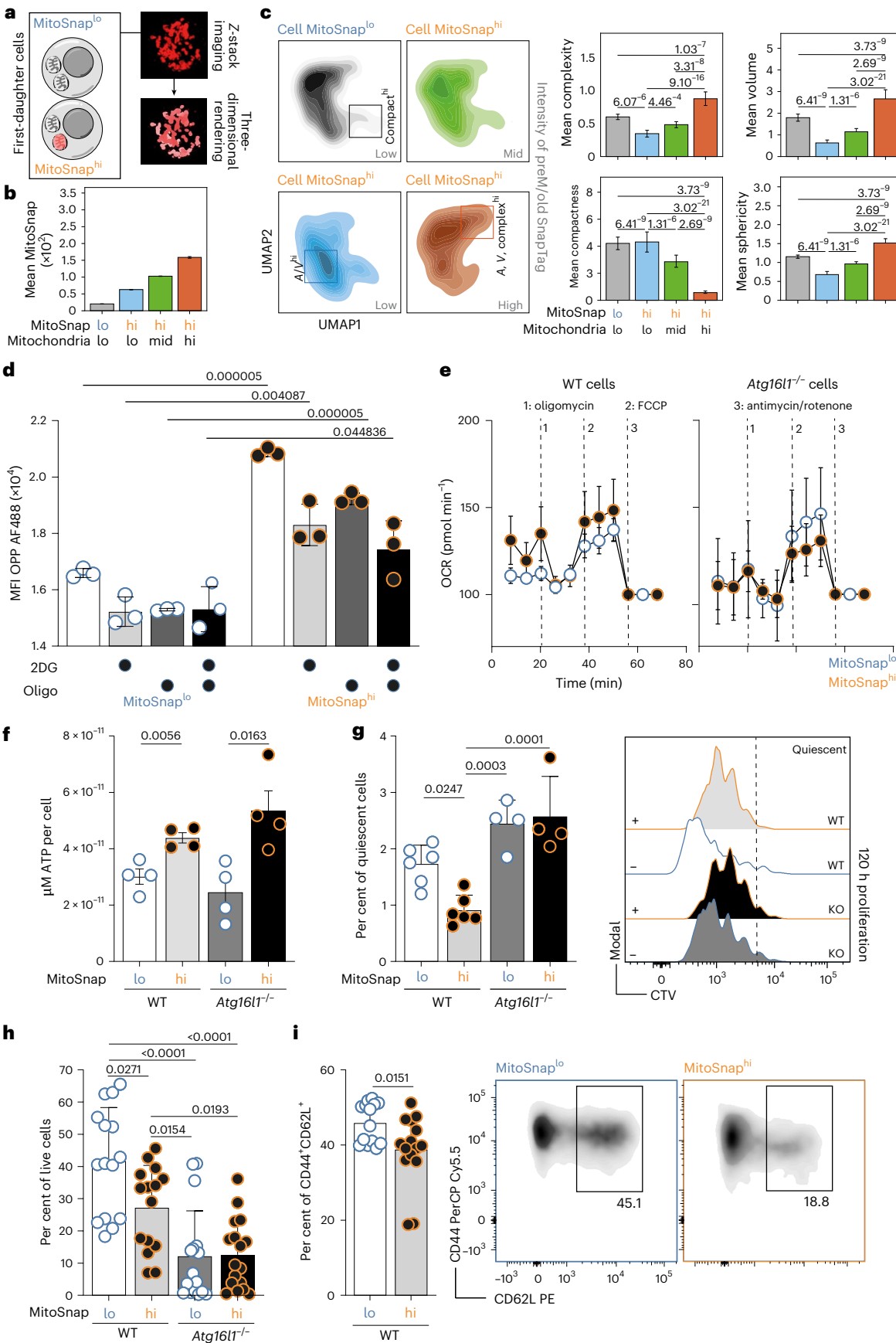

both bulk and single cell strategies relied on the expression of reporter genes or surface markers with dynamic expression or on in vivo models where symmetric and ACDs could not be distinguished[28,30–32,50], which impedes a direct link between transcriptional divergences and asymmetric inheritance of cell fate determinants. Here, unbiased clustering of single cell transcriptomes visualized with UMAPs defined nine clusters (Fig. 6c). Both cell types were distributed across clusters, but some were enriched in MitoSnap[hi] (clusters 2, 6 and 8) or MitoSnap[lo] (clusters 1 and 7) daughter cells. Gene expression of proteomics hits positively correlated with protein levels (Fig. 6d). Furthermore, MitoSnap[hi] proteome signatures aligned with transcriptomes of clusters 2 and 3, whereas MitoSnap[lo] signatures were enriched in clusters 1 and 4. Functional annotation of clusters further highlighted the divergent fates of MitoSnap[hi] and MitoSnap[lo] cells. Based on the genes mostly expressed in each cluster, we could assign functional signatures to each of them. Cluster 1, dominated by MitoSnap[lo] cells, displayed a memory-related signature with *Tcf7* (TCF1) and *Sell* (CD62L) expression, key for stemness maintenance and lymphoid homing. Cluster 7, also enriched in MitoSnap[lo], featured genes linked to mitochondrial function and biogenesis. By contrast, MitoSnap[hi] cells were more abundant in clusters where the gene signature could be linked to effector functions (clusters 5, 6 and 8), showing increased expression of genes involved in cell cycling and nutrient uptake, including higher expression of the serine importer gene *Slc1a5* (SLC1A5) (Extended Data Fig. 7a,b). A pseudotime analysis also allowed us to propose the potential differentiation trajectories of MitoSnap[hi] and MitoSnap[lo] cells, clearly indicating that MitoSnap[hi] exhibited a stronger path towards becoming terminally differentiated cytotoxic cells (Fig. 6e). These results were complemented by a stronger transcriptional signature of genes linked to memory fate commitment in MitoSnap[hi] CD8[+] T cells (Fig. 6f,g). Together, these integrated proteomic and transcriptomic analyses demonstrate that asymmetric mitochondrial inheritance generates divergent metabolic and transcriptional programme, coupling higher mitochondrial turnover rates to enhanced memory potential.

## Maintenance of old mitochondria triggers one-carbon metabolism

When performing our transcriptomics analysis, we also found a cluster enriched in genes that are linked to one-carbon (1C) metabolism (for example *Mthfd2*, *Phgdh* and *Shmt2*, cluster 2). This cluster was formed by a majority of MitoSnap[hi] cells, 1C metabolism signature being also stronger in this population in comparison with MitoSnap[lo] cells (Fig. 7a). In CD4[+] T cells, 1C metabolism is essential for proliferation and effector function as an inducer of mTOR activity[51]. Thus, aiming to functionally validate this finding, we again measured metabolic reliance of MitoSnap[hi] and MitoSnap[lo] sorted cells through Scenith (Fig. 7b), using inhibitors of serine hydroxymethyltransferase (SHMT1/2), methylene-tetrahydrofolate dehydrogenase (MTHFD1/2) and phosphoglycerate dehydrogenase (PHGDH) activity, key enzymes in the folate cycle of 1C metabolism (Fig. 7c). Our results suggest that in MitoSnap[hi] cells, SHMT1/2 and PHGDH inhibitors indeed suppress their translation rates,

a phenotype that was not shared by MitoSnap[lo] cells (Fig. 7d). Because SHMT1/2 inhibition led to the most significant reduction of translation rates in MitoSnap[hi] cells, we evaluated the expression of the SHMT2 enzyme in individual mitochondria by confocal microscopy. We found that organelles exhibiting high expression of old-OMP25-SnapTag labelling are those expressing more SHMT2, which corroborates the hypothesis that changes in 1C metabolism are due to the inheritance of preM/old mitochondria (Extended Data Fig. 8). As serine (Ser) and glycine (Gly) are the main amino acids fuelling 1C metabolism, we sorted MitoSnap[hi] and MitoSnap[lo] cells and cultured them for 3 days in Ser/Gly-depleted conditions (Fig. 7b), after which we could identify a shift in phenotype shared by both subsets that led to lower frequencies of effector-like CD44[+]CD25[+]CD8[+] T cell progenies (Fig. 7e). Ser/Gly restriction also led to increased survival of MitoSnap[hi] CD8[+] T cells in vitro. Because glucose-derived serine can be synthesized by PHGDH, we also kept cells in Ser/Gly-depleted medium combined with PHGDH inhibition, and we did not observe any further increase in survival rates (Fig. 7f). Thus, despite the higher expression of *Phgdh* in MitoSnap[hi] cells and considering the higher expression of nutrient transporters, 1C metabolism seems to mostly rely on amino acid uptake and not on de novo synthesis of serine.

To more definitively dissect metabolic differences between MitoSnap[hi] and MitoSnap[lo] cells, we performed targeted metabolomics of WT and autophagy-deficient first-daughter cells (Fig. 8a and sorting as in Extended Data Fig. 3a). We combined it with flux analysis (24 h) of both [13]C-glucose and [13]C-serine (Fig. 8b). Our results revealed that cells retaining preM older mitochondria, as a consequence of segregation and/or lower mitochondrial turnover, exhibit higher abundance of serine. We also observed a discrete but significant higher flux of [13]C carbon units originating from glucose or serine to these amino acids (Fig. 8c,d). By evaluating the metabolic profile of autophagy-deficient cells, we confirmed that mitochondrial turnover is a major driver of this phenotype, as MitoSnap[hi] and MitoSnap[lo] cells mostly did not exhibit significant differences in serine and glycine abundance and [13]C fraction contribution within these amino acids (Fig. 8d and Extended Data Fig. 9). However, the expression profile of other metabolites appeared to be influenced by the presence of preM/ old mitochondria, even under autophagy-deficient conditions, such as glutathione, pentose phosphate and NAD (Extended Data Fig. 10). Thus, these results further support that inheritance of heterogeneous mitochondrial pools drives T cell fate divergence and this is caused by distinct strategies to fulfil metabolic demands: MitoSnap[lo] cells are more quiescent and quickly turn over mitochondria, which includes mitochondrial biogenesis, while MitoSnap[hi] cells keep old mitochondria, rely on glycolysis and turn on one-carbon metabolism, one of the first consequences after a mitochondrial insult[52].

## Discussion

Most of the previous functional readouts evaluating the role of ACD in early fate decisions have relied on sorting daughter cells based on the expression of CD8 or the transcription factor c-Myc, with CD8[hi]/c-Myc[hi]

---

**Fig. 6 | Unequal inheritance of mitochondrial populations drives changes in the transcriptome and proteome of CD8[+] T cells. a**, The experimental layout: CTV-labelled naive MitoSnap CD8[+] T cells (*Atg16l1*[fl/+] *Omp25*[fl/+] *Ert2*[Cre]) were activated on anti-CD3-, anti-CD28- and Fc-ICAM-1-coated plates for 36–40 h. The cells were cultured in T cell medium containing 500 nM 4OHT. A total of 16 h post activation, the cells were collected and labelled with Snap-Cell 647-SiR (preM/old mitochondria) and cultured for further 24 h. The cells were collected and sorted into MitoSnap[hi] and MitoSnap[lo] populations, and their proteome and transcriptome were analysed. **b**, A volcano plot showing differentially inherited proteins by MitoSnap[hi] and MitoSnap[lo] cells. The data are pooled from six samples done in four independent experiments. MitoSnap[hi]- and MitoSnap[lo]-enriched proteins (represented by their encoding genes) are highlighted in orange and blue, respectively. A statistical analysis was performed using an

unpaired two-tailed Student's *t*-test. **c**, Left: UMAP and clustering of integrated MitoSnap[hi] and MitoSnap[lo] cells obtained from five mice per group. In italic is a selection of the top 50 differentially expressed genes in each cluster, which contributed for cluster classification. **d**, The expression of genes encoding proteins enriched in MitoSnap[hi] or MitoSnap[lo] were projected onto UMAP clusters from **c**. **e**, MitoSnap[hi] and MitoSnap[lo] cells were separately submitted to pseudotime analysis (see Methods for details). Inferred trajectory of each cell type is depicted in black. Cluster 1 was assigned as the origin. **f**, The expression of memory-signature genes in MitoSnap[hi] or MitoSnap[lo] cells were projected onto UMAP clusters from **c**. **g**, The expression of effector-signature genes in MitoSnap[hi] or MitoSnap[lo] cells were projected onto UMAP clusters from **c**. The UMAP projections were extracted from Loupe Cell Browser. The proteomics volcano plot was done using Tableau for **b**.

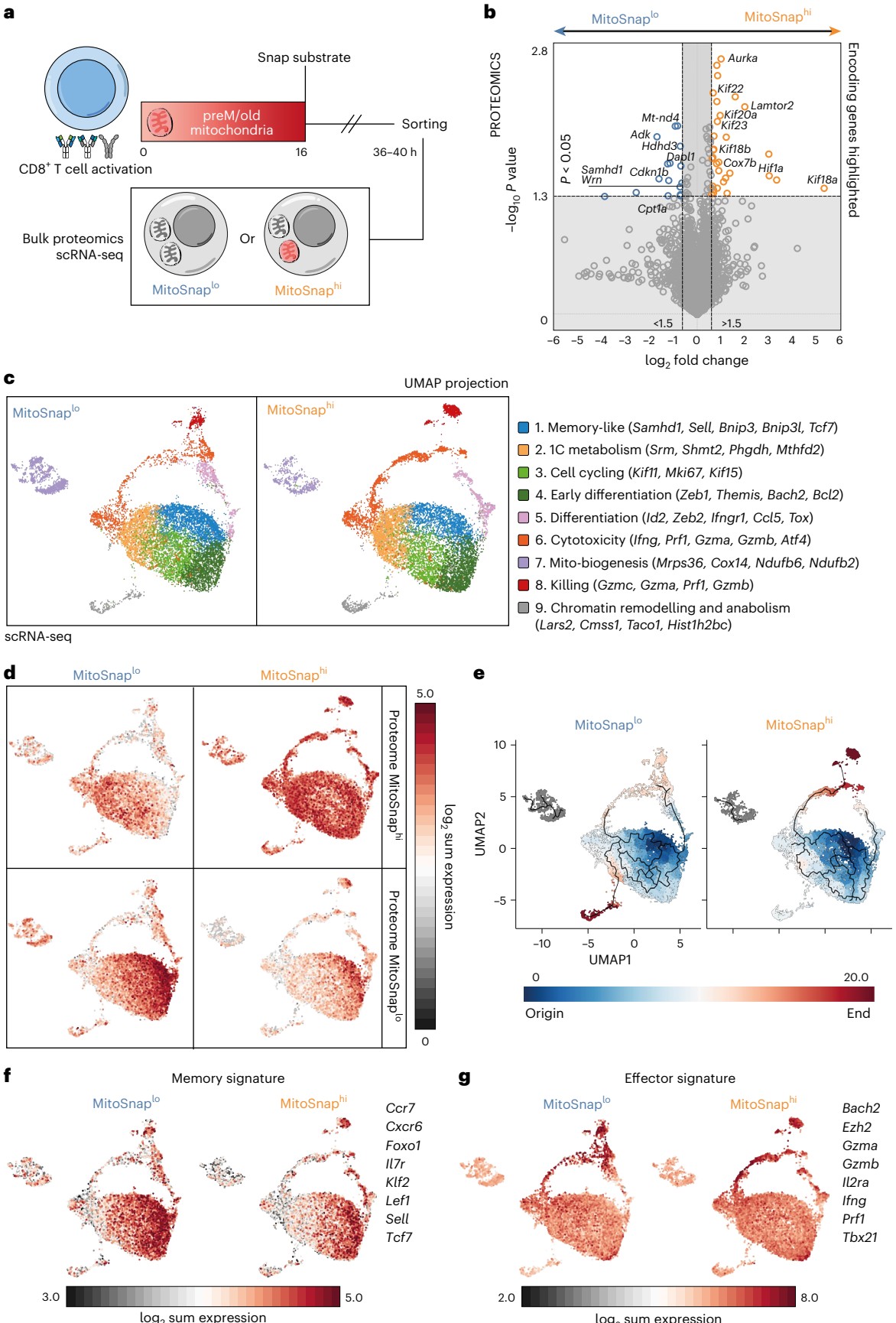

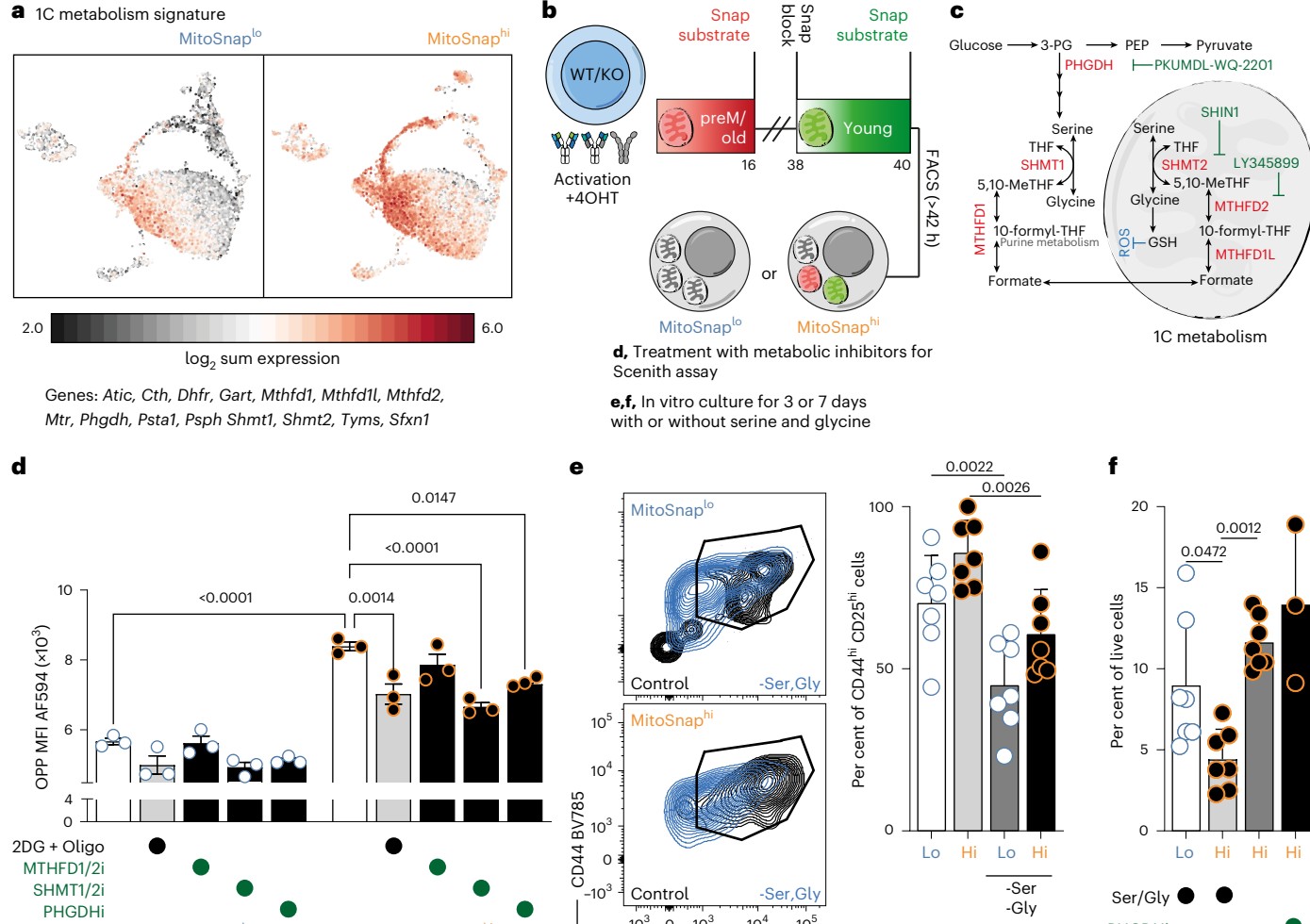

**Fig. 7 | Maintenance of old mitochondria triggers one-carbon metabolism, which can be modulated to rescue terminal differentiation. a**, The genes involved in one-carbon (1C) metabolism were projected onto UMAP clusters. **b**, CTV-labelled naive MitoSnap CD8+ T cells (*Atg16l1*fl/+ *Omp25*fl/+ *Ert2*Cre) were activated on anti-CD3-, anti-CD28- and Fc-ICAM-1-coated plates for 36–40 h. They were cultured in T cell medium containing 500 nM 4OHT. Sixteen hours post activation, the cells were collected and labelled with Snap-Cell 647-SiR (preM/old mitochondria) and cultured for further 24 h. Cells were subsequently collected and prepared for the Scenith assay, aiming to evaluate their metabolic reliance, or cultured in different serine and glycine availability conditions for 7 days. **c**, 1C metabolism summary. In red are depicted the enzymes found to have increased transcripts in MitoSnaphi cells. The inhibitors of specific 1C metabolism enzymes are highlighted in green, with inhibitory symbols. **d**, OPP incorporation was used as a readout of translation. A combination of both 2-deoxy-D-glucose and oligomycin was used to supress both metabolic pathways and obtain an OPP baseline. The inhibitors specific to MTHFD1/2, SHMT1/2 and PHGDH were used to assess reliance on 1C metabolism pathways. An analysis was done by

flow cytometry. The data from three biological replicates are represented as the mean ± s.e.m. A statistical analysis was performed using a two-way ANOVA with a Tukey's post hoc test. The exact *P* values are depicted. Representative data of one out of two experiments are shown. **e**, Sorted MitoSnaphi and MitoSnaplo cells were kept for 3 days in medium depleted of serine (Ser) and glycine (Gly). The frequency of CD44+CD25+ cells within surviving cells was calculated. Left: the gating strategy is depicted. The data from seven biological replicates are represented as the mean + s.e.m. A statistical analysis was performed using a two-way ANOVA with a Tukey's post hoc test. The exact *P* values are depicted. The pooled data from two independent experiments are shown. **f**, The frequency of MitoSnaplo and MitoSnaphi live cells after 7 days in culture under different serine and glycine availability conditions. The data points depict biological replicates (*n* = 7, except for the +PHGDHi condition, where *n* = 3) and are represented as the mean + s.e.m. A statistical analysis was performed using a one-way ANOVA with a Tukey's post hoc test. The exact *P* values are depicted. The pooled data are from two out of three independent experiments.

cells being effector-like and CD8lo/c-Myclo cells being memory-like progenies[22,24,27]. However, the expression of these molecules is highly dynamic and cannot be directly linked to asymmetric segregation events. Asymmetric mTOR activity in effector-like daughter cells has been linked to its translocation to lysosomes and amino acid sensing, but in vivo function readouts relied on correlative CD8 expression[49]. Concerning the asymmetric partitioning of degradation pathways, proteasome activity has been shown to contribute to distinct T-bet distribution between daughter cells, but the results were not directly linked to in vivo T cell fates[25]. A recent pioneering study used genetic barcoding to evaluate the transcriptome of genuine sister cells and

demonstrated that early fate trajectories can be established since first CD8+ T cell division[28]. However, there is currently no evidence to directly link this divergence to the inheritance of a fate determinant. A previous report provides preliminary evidence that mitochondrial clearance can impact on lymphocyte metabolism[36] but relied on the loss of Mitotracker Green FM fluorescence, a labelling that can be perturbed by the oxidative status of the cell, affecting cell viability and prone to cell efflux[53–55]. Thus, we are first to show that the asymmetric inheritance of an unequivocal preM cell cargo, which is regulated by differential autophagy-dependent mitochondrial turnover rates, causes divergent T cell fate commitment. This was possible because the MitoSnap

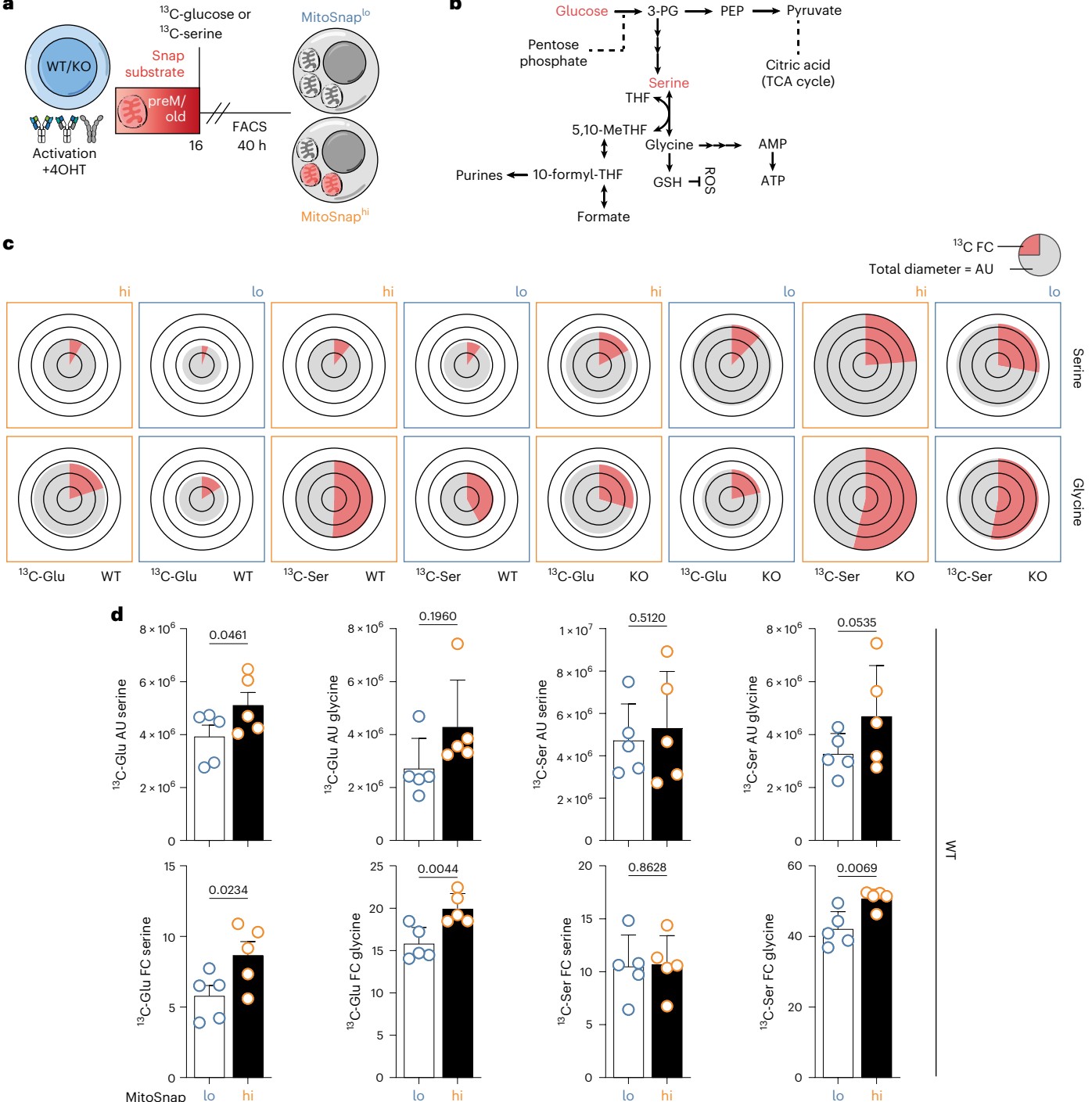

**Fig. 8 | Serine and glycine abundance and $^{13}$C incorporation (metabolic flux) are influenced by mitochondrial inheritance and autophagy status.** **a**, CTV-labelled naive MitoSnap CD8$^+$ T cells (WT-*Atg16l1*$^{fl/+}$ *Omp25*$^{fl/+}$ *Ert2*$^{Cre}$ or KO-*Atg16l1*$^{fl/fl}$ *Omp25*$^{fl/+}$ *Ert2*$^{Cre}$) were activated on anti-CD3-, anti-CD28- and Fc-ICAM-1-coated plates for 40 h. The cells were cultured in T cell medium containing 500 nM 4OHT. A total of 16 h post activation, the cells were collected and labelled with Snap-Cell 647-SiR (preM/old mitochondria) and cultured for further 24 h in presence of $^{13}$C-glucose (2 g l$^{-1}$) or $^{13}$C-serine (30 mg l$^{-1}$). MitoSnap$^{lo}$ and MitoSnap$^{hi}$ cells were sorted and their metabolome was evaluated.

**b**, 1C metabolism summary. Labelled glucose or serine was used to evaluate metabolic flux. **c**, Pie charts were generated using Travis Pies v1.310000. The diameter represents metabolite abundance (AU) and red fractions represent the fraction contribution (FC) of labelled carbon units originating from $^{13}$C-glucose (left half) or $^{13}$C-serine (right half) (metabolic flux). **d**, AU and FC quantification from WT cells in **c** represented as the mean + s.e.m. (see Extended Data Fig. 9 for KO cells). A statistical analysis was performed using a paired two-tailed Student's *t*-test. WT, five biological replicates; KO, three biological replicates.

system allows specific labelling of mitochondria and discrimination of segregation, degradation and biogenesis events that can influence organelle inheritance. Tagging mitochondria before and after mitosis allows precise discrimination of segregation and degradation events,

something which was not achieved in previous reports using dyes, expression of surface markers or reporter genes.

Mitochondria are organelles required to meet the cell's energetic demands. They are the site of OXPHOS, tricarboxylic acid (TCA) cycle

and fatty acid oxidation, pathways involved in the generation of ATP. They are also involved in maintaining the redox balance of the cell, as they can produce ROS, are involved in calcium signalling, can drive apoptotic cell death and, by being core metabolic modulators, also contribute to epigenetic regulation of cell function[5,56]. The results from several studies provide evidence that different T cells subsets exhibit distinct mitochondrial metabolism and function: effector cells are highly glycolytic and memory cells rely on fatty acid oxidation[34,35,57,58]. Accordingly, mitochondrial quality control plays an important role in T cell fate decisions with mitophagy being a crucial regulator of cell survival[17,19,59,60]. Thus, mitochondria constitute a suitable cell cargo to be linked to differentiation trajectories, which was corroborated by our initial proteomics screening identifying mitochondrial-related proteins being differentially enriched in memory-like and effector-like CD8[+] T cell daughters.

The emergence of cells that maintain or lose their MitoSnap labelling during CD8[+] T cell proliferation could result from different cell biological processes, and we dissected the mechanisms underlying the inheritance of mitochondria from the mother cell. First, we identified that ACD contributes to the polarized inheritance of preM/old mitochondria. However, progenies able to clear preM/old mitochondria also rapidly lost their labelling for postM/young mitochondria, suggesting that MitoSnap[lo] cells emerge from both segregation and degradation events, with mitophagy levels being higher in this population. Autophagy and mitophagy support memory CD8[+] T cell responses, but it remained unclear when these mechanisms are required to contribute to the formation of memory precursors or the maintenance of long-lived cells[15–17,19]. To address whether autophagy plays a role in unequal mitochondrial inheritance, we used autophagy-deficient cells and found that asymmetric inheritance of preM/old mitochondria was impaired and, as opposed to autophagy-competent cells, old mitochondria were kept for several days. These results and the more symmetric proteome of CD8[hi] and CD8[lo] progenies from autophagy-deficient or old CD8[+] T cells, corroborate our initial hypothesis and place ACD and autophagy/mitophagy as mechanisms that work synergistically to promote early asymmetric inheritance of cell fate determinants. This was confirmed by our finding that significant differences found between the metabolomes of MitoSnap[hi] and MitoSnap[lo] first-daughter cells were lost when they emerged from autophagy-deficient mother cells.

By following the frequencies of cells inheriting or not preM/old mitochondria (MitoSnap[hi/lo]) over the course of the immune responses it became clear that MitoSnap[lo] cells were more functional memory cells, as they showed better maintenance, re-expansion potential and ability to produce effector-cytokines upon restimulation. This resembles results obtained for CD8[hi]/c-Myc[hi] and CD8[lo]/c-Myc[lo] cells[22,27], with the advantage that we can finally draw a definitive link between the inheritance of a cell cargo that already existed in the mother cell to the biased fate of its progenies. Due to poor survival after adoptive transfer and delayed antigenic challenge, we were unable to assess the in vivo fate of autophagy-deficient progenies, highlighting the importance of mitochondrial turnover for the long-term maintenance of T cells. We then directed our attention to determine what drives the different fates of MitoSnap[hi] and MitoSnap[lo] cells. By using sorted populations or approaches that provide single-cell resolution, we determined that the metabolism, survival and proliferative capacity of these progenies are different. Exhibiting lower translation rates, higher frequencies of slow-dividing cells and CD62L expression and better survival capacity in absence of antigen, MitoSnap[lo] cells clearly showed a stronger memory phenotype than MitoSnap[hi] cells[23,42]. Although, surprisingly, we could not observe significant differences in mitochondrial respiration rates, MitoSnap[hi] cells relied more on glycolysis, a feature seen in effector CD8[+] T cells. As old organelles also produced mitochondrial ROS as measured by MitoSOX, it is reasonable to assume that they have deteriorated mitochondrial fitness and that this might promote their early metabolic shift[34]. Mitophagy has recently been reported

to contribute to memory CD8[+] T cell formation[17]. Our results add to this, showing that mitophagy contributes to the decision for memory CD8[+] T cell fate commitment as early as the first mitosis following CD8[+] T cell stimulation, as directly measured by the rapid loss of young mitochondria in MitoSnap[lo] cells.

Finally, to obtain an unbiased overview of the differences between MitoSnap[lo] and MitoSnap[hi] cells following the first mitosis post naive CD8[+] T cell activation, we performed both bulk proteomics and single cell transcriptomics of these two populations. In line with our expectations, we observed proteins linked to effector cell fate decision in MitoSnap[hi] cells and proteins linked to DNA health and mitochondrial biogenesis in MitoSnap[lo] cells. It also came to our attention that a long list of kinesins (*Kif* genes) was enriched in effector-like MitoSnap[hi] daughters. Kinesins are motor proteins directly involved in intracellular trafficking of cell components along microtubules, which is important for organelle movement and for cell division events[61]. We speculate that these proteins might promote polarization of autophagosomes and mitochondria towards the MitoSnap[hi] daughter cell, but addressing this in T cell is challenging because of their small cytoplasm. Single cell transcriptomics allowed us to identify clusters that were enriched in MitoSnap[hi] and MitoSnap[lo] cells. The presence of a memory-like cluster enriched in MitoSnap[lo] cells, where this signature was also stronger than in MitoSnap[hi] cells, further cements this cell type as the one inheriting memory potential.

Transcriptomics analysis also led us to identify a new metabolic pathway more abundant in MitoSnap[hi] cells, with a signature characterized by the significantly upregulated expression of genes involved in 1C metabolism, which was confirmed by metabolomics analysis. 1C metabolism comprises methionine and folate cycles that provide 1C units to boost de novo synthesis of nucleotides and promote amino acid homeostasis and redox defence, particularly important in dividing cells such as cancer cells[62]. Enzymes involved in 1C metabolism can be found in the cytoplasm and the mitochondria, and both sets were upregulated in MitoSnap[hi] cells. This was confirmed by evaluating the expression of SHMT2 in individual mitochondria by microscopy. However, when comparing the metabolome of MitoSnap[hi] and MitoSnap[lo] cells from WT and autophagy-deficient cells, it became clear that autophagy loss is not solely responsible for the unequal expression of several metabolites, highlighting the synergy between mitochondrial segregation and degradation events in T cell fate commitment. Serine is an important donor of the 1C units when it is converted to glycine, and in CD8[+] T cells, this amino acid has been shown to be important for clonal expansion of effector cells[63]. 1C metabolism has also been directly investigated in different CD4[+] T cell subsets and studies support its role in mTOR activation and the establishment of proinflammatory and highly proliferative populations[51]. Because expression of several amino acid transporters, including serine transporters, is upregulated in MitoSnap[hi] cells, which also exhibit defective translation upon 1C metabolism inhibition, our results provide further evidence of the role of this pathway in effector T cell metabolism and also suggest that it can be effectively targeted to modulate T cell fate decisions.

Collectively, our results support the notion that unequal organelle inheritance plays an important role in CD8[+] T cell fate decision and contributes to the metabolic status of cell progenies. Previous reports using the SnapTag model in epithelial-stem like cell lines found that inheritance of old organelles promotes oxidative metabolism leading to cell differentiation[64]. In intestinal stem cells, progenies inheriting old mitochondria produce elevated levels of α-ketoglutarate, which benefits Paneth cell formation and supports niche renewal[38]. In cells from the haematopoietic lineage, the polarized presence of organelles during mitosis followed by long-term quantitative single-cell imaging has been reported, with the caveat that they were identified by dyes or probes that limit interpretation about their inheritance[65]. Here, we show that organelle inheritance in primary T cells results from both degradation and segregation and that mitophagy and ACD work

synergistically to form early memory-like cells and effector-like cells. As cells retaining (or not) older organelles are endowed with distinct metabolic signatures, our results suggest that therapeutic modulation of T cells can have different outcomes depending on when it is performed. PreM modulation will globally impact on T cell differentiation, and postmitotic approaches can selectively target a certain cell type, memory or effector, by inhibiting or improving its function. We anticipate that these findings will be relevant to a better understanding of how T cell diversity is early imprinted. Whether knowledge gained from this study is likely to be applicable to other cells that undergo ACD is to be investigated. Overall our data will pave the way for the development of more efficient therapeutic strategies in the context of regenerative medicine and particularly important in the context of ageing.

## Online content

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

## Methods

### Study design

This study aimed to evaluate whether organelle inheritance controls CD8[+] T cell differentiation. To achieve that, we investigated the role of ACD and autophagy in patterns of mitochondria inheritance. The novel MitoSnap model was used to allow specific tracking of old versus young organelles. We used imaging analysis of mitotic CD8[+] T cells, flow cytometry readouts that allow single cell resolution, metabolic analysis and unbiased OMICS approaches to measure differences in phenotype and function between MitoSnap[lo] and MitoSnap[hi] progenies. We used adoptive cell transfers of TCR-transgenic OT-I MitoSnap cells coupled to *Listeria monocytogenes*-OVA infections as a tool to assess immune responses and the impact of old organelle inheritance in vivo. All conclusions rely on at least two experiments. No randomization or blinding was used. No statistical methods were used to predetermine sample sizes but our sample sizes are similar to those reported in previous publications from the field, including our own.

### Animal models

All animal work was reviewed and approved by Oxford Ethical committee and the UK Home office under the project licences PPL30/3388 and P01275425. Mice were bred under specific pathogen-free conditions in-house, housed on a 12 h dark–light cycle, with a 30 min period of dawn and dusk and fed ad libitum. The temperature was kept between 20 °C and 24 °C, with a humidity level of 45–65%. Housing cages were individually ventilated and provided an enriched environment for the animals. *Omp25*-SnapTag[fl/fl] mice were kindly provided by the lab of Professor Pekka Katajisto. This strain was then bred with CD45.1 *Atg16l1*[fl/fl] *Ert2*[Cre] OT-I mice expressing a TCR specific for OVA[257–264] SIINFEKL peptide[41] and maintained as CD45.1 or CD45.1/2 mice. Host mice in adoptive transfer experiments were either B6.SJL.CD45.1 or C57BL/6 naive mice. The 6–16-week-old mice were considered young, and >100-week-old mice were considered aged.

### CD8[+] T cell isolation and activation

Spleen and inguinal lymph nodes were collected. Single-cell suspensions were used for naive CD8[+] T cell isolation using EasySep Mouse Naive CD8[+] T Cell Isolation Kit (Stemcell Technologies) following the manufacturer's instructions. Purified populations were cultured (at 37 °C, 5% CO₂) in T cell medium: RPMI-1640 containing HEPES buffer and L-glutamine (R5158, Sigma-Aldrich) supplemented with 10% filtered foetal bovine serum (Sigma-Aldrich), 1× penicillin–streptomycin (Sigma-Aldrich), 1× non-essential amino acids (Gibco), 50 µM β-mercaptoethanol (Gibco) and 1 mM sodium pyruvate (Gibco). T cell activation was done on anti-CD3 (5 µg ml[−1]) (145-2C11, BioLegend), anti-CD28 (5 µg ml[−1]) (37.51, BioLegend) and recombinant human or murine Fc-ICAM-1 (10 µg ml[−1]) (R&D Systems) coated plates. The 36–40 h post activation cells were used in downstream assays. Autophagy deletion and/or SnapTag expression were induced by culturing cells in presence of 500 nM (*Z*)-4-hydroxytamoxifen (Sigma-Aldrich, H7904-5MG). To determine cell division events, cells were stained with Cell Trace Violet (CTV) (Life Technologies) following manufacture's guidelines.

### SnapTag labelling protocol

MitoSnap CD8[+] T cells were labelled in one or three steps. Labelling of preM/old organelles was done by collecting CD8[+] T cells 12–16 h post activation and washing them in PBS (500*g*). The cells were incubated in T cell medium containing the first fluorescent SnapSubstrate for 30 min at 37 °C, washed in PBS and put back in culture in their original wells for further 20–24 h. When postM/young organelle labelling was also performed, cells were collected, washed and incubated with T cell medium containing 5 µM of unlabelled SnapSubstrate (Snap-Cell Block S9106S, New England Biolabs (NEB)) for 30 min at 37 °C. After washing, cells rested for 30–60 min in T cell medium and then were incubated with

the second fluorescent SnapSubstrate for 30 min at 37 °C. Fluorescent cell-permeable Snap-Cell substrates (NEB) were used in the following concentrations: 3 µM (Snap-Cell 647-SiR S9102S), 3 µM (Snap-Cell TMR-Star S9105S) and 5 µM (Snap-Cell Oregon Green S9104S). BafA (20 nM) or CCCP (1 µM) were used to assess the role of autophagy and mitophagy, respectively, in the loss of OMP25 SnapTag labelling.

### Cell survival and proliferation assays

Following activation, isolated MitoSnap CD8[+] T cells (WT versus ATG16L1-deficient or MitoSnap[hi] versus MitoSnap[lo] first-daughter cells) were cultured in T cell medium supplemented with murine IL-2, IL-7 and IL-15 (5 ng ml[−1]). Cell proliferation was evaluated 3 days later, and cell survival was assessed 7 days later. For experiments evaluating the role of serine and glycine depletion, RPMI-1640 was substituted by 1× RPMI-1640 media without glucose, glycine and serine (Teknova), supplemented with 2 g l[−1] of D-glucose (standard concentration found in standard RPMI medium). PHGDH inhibition was done adding 10 µM of PKUMDL-WQ-2201 to the medium.

### Adoptive transfer and immunization

Fluorescence-activated cell sorting (FACS)-purified CTV-labelled MitoSnap[hi] or MitoSnap[lo] cells (equal numbers in the same experiment to allow comparison between the two groups) were intravenously injected into naive recipients. In the following day or >30 days later, mice were infected with 2 × 10[3] colony forming units of *Listeria monocytogenes* expressing ovalbumin (LM-OVA) intravenously. LM-OVA was kindly provided by Prof. Audrey Gerard (Kennedy Institute of Rheumatology, University of Oxford). LM-OVA growth was done from frozen aliquots in Brain Heart Infusion (BHI) broth (Sigma, #53286-100G). Bacteria were used for infections when reaching exponential growth. Immune responses were tracked in the blood and at the memory-phase spleens were collected.

### Immunofluorescence staining and confocal microscopy

At different timepoints post stimulation (preM or mitotic/postmitotic), CD8[+] T cells were collected. In some experiments, cells were incubated with 1–2 µM MitoSOX Mitochondrial Superoxide Indicator (Invitrogen) for 15 min at 37 °C before collection. The cells were washed in PBS and transferred on poly-L-lysine (Sigma-Aldrich) treated coverslips, followed by incubation for 45–60 min at 37 °C. Attached cells were fixed with 2% methanol-free paraformaldehyde in PBS (ThermoScientific) for 10 min, permeabilized with 0.3% Triton X-100 (Sigma-Aldrich) for 10 min and blocked in PBS containing 2% bovine serum albumin (Sigma-Aldrich) and 0.01% Tween 20 (Sigma-Aldrich) for 1 h at room temperature. The following antibodies were used to perform immunofluorescence stainings in murine cells: mouse anti-β-tubulin (Sigma-Aldrich), anti-mouse IgG AF488 (Abcam), anti-CD8 APC 53-6.7 (BioLegend) and anti-LC3B (D11) XP Rabbit mAb PE (Cell Signalling). DAPI (Sigma-Aldrich) was used to detect DNA. ProLong Gold Antifade Mountant (ThermoScientific) was used as mounting medium. Mitotic cells (late anaphase to cytokinesis) were identified by nuclear morphology and/or presence of two microtubule organizing centres and a clear tubulin bridge between two daughter cells. A total of 40–80 *Z*-stacks (0.13 µM) were acquired with a ZEISS 980 Airyscan 2 with a C-Apochromat 63×/1.2 W Corr magnification objective and the ZenBlue software. The data were analysed using Fiji/ImageJ. Thresholds for quantification were set up individually for each fluorophore. Asymmetry rates were calculated based on the integrated density (volume and fluorescence intensity measurements were considered) of cell cargoes inherited by each daughter cell. This was done by using the following calculation: (P1 − P2)/(P1 + P2), where P1 is the daughter cell with higher integrated density of CD8 or old mitochondria. Any values above 0.2 or below −0.2 were considered asymmetric, which corresponds to one daughter cell inheriting at least 1.5× more of a cell cargo than its sibling.

## Planar SLBs

Planar SLBs were made as described previously[66]. In brief, glass coverslips were plasma-cleaned and assembled into disposable six-channel chambers (Ibidi). SLBs were formed by incubation of each channel with small unilammellar vesicles containing 12.5 mol% 1,2-dioleoyl-*sn*-glycero-3-[(*N*-(5-amino-1-carboxypentyl) iminodiacetic acid) succinyl] (nickel salt) and 0.05 mol% 1,2-dioleoyl-*sn*-glycero-3-phosphoethanolamine-*N*-(cap biotinyl) (sodium salt) in 1,2-dioleoyl-*sn*-glycero-3-phosphocholine at total phospholipid concentration 0.4 mM. Chambers were filled with human serum albumin-supplemented HEPES-buffered saline (HBS), subsequently referred to as HEPES-buffered saline/human serum albumin. Following blocking with 5% casein in PBS containing 100 μM $NiSO_4$, to saturate NTA sites, fluorescently labelled streptavidin was then coupled to biotin head groups. Biotinylated 2C11-Fab fragments (30 molecules per square micrometre) and His-tagged ICAM-1 (200 molecules per square micrometre) and CD80 (100 molecules per square micrometre) were then incubated with the bilayers at concentrations to achieve the indicated site densities. Bilayers were continuous liquid disordered phase as determined by fluorescence recovery after photobleaching with a 10 μm bleach spot on an FV1200 confocal microscope (Olympus).

## T cell immunological synapse formation on planar SLBs

Naive murine CD8+ T cells were incubated at 37 °C on an SLB. After 10 min, the cells were fixed with 4% methanol-free formaldehyde in PHEM buffer (10 mM EGTA, 2 mM $MgCl_2$, 60 mM PIPES, 25 mM HEPES, pH 7.0) and permeabilized with 0.1% Triton X-100 for 20 min at room temperature. Anti-CD3 staining was used to identify TCR regions and actin was labelled with fluorescent phalloidin.

## Total internal reflection fluorescence microscopy

Total internal reflection fluorescence microscopy was performed on an Olympus IX83 inverted microscope equipped with a four-line (405 nm, 488 nm, 561 nm and 640 nm laser) illumination system. The system was fitted with an Olympus UApON 150 × 1.45 numerical aperture objective and a Photometrics Evolve Delta EMCCD camera to provide Nyquist sampling. A quantification of fluorescence intensity was performed with ImageJ.

## Flow cytometry

Blood samples used for kinetics analysis were obtained from the tail vein at weeks 1, 2 and 3 post-LM-OVA challenge. At end timepoints, spleens were collected and single-cell splenocytes were prepared by meshing whole spleens through 70 μm strainers using a 1 ml syringe plunger. When cytokine production was assessed, splenocytes were incubated at 37 °C for 1 h with 1 μM of SIINFEKL peptide, followed by 4 h in presence of SIINFEKL peptide + 10 μg ml$^{-1}$ of brefeldin A (Sigma-Aldrich). Specific CD8+ T cells were evaluated by incubation with SIINFEKL$_{257-264}$-APC-labelled or SIINFEKL$_{257-264}$-BV421-labelled tetramers (NIH Tetramer Core Facility at Emory University). Erythrocytes were lysed by red blood cell lysis buffer (Invitrogen). Conjugated antibodies used for surface staining were: anti-CD127 A7R34, anti-CD25 PC61 (AF700, PE-Cy7, BioLegend; APC, eBioscience), anti-CD44 IM7 (AF700, BV785, PE, PerCPCy5.5, BioLegend), anti-CD45.1 A20 (BV785, FITC, PB, BioLegend), anti-CD45.2 104 (AF700, BV711, FITC, BioLegend), anti-CD62L MEL-14 (FITC, BioLegend; eF450, eBioscience), anti-KLRG1 2F1 (BV711, BV785, BioLegend), anti-CD8 53-6.7 (BV510, BV605, FITC, PE, BioLegend) and anti-TCRβ H57-597 (APC-Cy7, PerCPCy5.5, BioLegend). Cells were incubated for 20 min at 4 °C. When intracellular staining was performed, cells were fixed/permed with 2× FACS Lysis Solution (BD Biosciences) with 0.08% Tween 20 (Sigma-Aldrich) for 10 min at room temperature, washed in PBS and incubated for 1 h at RT with anti-IL-2 JES6-5H4 (APC, BioLegend), anti-IFN-γ XMG1.2 (BV421, BioLegend) and anti-TNF MP6-XT22 (PE-Cy7, Thermo Fisher). Identification of viable cells was done by fixable near-IR dead cell staining (Life Technologies).

TMRM (100 nM) and MitoSOX (2 μM) labelling was done before staining of surface markers, by incubating cells for 15 min at 37 °C in T cell medium. All samples were washed and stored in PBS containing 2% FBS (Sigma-Aldrich) and 5 mM of EDTA (Sigma-Aldrich) before acquisition. Stained samples were acquired on a FACS LSR II (R/B/V) or a Fortessa X-20 (R/B/V/YG) flow cytometer (BD Biosciences) with FACSDiva software. Data analysis was done using FlowJo software (FlowJo Enterprise, version 10.10, BD Biosciences).

## Cell sorting (FACS)

After activation, CTV- and SnapSubstrate-labelled MitoSnap CD8+ T cells were collected and stained for phenotypical markers (anti-CD44 IM7, anti-CD45.1 A20, anti-CD45.2 104 and anti-CD8 53-6.7 conjugated to different fluorophores depending on experiment, all BioLegend). Dead cells were excluded by staining cells with a fixable Live/Dead dye (Invitrogen, L34993 or L34957). Subpopulations of interest were sorted on a FACS Aria III cell sorter (BD Biosciences).

## Metabolic reliance measured by protein translation (Scenith)

We used a modified version of the Scenith assay[45], which describes a high correlation between protein translation and ATP production. New protein synthesis was measured using the Click-iT Plus OPP Protein Synthesis Assay (Thermo Fisher, C10456/C10457) according to manufacturer's protocol. In short, the cells were incubated in T cell medium for 30 min at 37 °C without any metabolic inhibitors or in presence of 1 μM oligomycin (Merck), 100 mM 2DG (Merck), a combination of both, 1 μM of SHIN1 (Cambridge Bioscience), 20 μM of PKUMDL-WQ-2201 (Sigma-Aldrich) or 1 μM of LY345899 (Sigma-Aldrich). This was followed by incubation with 10 μM of alkynylated puromycin analogue OPP for 30 min at 37 °C. Click Chemistry was performed with Alexa Fluor 488 or Alexa Fluor 594 dye picolyl azide. Metabolic reliance was assessed by comparing the OPP gMFI, used as an indicator of the relative translation rate, of inhibited samples to the vehicle control.

## Western blot

Following (*Z*)-4-hydroxytamoxifen (Sigma-Aldrich, H7904-5MG) treatment for 24 h and/or BafA treatment (10 nM) for 2 h or not, cells were washed with PBS and lysed in RIPA lysis buffer (Sigma-Aldrich) supplemented with complete Protease Inhibitor Cocktail (Roche) and PhosSTOP (Roche). Protein concentration was calculated by using the BCA assay (Thermo Fisher). Samples were diluted in 4× Laemmli sample buffer (Bio-Rad) and boiled at 100 °C for 5 min. A total of 20 μg protein per sample were used for SDS–polyacrylamide gel electrophoresis analysis. NuPAGE Novex 4–12% Bis–Tris gradient gel (Invitrogen) with MOPS running buffer (Invitrogen) was used. Proteins were transferred to a PVDF membrane (Merck Millipore) and blocked with 5% skimmed milk-TBST (TBS 10× (Sigma-Aldrich) diluted to 1× in distilled water containing 0.1% Tween 20 (Sigma-Aldrich)) for 1 h. Membranes were incubated at 4 °C overnight with primary antibodies diluted in 1% skimmed milk-TBST and at room temperature for 1 h with secondary antibodies diluted in 1% skimmed milk-TBST supplemented 0.01% SDS. Primary antibodies used were: anti-ATG16L1, clone EPR15638 (Abcam, ab187671) and anti-GAPDH and clone 6C5 (Sigma-Aldrich, MAB374). Secondary antibodies used were: IRDye 680LT Goat anti-Mouse IgG (H+L) (Licor, 926-680-70) and IRDye 800CW Goat anti-Rabbit IgG (H+L) (Licor, 926-322-11). Images were acquired using the Odyssey CLx Imaging System. The data were analysed using Image Studio Lite or Fiji.

## Mitochondrial isolation and flow cytometry (MitoFlow)

Autophagy-sufficient (*Atg16l1*$^{fl/+}$ *Omp25*$^{fl/+}$ *Ert2*$^{Cre}$) and autophagy-deficient (*Atg16l1*$^{fl/fl}$ *Omp25*$^{fl/+}$ *Ert2*$^{Cre}$) MitoSnap CD8+ T cells were activated, labelled for preM/old (SNAP-Cell TMR-Star or 647-SiR, NEB) and postM/young organelles (SNAP-Cell Oregon Green, NEB), as previously described, and after 40 h washed with complete T cell medium.

Cell pellets were resuspended in ice-cold mitochondria isolation buffer (320 mM sucrose, 2 mM EGTA, 10 mM Tris–HCl, at pH 7.2 in water) and homogenized with a Dounce homogenizer with a 2 ml reservoir capacity (Abcam). We performed 20 strokes with a type B pretzel. The homogenizer was rinsed with distilled water before each sample was processed to avoid cross-contamination. Differential centrifugation of homogenates was done at 1,000$g$ (4 °C for 8 min), which resulted in a pellet containing whole cells and isolated nuclei first. The supernatant containing the mitochondria was then transferred into new tubes and centrifuged at 17,000$g$ (4 °C for 15 min). Enriched mitochondria, which appeared as brown-coloured pellets, were fixed in 1% paraformaldehyde in 0.5 ml PBS on ice for 15 min, followed by a wash with PBS. Mitochondria were resuspended in blocking buffer containing anti-Tom20-BV421 antibody for 20 min at room temperature. After washing with PBS, mitochondria were resuspended in 250 µl filtered (0.2 µm) PBS and acquired using a BD Fortessa X-20 flow cytometer. The threshold for SSC-A (log-scale) was set to the minimum value (20,000) to allow acquisition of subcellular particles. Submicron particle size reference beads (0.5, 1 and 2 µm, Thermo Fisher Scientific) were also used to identify mitochondria.

### Seahorse assay

MitoSnap$^{hi}$ and MitoSnap$^{lo}$ cells were purified by FACS and their OCR were measured using a XF96 MitoStress Test (Seahorse Agilent, 103015-100). Activated CD8$^+$ T cells were washed in RPMI-1640 without sodium bicarbonate, 10 mM glucose, 1% FCS, 2 mM pyruvate and seeded in a XF plate (Agilent, 103793-100) coated with poly-L-lysine (Sigma-Aldrich) at equal densities in corresponding assay medium (XF Assay Medium, 103680-100) pH 7.4 supplemented with 10 mM glucose, 1 mM sodium pyruvate and 2 mM L-glutamine. Test compounds were sequentially injected to obtain the following concentrations: 1 µM oligomycin, 1.5 µM FCCP, 1 µM rotenone and 1 µM antimycin A. OCRs were normalized to cell number using CyQuant (Molecular Probes).

### ATP synthesis assay

Sorted MitoSnap$^{hi}$ and MitoSnap$^{lo}$ CD8$^+$ T cells were boiled in 100 mM Tris, 4 mM EDTA, pH 7.74 buffer for 2 min at 100 °C. Following centrifugation, the supernatant was used for analysis. ATP levels were assessed using the ATP Bioluminescence Assay Kit CLS II (Roche) following the manufacturer's instructions. The samples and ATP standard mixtures were swiftly combined with an equal volume of luciferase and promptly measured in a luminometer (BMG CLARIOstar Plus microplate reader). Normalization was performed by adjusting values based on the total number of sorted cells. The experiment was performed twice. Each experiment was done with two samples per group (each sample pooled from two biological replicates).

### Proteomics

Proteomics analysis was done as previously described[67]. CD8$^{hi}$ and CD8$^{lo}$ or MitoSnap$^{hi}$ and MitoSnap$^{lo}$ daughter cells following naive CD8$^+$ T cell activation were purified by FACS. The cell pellets were washed 2× in PBS before being stored at −80 °C before proteomics analysis. Samples were resuspended in 200 µl of S-Trap lysis buffer (10% SDS, 100 mM triethylammonium bicarbonate) and sonicated for 15 min (30 s on, 30 s off, 100% amplitude, 70% pulse). The samples were centrifuged and supernatants were transferred to fresh tubes. Protein quantification was done using the Micro BCA Protein Assay Kit (Thermo Fisher). A total of 150 µg of protein was processed using S-Trap mini columns (Protifi, #CO2-mini-80). The samples were digested overnight with 3.75 µg of trypsin (Thermo Fisher, Pierce Trypsin Protease MS-Grade, #90057) with a second digest with the same amount of trypsin for 6 h the following day. The peptides were extracted, dried under vacuum and resuspended to 50 µl with 1% formic acid (Thermo Fisher, #85178) and quantified using the Pierce Quantitative Fluorometric Peptide Assay (Thermo Fisher, #23290).

Peptides were injected onto a nanoscale C18 reverse-phase chromatography system (UltiMate 3000 RSLC nano, Thermo Fisher) and electrosprayed into an Orbitrap Exploris 480 mass spectrometer (MS) (Thermo Fisher). For liquid chromatography the following buffers were used: buffer A (0.1% formic acid in Milli-Q water (v/v)) and buffer B (80% acetonitrile and 0.1% formic acid in Milli-Q water (v/v)). The samples were loaded at 10 µl min$^{-1}$ onto a trap column (100 µm × 2 cm, PepMap nanoViper C18 column, 5 µm, 100 Å, Thermo Fisher) equilibrated in 0.1% trifluoroacetic acid. The trap column was washed for 3 min at the same flow rate with 0.1% trifluoroacetic acid then switched in-line with a Thermo Fisher, resolving C18 column (75 µm × 50 cm, PepMap RSLC C18 column, 2 µm, 100 Å). Peptides were eluted from the column at a constant flow rate of 300 nl min$^{-1}$ with a linear gradient from 3% buffer B to 6% buffer B in 5 min, then from 6% buffer B to 35% buffer B in 115 min, and finally from 35% buffer B to 80% buffer B within 7 min. The column was then washed with 80% buffer B for 4 min. Two blanks were run between each sample to reduce carry-over. The column was kept at a constant temperature of 50 °C. The data was acquired using an easy spray source operated in positive mode with spray voltage at 2.60 kV, and the ion transfer tube temperature at 250 °C. The MS was operated in DIA mode. A scan cycle comprised a full MS scan ($m/z$ range from 350 to 1,650), with RF lens at 40%, AGC target set to custom, normalized AGC target at 300%, maximum injection time mode set to custom, maximum injection time at 20 ms, microscan set to 1 and source fragmentation disabled. MS survey scan was followed by MS/MS DIA scan events using the following parameters: multiplex ions set to false, collision energy mode set to stepped, collision energy type set to normalized, HCD collision energies set to 25.5%, 27% and 30%, orbitrap resolution 30000, first mass 200, RF lens 40%, AGC target set to custom, normalized AGC target 3,000%, microscan set to 1 and maximum injection time 55 ms. The data for both MS scan and MS/MS DIA scan events were acquired in profile mode.

Analysis of the DIA data was carried out using Spectronaut Biognosys, AG (version 14.7.201007.47784 for CD8$^{hi}$ and CD8$^{lo}$ cells obtained from young, *Atg16l1*-deficient and old mice; version 17.6.230428.55965 for MitoSnap$^{hi}$ and MitoSnap$^{lo}$ cells). The data were analysed using the direct DIA workflow, with the following settings: imputation, profiling and cross run normalization were disabled; data Filtering to Qvalue; Precursor Qvalue Cutoff and Protein Qvalue Cutoff (Experimental) set to 0.01; maximum of two missed trypsin cleavages; PSM, protein and peptide false-discovery rate levels set to 0.01; cysteine carbamidomethylation set as fixed modification and acetyl (N-term), deamidation (asparagine, glutamine), oxidation of methionine set as variable modifications. The database used for CD8$^{hi}$ and CD8$^{lo}$ cells was mouse_swissprot_isoforms_extra_trembl_06_20.fasta (2020-06) and for mitosnap samples was the *Mus musculus* proteome obtained from uniprot.org (2022-02). Data filtering, protein copy number and concentration quantification was performed in the Perseus software package, version 1.6.6.0. Copy numbers were calculated using the proteomic ruler as described[33]. The samples were grouped according to the condition. Outliers were excluded following analysis of histone counts in each sample. The *P* values were calculated via a two-tailed, unequal-variance *t*-test on log-normalized data. Elements with *P* values <0.05 were considered significant, with a fold-change cut-off >1.5 or <0.67.

### Single cell transcriptomics

Single cell RNA sequencing libraries were prepared using the Chromium Single Cell 3′ GEX v3.1 assay (10x Genomics). In short, cell suspensions were encapsulated into gel beads in emulsion using the Chromium Controller. Within each gel bead in emulsion, cell lysis and barcoded reverse transcription of RNA occurred, followed by complementary DNA amplification. The amplified cDNA underwent library construction via fragmentation, end-repair, A-tailing, adaptor ligation and index PCR. Final libraries were sequenced on an Illumina NovaSeq 6000 system. Initial data processing was conducted with Cell Ranger 7.2.0.

Filtered output matrices were processed using Seurat. After loading the data and assigning unique identifiers to each dataset, cells with more than 10% mitochondrial gene content were excluded to ensure data quality. The datasets were normalized using SCTransform, and PCA was conducted for dimensionality reduction. Integration of the datasets was achieved using the Harmony algorithm, followed by clustering and differential expression analysis. Finally, the integrated data were visualized using UMAP. UMAP projections of resulting clusters and gene expression of gene sets were generated using Loupe Browser. This methodology enabled a robust analysis while accounting for technical variations and maintaining biological integrity.

### Trajectory analysis and visualization

To investigate the pseudotemporal dynamics of our single-cell RNA sequencing data, we utilized Monocle3[68], an advanced tool designed for trajectory inference and differential expression analysis. In previous steps, using the Seurat package, the cells were preprocessed, and size factors were estimated and normalized. Dimensionality reduction was performed using UMAP to capture the intrinsic structure of the data, followed by clustering to identify distinct cell populations. These clusters were annotated based on known marker genes, and the cluster annotations were integrated into the metadata for further analysis.

For trajectory inference, we converted the Seurat object to a Monocle3 object[69]. Separate trajectory analyses were conducted for the MitoSnap$^{hi}$ and MitoSnap$^{lo}$ original identities. Each subset of cells was reclustered using UMAP, and principal graphs were constructed to capture the developmental trajectories of the cells. Pseudotime values were then computed to represent the progression of cells along these inferred trajectories.

### Mitochondrial architecture analysis

MitoSnap$^{hi}$ and MitoSnap$^{lo}$ cells were FACS isolated and prepared for immunofluorescence as described above. Z-stacks (0.13 µm) were acquired with a ZEISS 980 Airyscan 2 with a C-Apochromat 63×/1.2 W Corr magnification objective and the ZenBlue software. Image voxels were made isotropic via upscaling using the 'zoom' function of the SciPy Multidimensional image package[70]. A feathering algorithm, to pick up on mitochondria outlines, in combination with multi-Otsu thresholding and a custom set of operators (that is, erosions, closing and filling of holes) from scikit-image (Python) were used for segmentation of individual mitochondria[71]. Both the Tom20 and MitoSnap channels were combined to enable identification of all mitochondria. Segments smaller than 500 pixels (noise) were excluded from further analysis. Separate organelles were given a unique identifier using the 'label' function from scikit-image. Each segmented organelle was used to create a mask on the isotropic pixel converted images, which allowed quantification of fluorescence intensity values (calculated as the average pixel intensity/number of pixels in the segmented volume). Segmented organelles had their geometric parameter measurements extracted and standardized, and included sphericity, roundness, compactness, surface area[72] and volume. Mitochondrial complexity was calculated as previously described[73].

The data were visualized with UMAP. Only geometric parameters were used to remove any bias in the analysis. A Gaussian kernal density estimate was plotted to produce the contour/density map of UMAP points for visualization of mitochondria distribution shifts. Mitochondria were grouped and analysed for geometric parameters based on their cell of origin (MitoSnap$^{hi}$ versus MitoSnap$^{lo}$) and intensity of SnapTag labelling. MitoSnap$^{lo}$ cells exhibited only mitochondria with low intensity of SnapTag labelling. In MitoSnap$^{hi}$ cells, low intensity was determined by intensity values below the maximum intensity value found in MitoSnap$^{lo}$ cells; mid intensity was determined as values above this threshold, but below the 75th percentile value found in MitoSnap$^{hi}$ cells; high mitochondria were identified using SnapTag labelling intensity values above the 75th percentile value for found in MitoSnap$^{hi}$ cells.

### Metabolomics

CTV-labelled naive MitoSnap CD8$^+$ T cells (WT-$Atg16l1^{fl/+}$ $Omp25^{fl/+}$ $Ert2^{Cre}$ or knockout (KO)-$Atg16l1^{fl/fl}$ $Omp25^{fl/+}$ $Ert2^{Cre}$) were activated on anti-CD3, anti-CD28 and Fc-ICAM-1 coated plates for 40 h. The cells were cultured in T cell medium containing 500 nM Z-4-hydroxytamoxifen (4OHT). Then, 16 h post activation, the cells were collected and labelled with Snap-Cell 647-SiR (preM/old mitochondria) and cultured for further 24 h in presence of $^{13}$C-glucose (2 g l$^{-1}$) or $^{13}$C-serine (30 mg l$^{-1}$). Metabolites were extracted from cell pellets of sorted MitoSnap$^{hi}$ and MitoSnap$^{lo}$ daughter cells. A total of 10 µl of each sample was loaded into a Dionex UltiMate 3000 LC System (Thermo Scientific Bremen, Germany) equipped with a C18 column (Acquity UPLC -HSS T3 1.8 µm; 2.1 × 150 mm, Waters) coupled to a Q Exactive Orbitrap MS (Thermo Scientific) operating in negative ion mode. A step gradient was carried out using solvent A (10 mM TBA and 15 mM acetic acid) and solvent B (100% methanol). The gradient started with 0% of solvent B and 100% solvent A and remained at 0% B until 2 min post injection. A linear gradient to 32.3% B was carried out until 7 min and increased to 36.3% until 14 min. Between 14 and 26 min, the gradient increased to 90.9% of B and remained at 90.9% B for 4 min. At 30 min the gradient returned to 0% B. The chromatography was stopped at 40 min. The flow was kept constant at 0.25 ml min$^{-1}$ and the column was placed at 40 °C throughout the analysis. The MS operated in full scan mode ($m/z$ range 70.0000–1,050.0000) using a spray voltage of 4.80 kV, capillary temperature of 300 °C, sheath gas at 40.0 and auxiliary gas at 10.0. The AGC target was set at $3 × 10^6$ using a resolution of 70,000, with a maximum IT fill time of 512 ms. Data collection was performed using the Xcalibur software (Thermo Scientific). The data analyses were performed by integrating the peak areas (El-Maven–Polly, Elucidata).

### Statistical analysis

To test if data point values were in a Gaussian distribution, a normality test was performed before applying parametric or non-parametric statistical analysis. When two groups were compared, paired or unpaired Student's t-test or Mann–Whitney test were applied. When comparisons were done across more than two experimental groups, analyses were performed using a Dunn test with Bonferroni correction or one-way analysis of variance (ANOVA) or two-way ANOVA with a post hoc Tukey's test multiple testing correction. The P values were considered significant when <0.05, and the exact P values are provided in the figures. All analyses were done using GraphPad Prism 10 software.

**Inclusion and ethics statement.** We affirm our commitment to ethical integrity, fairness, and inclusion by engaging diverse collaborators and local researchers throughout the study, while carefully and transparently describing author contributions.

### Reporting summary

Further information on research design is available in the Nature Portfolio Reporting Summary linked to this article.

## Data availability

The mass spectrometry proteomics data have been deposited to the ProteomeXchange Consortium via the PRIDE partner repository with the dataset identifier PXD053316. Single cell transcriptomics data have been deposited at GEO (GSE270704). Raw data generated or analysed in this study are available from the corresponding lead author on reasonable request. Source data are provided with this paper.

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

## Acknowledgements

We thank D. Cantrell CBE, FRSE, FRS, FMedSci, A. Howden and the FingerPrints Proteomics Core Facility of the University of Dundee for their support with proteomics analysis. We thank T. Conrad, C. Fischer, C. Dietrich, F. Solinas and C. Braeuning from the BIH/MDC Genomics Platform for their support in generating the scRNA-seq data. We thank E. Johnson (Dunn School, University of Oxford) for performing electron microscopy experiments. We thank P. C. Moreira, D. Andrew and M. Medghalchi (Kennedy Institute of Rheumatology BSU staff) for their support. We thank L. Uhl and H. Cai for helping with LM-OVA infections. This work was funded by grants from the Wellcome Trust (investigator award 103830/Z/14/Z and 220784/Z/20/Z to A.K.S., Sir Henry Wellcome Fellowship 220452/Z/20/Z to M.B. and PhD studentship award 203803/Z16/Z to F.C.R.), the Helmholz association (Helmholtz Distinguished Professorship Funding to recruit top-level international female scientists (W3) to A.K.S.), the European Union's Horizon 2020 (Marie Sklodowska-Curie grant agreement number 893676 to M.B. and ERC-2021-SyG_951329 to E.C.B. and M.L.D.), the Swiss National Science Foundation (Early Postdoc. Mobility P2EZP3_188074 to M.B.), the European Molecular Biology Organization (EMBO LT postdoctoral fellowship—ALTF1155-2019 to A.V.L.V.) and the Kennedy Trust for Rheumatology Research (KTTR) to Y.F.Y. and M.L.D. Flow cytometry was performed at the KTTR-supported facility at the Kennedy Institute of Rheumatology (Oxford), and light microscopy was performed at the Oxford-ZEISS Centre of Excellence in Biomedical Imaging.

## Author contributions

M.B., A.V.L.V. and A.K.S. designed the experiments. M.B., A.V.L.V., E.B.C., Y.Y., F.C.R., S.R. and R.M. performed the experiments. T.Y., H.B., E.K., L.S., M.L.D. and P.K. provided expert assistance and guidance. M.B., A.V.L.V., A.H.K., E.J. and E.B.C. analysed the experiments. M.B. prepared figures. M.B. and A.K.S. wrote the paper.

## Competing interests

The authors declare no competing interests.

## Additional information

**Extended data** is available for this paper at https://doi.org/10.1038/s41556-025-01835-2.

**Correspondence and requests for materials** should be addressed to Mariana Borsa or Anna Katharina Simon.

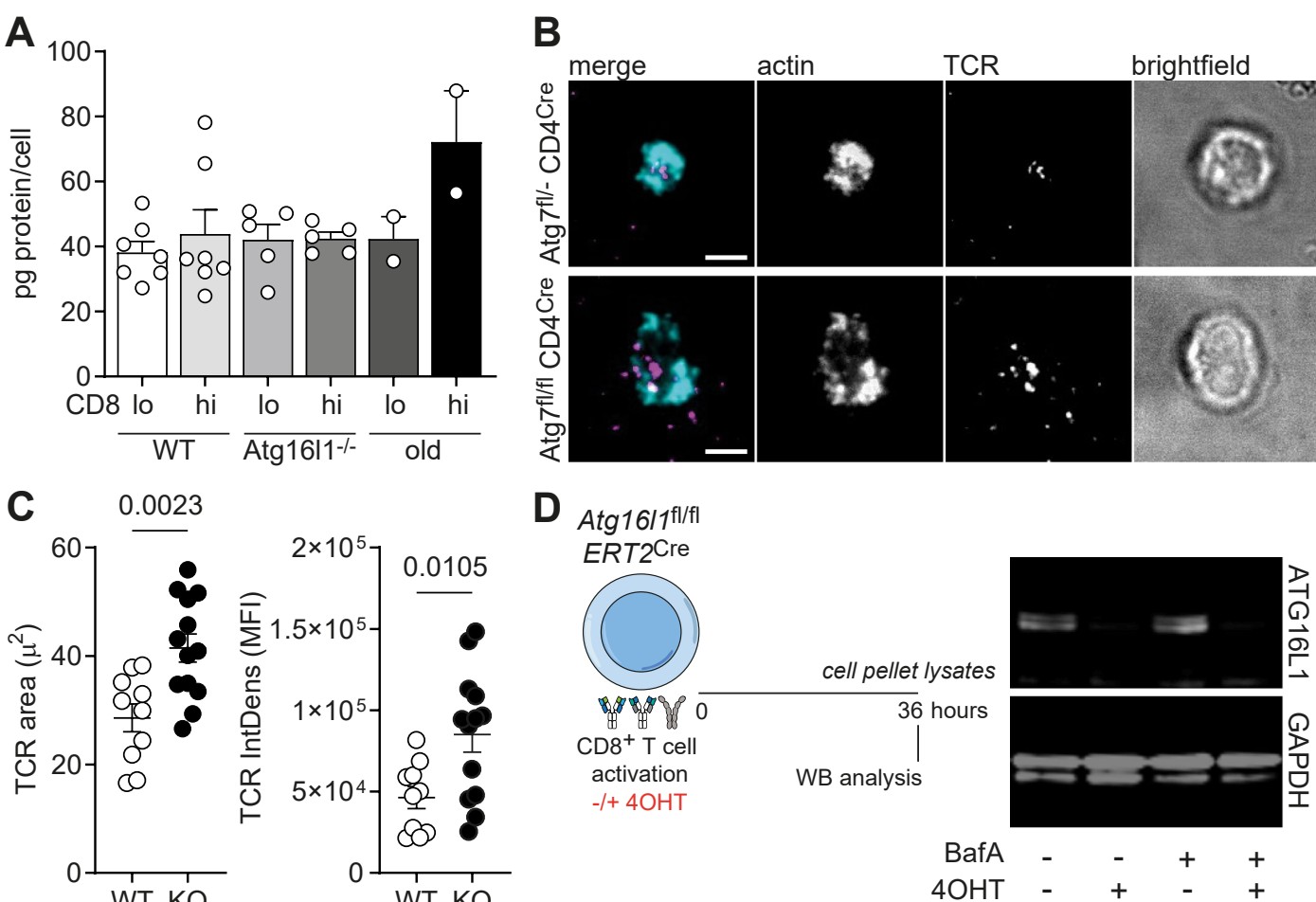

**Extended Data Fig. 1 | Constitutive autophagy-deficiency changes immune synapse architecture, which makes inducible-deletion of *Atg16l1* a more suitable model to evaluate the impact of asymmetric inheritance in CD8+ T cells. (A)** CD8hi and CD8lo first-daughter cells were sorter and sent for proteomics analysis. Protein concentration per cell was comparable across groups. Data are represented as mean ± SEM. **(B)** Representative TIRF-images of immunological synapses of autophagy-sufficient and -deficient cells. Naïve CD8+ T cells were added on a planar supported lipid bilayer (PSLB) containing anti-TCR, ICAM-1, and CD80. Cells were fixed after 10 min and stained for CD3 and actin. Scale bar represents 1 μm. **(C)** Measurements of TCR area and intensity at immunological synapses. Data are represented as mean ± SEM and originated from 2 biological replicates. Statistical analysis was performed using an unpaired two-tailed Student's *t* test. Exact *P* values are depicted in the figure. **(D)** *Atg16l1*fl/fl *Ert2*Cre CD8+ T cells were isolated and activated on anti-CD3, anti-CD28 and human-Fc-ICAM-1 coated plates. Cells were cultured in medium containing (Z)-4-Hydroxytamoxifen (4OHT). After 36 h, cells were harvested and treated or not with Bafilomycin A (BafA) for 2 h. Cell lysates had their ATG16L1 expression determined by immunoblotting. Data representative of 1 out of 4 experiments.

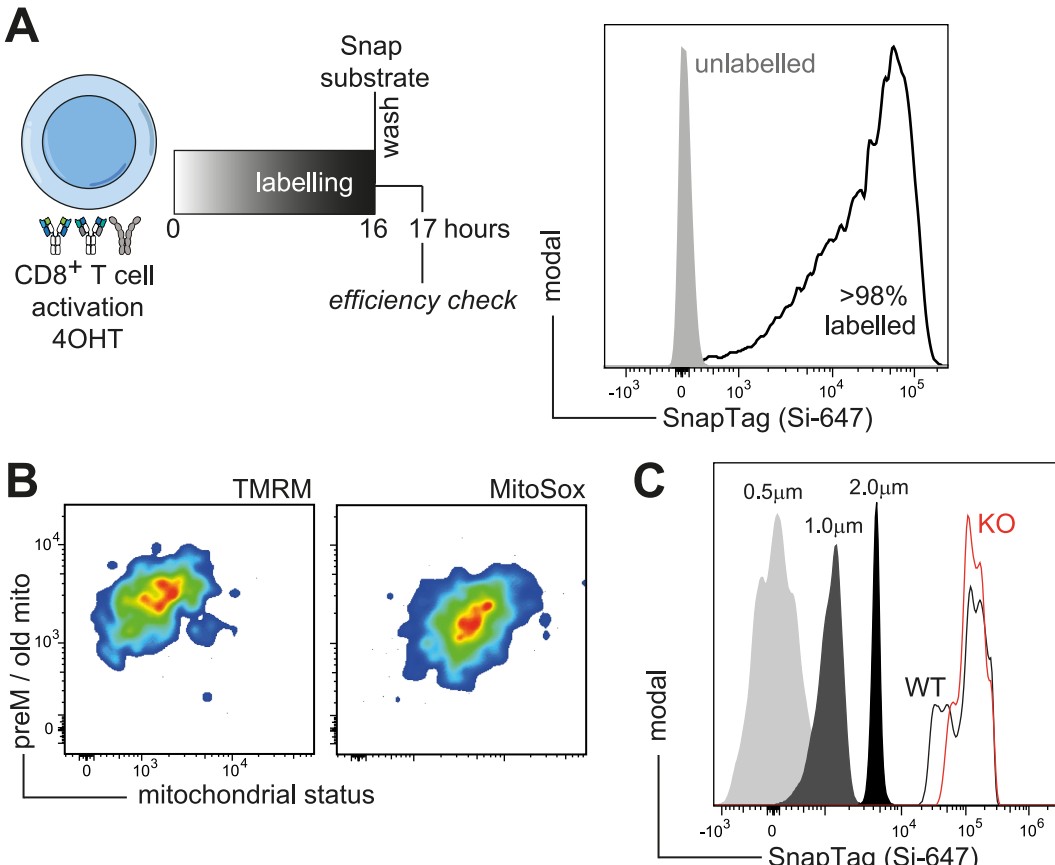

**Extended Data Fig. 2 | SnapTag labelling is highly efficient and resistant to mitochondrial enrichment. (A)** Naïve MitoSnap CD8+ T cells (*Omp25*-SnapTag[fl/−] *Ert2*[Cre]) were cultured in medium containing (Z)-4-Hydroxytamoxifen (4OHT) and activated for 16 h prior to harvesting and labelling with a cell permeable SnapSubstrate (SNAP-Cell® 647-SiR). Efficiency of labelling was assessed by flow cytometry 30 min after substrate washing. **(B)** Representative plots of co-expression of Tetramethylrhodamine, methyl ester (TMRM) or MitoSox in first-daughter MitoSnap cells tagged with SNAP-Cell® 647-SiR 16 h

post-activation. Experiment had 2 biological replicates. **(C)** Representative histograms exhibiting beads of known size (0.5 µm, 1 µm and 2 µm) and mitochondrial populations enriched from *Omp25*-SnapTag[fl/−] *Ert2*[Cre] (WT) and *Omp25*-SnapTag[fl/−] *Atg16l1*[fl/fl] *Ert2*[Cre] (KO) cells. Cells (obtained from 3 biological replicates) were activated for 40 h and labelled with two SnapSubstrates (pre-mitotic (preM)/old and post-mitotic (postM) young mitochondria) as represented in Fig. 2a.

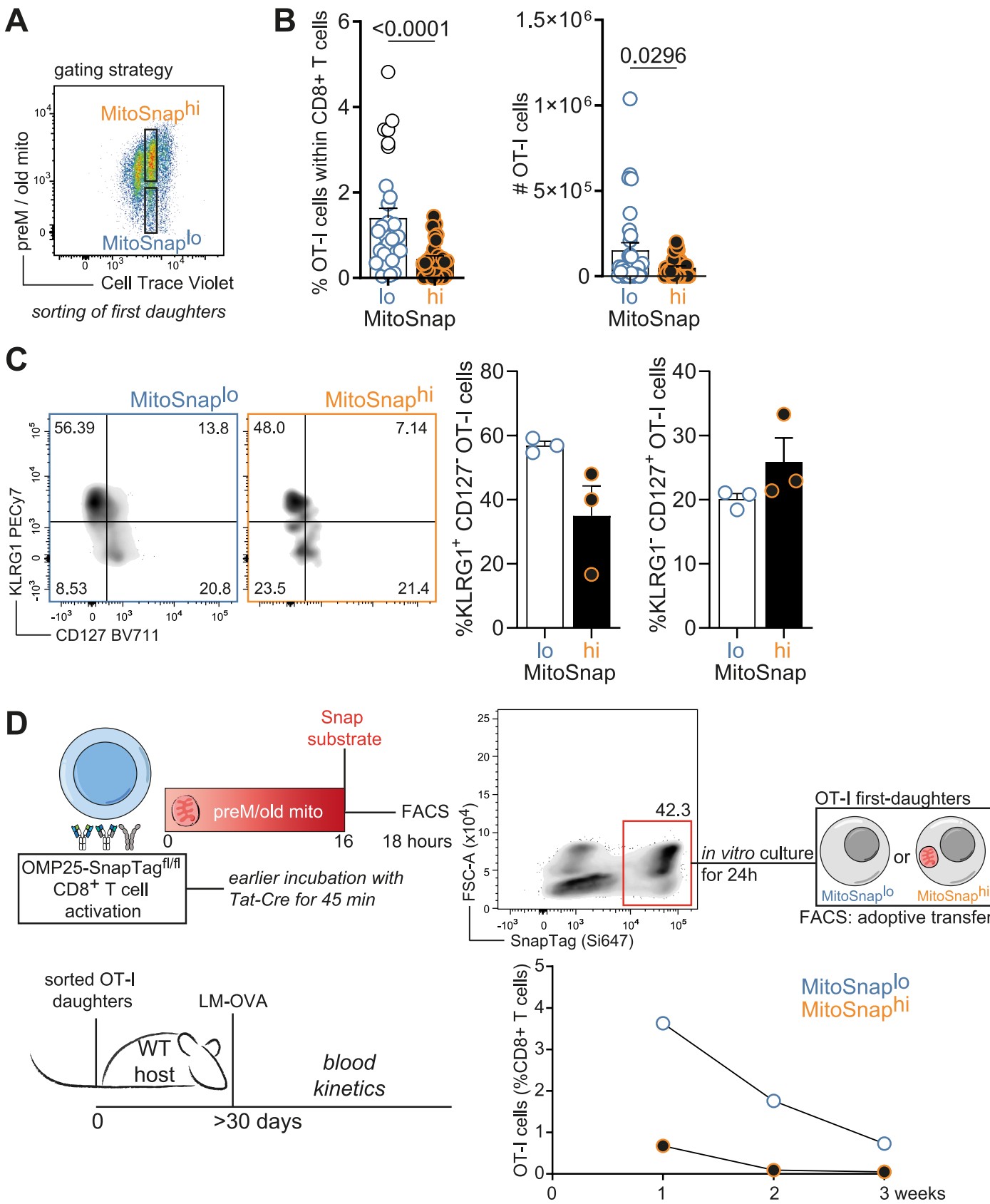

**Extended Data Fig. 3 | See next page for caption.**

**Extended Data Fig. 3 | MitoSnap^lo and MitoSnap^hi progenies show distinct survival and re-expansion rates, do not differ phenotypically at the memory-phase following cognate-antigen challenge, and the Tat-Cre recombinase system replicates results obtained using _Ert2_^Cre system. (A)** Experimental layout: CTV labelled naïve MitoSnap CD8 + T cells were activated on anti-CD3, anti-CD28 and Fc-ICAM-1 coated plates for 40 h. Cells were cultured in T cell medium containing 500 nM Z-4-Hydroxytamoxifen (4OHT). 16 h post-activation, cells were harvested and labelled with Snap-Cell 647-SiR to tag preM/old mitochondria and cultured for a further 24 h. Cells were harvested and sorted as in the representative plot into MitoSnap^hi and MitoSnap^lo populations. **(B)** Frequency within CD8^+ T cells and numbers of adoptively transferred OT-I cell progenies in the spleens of recipient mice (pooled data from 8 experiments where 5-50×10^3 cells were adoptively transferred, n = 30). Statistical analysis was performed using a two-tailed Mann-Whitney test. **(C)** OT-I MitoSnap CD8^+ T cells (_Omp25_-SnapTag^fl/fl _Ert2_^Cre) were activated, labelled for preM/old and postM/young mitochondria (Fig. 2a) and sorted into MitoSnap^hi and MitoSnap^lo prior to adoptive transfer (5×10^4 cells intravenously) into new hosts. Progenies emerging

from MitoSnap^hi and MitoSnap^lo cells were monitored over the course of an immune response against _Listeria monocytogenes_ expressing OVA (LM-OVA) (Fig. 3a). At 30 days post-challenge, phenotype of OT-I cells was evaluated by the expression of KLRG1 and CD127. Frequencies of short-lived effector cells (KLRG1^+ CD127^-) and memory-committed (KLRG1^- CD127^+) were calculated. Gating strategy is depicted on the right. Representative data of 1 (n = 3 biological replicates) out of 3 experiments. Data are represented as mean ± SEM. **(C)** Tat-cre driven-recombination was used as an alternative for the 4OHT-driven recombination using _Ert2_^Cre. Recombination efficiency was evaluated by SnapTag labelling. Cells that did express _Omp25_-SnapTag up to 16 h post Tat-cre recombination were sorted and cultured for further 24 h. First-daughter OT-I MitoSnap^hi and MitoSnap^lo cells were sorted and used in adoptive transfer experiments (2×10^4 cells intravenously) (similar to Fig. 3a). Progenies emerging from MitoSnap^hi and MitoSnap^lo cells were monitored over the course of an immune response against _Listeria monocytogenes_ expressing ovalbumin (OVA) (LM-OVA). Representative data of 1 (n_hi=4, n_lo = 5) out of 2 experiments.

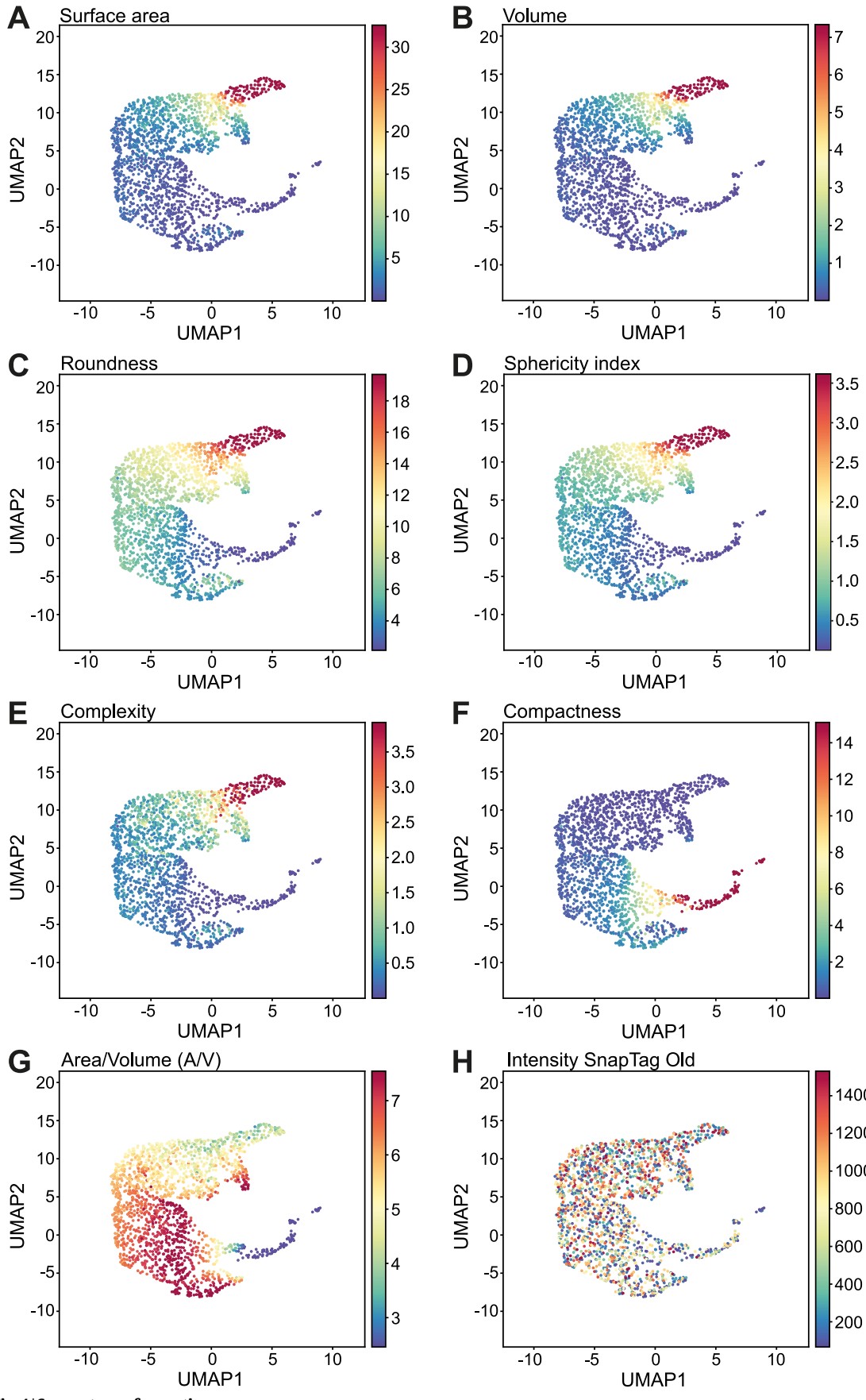

**Extended Data Fig. 4 | See next page for caption.**

**Extended Data Fig. 4 | UMAP projections from individual mitochondria reveal their heterogeneity.** OT-I MitoSnap CD8$^+$ T cells (*Omp25*-SnapTag$^{fl/fl}$ *Ert2*$^{Cre}$) were activated, labelled for preM/old mitochondria and, following one division cycle (first-daughter cells), sorted into MitoSnap$^{hi}$ and MitoSnap$^{lo}$ populations. Cells were prepared for immunofluorescence and imaged by confocal microscopy. Individual mitochondria were segmented and had their geometric parameter measurements extracted and standardized. Fluorescence intensity of preM/ old mitochondria SnapTag labelling was also quantified. UMAP projections (generated using geometrical parameters) highlighting: **(A)** surface area; **(B)** volume; **(C)** roundness; **(D)** sphericity; **(E)** complexity; **(F)** compactness; **(G)** area/volume; **(H)** intensity of preM/old SnapTag labelling in individual mitochondria. 1961 mitochondria were used to generate these projections. Pooled data from 2 independent experiments, each containing sorted cells from 2-3 biological replicates.

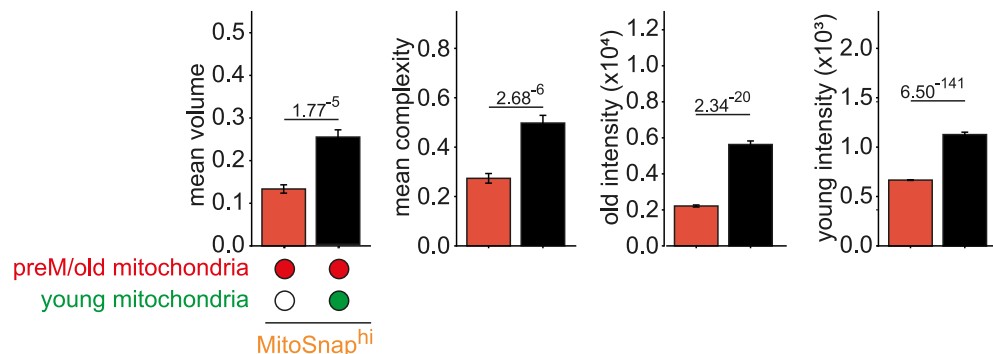

**Extended Data Fig. 5 | Maintenance of young OMP-25 preferentially occurs in less compact mitochondria structures.** FACS-purified MitoSnap[hi] CD8[+] T cells from 3 biological replicates (based on preM/old and young MitoSnap expression) were imaged by confocal microscopy and had the mitochondrial architecture of young[+] (n = 1262) and young[−] (n = 1171) networks compared. This was accompanied by quantification of MitoSnap labelling (young and preM/old OMP-25-SnapTag) intensities in each individual mitochondrion. Statistical analysis was performed using a non-parametric Dunn test with Bonferroni correction. Exact P values are depicted in the figure.

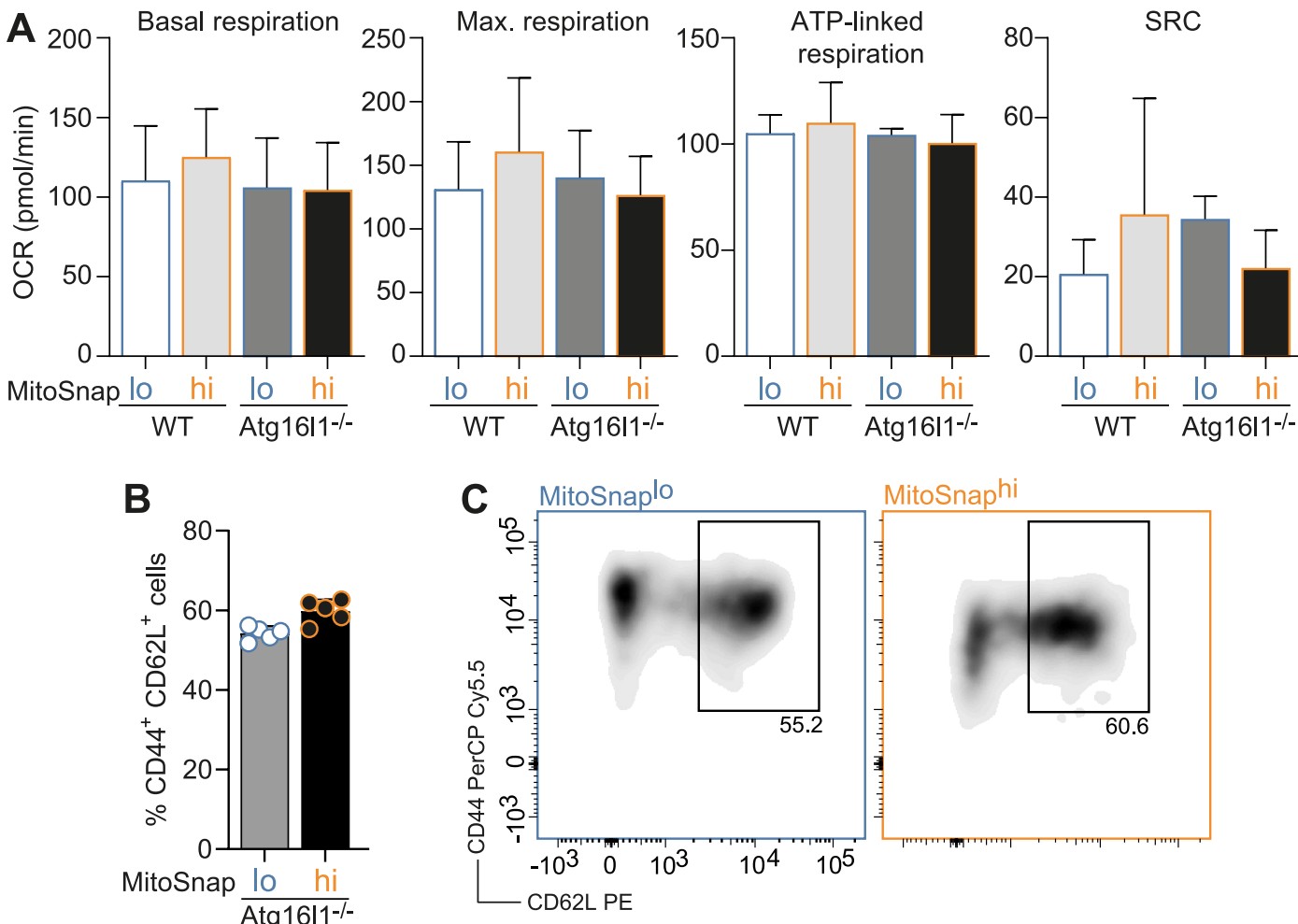

**Extended Data Fig. 6 | MitoSnap$^{hi}$ and MitoSnap$^{lo}$ progenies show similar oxygen consumption rate and autophagy-deficient daughter-cells do no differ phenotypically in survival assays. (A)** OT-I MitoSnap CD8$^+$ T cells (*Omp25*-SnapTag$^{fl/fl}$ *Ert2*$^{Cre}$ or *Omp25*-SnapTag$^{fl/-}$ *Atg16l1*$^{fl/fl}$ *Ert2*$^{Cre}$) were activated, labelled for preM/old and postM/young mitochondria (Fig. 2a) and sorted into MitoSnap$^{hi}$ and MitoSnap$^{lo}$ cells. Their oxygen consumption rate (OCR) was then measured using a XF96 MitoStress Test. Basal respiration, maximal respiration, ATP-linked respiration and spare respiratory capacity (SRC) were calculated. 2-4 biological replicates from independent sorting experiments. Data are represented as

mean ± SEM. **(B)** MitoSnap CD8$^+$ T cells (*Omp25*-SnapTag$^{fl/-}$ *Atg16l1*$^{fl/fl}$ *Ert2*$^{Cre}$) were activated, labelled for preM/old and postM/young mitochondria (Fig. 2a), sorted into MitoSnap$^{hi}$ and MitoSnap$^{lo}$ populations after 36-40 h and cultured for further 7 days in medium containing IL-2, IL-7 and IL-15. Surviving cells had their phenotype evaluated concerning the co-expression of CD44 and CD62L. **(C)** Representative plots showing gating strategy. Data are represented as mean ± SEM. Datapoints represent 5 technical replicates from 1 biological sample per group. Representative data from 1 out of 3 experiments (total of 3 biological replicates across experiments).

**A** cell cycling

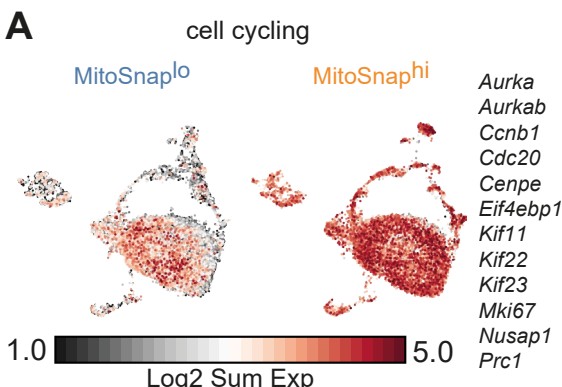

MitoSnap^lo MitoSnap^hi

*Aurka*
*Aurkab*
*Ccnb1*
*Cdc20*
*Cenpe*
*Eif4ebp1*
*Kif11*
*Kif22*
*Kif23*
*Mki67*
*Nusap1*
*Prc1*

1.0 — 5.0
Log2 Sum Exp

**B** nutrient transport

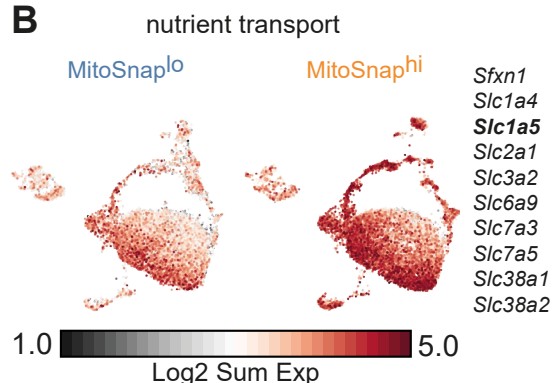

MitoSnap^lo MitoSnap^hi

*Sfxn1*
*Slc1a4*
**Slc1a5**
*Slc2a1*
*Slc3a2*
*Slc6a9*
*Slc7a3*
*Slc7a5*
*Slc38a1*
*Slc38a2*

1.0 — 5.0
Log2 Sum Exp

**Extended Data Fig. 7 | MitoSnap^hi and MitoSnap^lo progenies show distinct transcriptional profile.** OT-I MitoSnap CD8⁺ T cells (*Omp25*-SnapTag^fl/fl *Ert2*^Cre) were activated, labelled for preM/old and postM/young mitochondria (Fig. 2a) and sorted into MitoSnap^hi and MitoSnap^lo populations. Single-cell transcriptomics analysis revealed that genes linked to **(A)** cell cycling and **(B)** involved in nutrient transport are preferentially found in MitoSnap^hi CD8⁺ T cells. UMAP projections were extracted from Loupe Cell Browser.

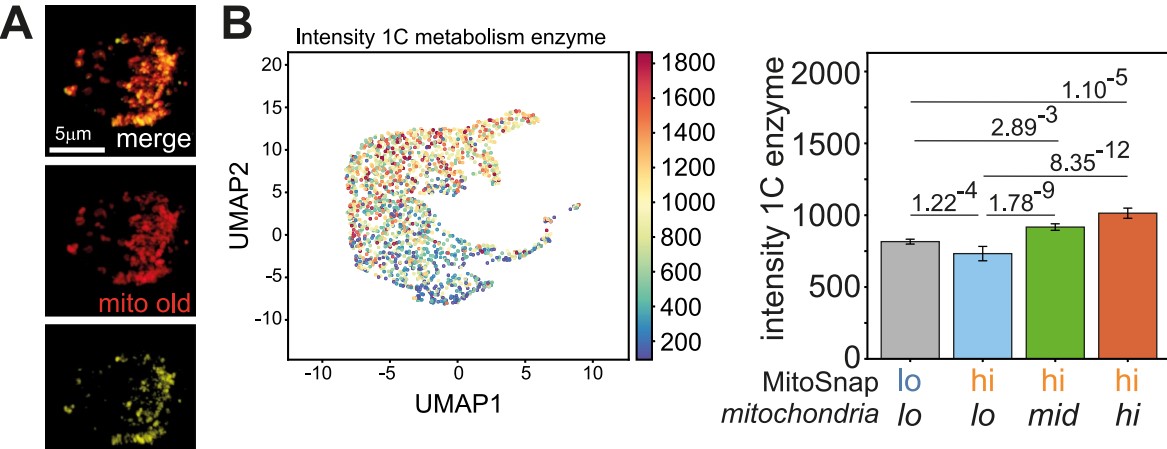

**Extended Data Fig. 8 | Pre-mitotic old SnapTag$^{hi}$ mitochondria express the highest levels of 1 C metabolism enzymes.** FACS-purified MitoSnap$^{lo}$ and MitoSnap$^{hi}$ CD8$^+$ T cells were imaged by confocal microscopy and had the expression of 1C metabolism enzymes (SHMT2 and MTHFD2) evaluated (same cells used to extract geometric parameter measurements). **(A)** Representative confocal microscopy images showing pre-mitotic/old SnapTag labelling and anti-SHMT2 staining in a MitoSnap$^{hi}$ cell with heterogeneous mitochondria. **(B)** UMAP projections (generated using geometrical parameters) highlighting expression of the 1 C enzyme SHMT2 in individual mitochondria (left panel). 1961 mitochondria were used to generate these projections. Higher values indicate higher intensity. Quantification of SHMT2 MFI is represented as mean ± SEM (right panel). Statistical analysis was performed using a non-parametric Dunn test with Bonferroni correction. Exact $P$ values are depicted in the figure. Pooled data from 2 independent experiments with pooled cells from at least 2-3 biological replicates.

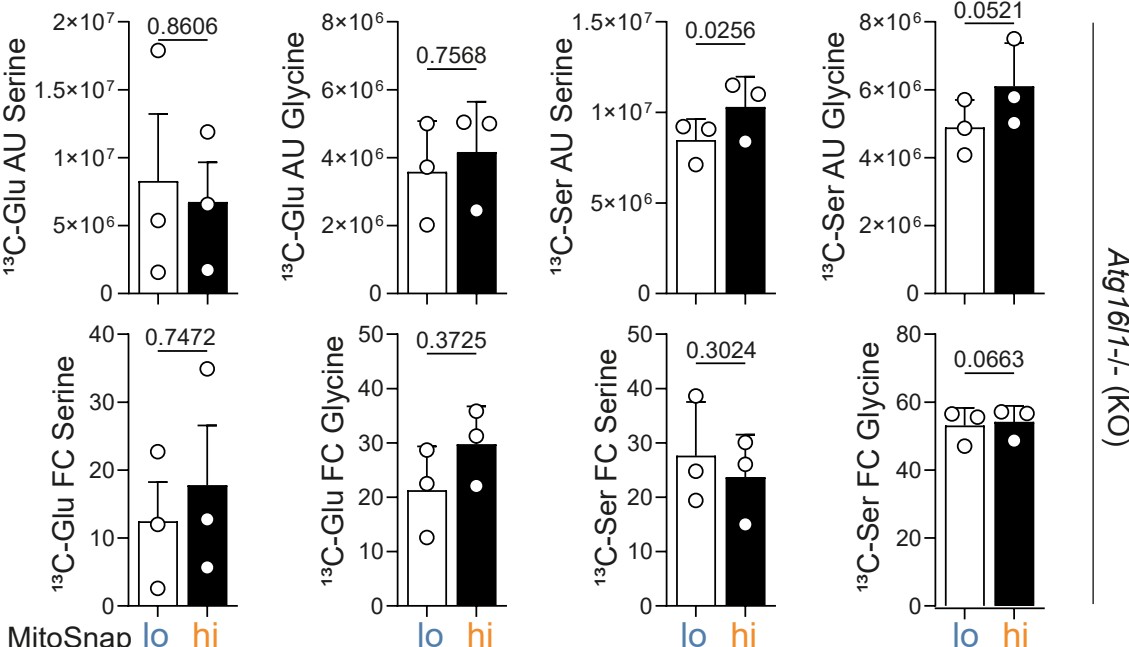

**Extended Data Fig. 9 | Abundance and flux of the amino acids serine and glycine in autophagy-deficient cells.** CTV labelled naïve autophagy-deficient MitoSnap CD8[+] T cells (*Atg16l1*[fl/fl] *Omp25*[fl/+] *Ert2*[Cre]) were activated on anti-CD3, anti-CD28 and Fc-ICAM-1 coated plates for 40 h. Cells were cultured in T cell medium containing 500 nM Z-4-Hydroxytamoxifen (4OHT). 16 h post-activation, cells were harvested and labelled with Snap-Cell 647-SiR (old mitochondria) and cultured for further 24 h in presence of [13]C Glucose (2 g/L) or [13]C Serine

(30 mg/L). MitoSnap[lo] and MitoSnap[hi] cells were sorted and their metabolome was evaluated. Quantification of abundance and [13]C fraction contribution originating from [13]C-Glucose or [13]C-Serine in ATG16L1-KO MitoSnap[hi] and MitoSnap[lo] progenies (refer to Fig. 8a-c). Data are represented as mean ± SEM. Statistical analysis was performed using paired two-tailed Student's *t* test. n = 3 biological replicates.

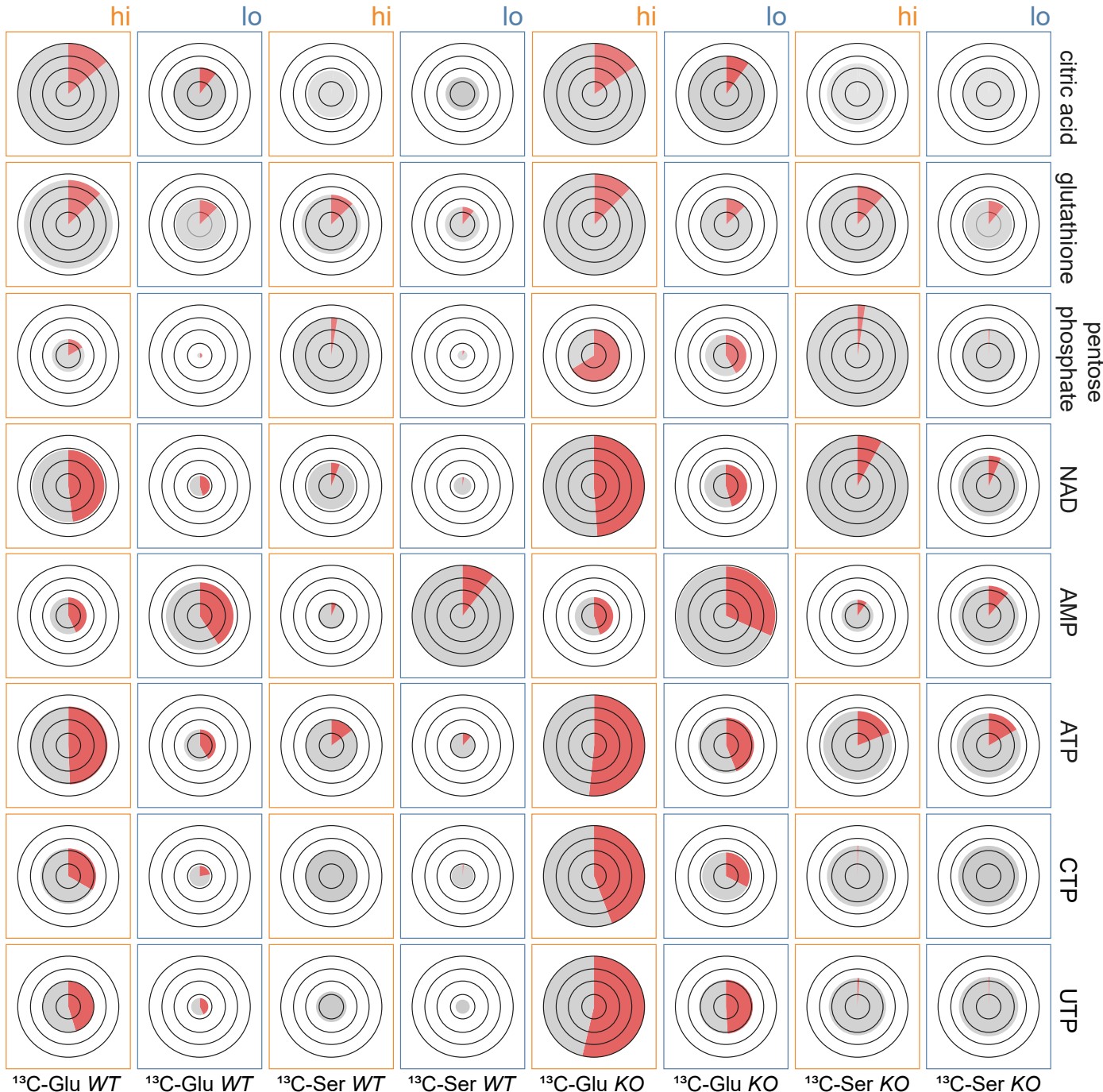

**Extended Data Fig. 10 | The abundance (AU) and fractional contribution (FC) of glucose- or serine-derived ¹³C in metabolites related to one-carbon metabolism are influenced by both mitochondrial inheritance and autophagy status.** Pie Charts were generated using Travis Pies v1.31. Diameter represents metabolite abundance and red fractions represent the contribution of labelled carbon units originating from ¹³C-Glucose (¹³C-Glu) or ¹³C-Serine (¹³C-Ser) in WT (n = 5) and KO (n = 3) CD8⁺ T cells. Experimental layout can be found in Fig. 8.

Anna Katharina Simon

# Reporting Summary

## Statistics

For all statistical analyses, confirm that the following items are present in the figure legend, table legend, main text, or Methods section.

| n/a | Confirmed | |
|---|---|---|
| ☐ | ☒ | The exact sample size (*n*) for each experimental group/condition, given as a discrete number and unit of measurement |
| ☐ | ☒ | A statement on whether measurements were taken from distinct samples or whether the same sample was measured repeatedly |
| ☐ | ☒ | The statistical test(s) used AND whether they are one- or two-sided *Only common tests should be described solely by name; describe more complex techniques in the Methods section.* |
| ☒ | ☐ | A description of all covariates tested |
| ☐ | ☒ | A description of any assumptions or corrections, such as tests of normality and adjustment for multiple comparisons |
| ☐ | ☒ | A full description of the statistical parameters including central tendency (e.g. means) or other basic estimates (e.g. regression coefficient) AND variation (e.g. standard deviation) or associated estimates of uncertainty (e.g. confidence intervals) |
| ☒ | ☐ | For null hypothesis testing, the test statistic (e.g. *F*, *t*, *r*) with confidence intervals, effect sizes, degrees of freedom and *P* value noted *Give P values as exact values whenever suitable.* |
| ☒ | ☐ | For Bayesian analysis, information on the choice of priors and Markov chain Monte Carlo settings |
| ☒ | ☐ | For hierarchical and complex designs, identification of the appropriate level for tests and full reporting of outcomes |
| ☒ | ☐ | Estimates of effect sizes (e.g. Cohen's *d*, Pearson's *r*), indicating how they were calculated |

*Our web collection on statistics for biologists contains articles on many of the points above.*

## Software and code

Policy information about availability of computer code

| Data collection | Data were collected using: BD FACSDiva (Fortessa, LSR or ARIA III hardware), ZenBlue or Olympus Software (ZEISS 980 Airyscan 2, Olympus IX83 inverted microscope),  Seahorse Software. |
|---|---|
| Data analysis | Flow cytometry data were analyzed by Flowjo 10.10.0. Microscopy analysis was performed using ImageJ software. Proteomics data was analyzed using Excel datasheets and visualized in Tableau. scRNAseq was analyzed using R software and visualized in Loupe Software. Mitochondria segmentation imaging analysis was done using scikit-image (Python). Data was exported into Graphpad prism (version 10.4.2) for statistical analysis. |

For manuscripts utilizing custom algorithms or software that are central to the research but not yet described in published literature, software must be made available to editors and reviewers. We strongly encourage code deposition in a community repository (e.g. GitHub). See the Nature Portfolio guidelines for submitting code & software for further information.

## Data

Policy information about availability of data

All manuscripts must include a data availability statement. This statement should provide the following information, where applicable:

- Accession codes, unique identifiers, or web links for publicly available datasets
- A description of any restrictions on data availability
- For clinical datasets or third party data, please ensure that the statement adheres to our policy

The mass spectrometry proteomics data have been deposited to the ProteomeXchage Consortium via the PRIDE partner repository with the dataset identifier PXD053316. Single cell transcriptomics data have been deposited at GEO (GSE270704). Metabolomics data are found in the Source Data File. Additional flow cytometry and imaging data supporting the findings are available from the first/corresponding author upon request.

## Research involving human participants, their data, or biological material

Policy information about studies with human participants or human data. See also policy information about sex, gender (identity/presentation), and sexual orientation and race, ethnicity and racism.

| | |
|---|---|
| Reporting on sex and gender | *Use the terms sex (biological attribute) and gender (shaped by social and cultural circumstances) carefully in order to avoid confusing both terms. Indicate if findings apply to only one sex or gender; describe whether sex and gender were considered in study design; whether sex and/or gender was determined based on self-reporting or assigned and methods used. Provide in the source data disaggregated sex and gender data, where this information has been collected, and if consent has been obtained for sharing of individual-level data; provide overall numbers in this Reporting Summary. Please state if this information has not been collected. Report sex- and gender-based analyses where performed, justify reasons for lack of sex- and gender-based analysis.* |
| Reporting on race, ethnicity, or other socially relevant groupings | *Please specify the socially constructed or socially relevant categorization variable(s) used in your manuscript and explain why they were used. Please note that such variables should not be used as proxies for other socially constructed/relevant variables (for example, race or ethnicity should not be used as a proxy for socioeconomic status). Provide clear definitions of the relevant terms used, how they were provided (by the participants/respondents, the researchers, or third parties), and the method(s) used to classify people into the different categories (e.g. self-report, census or administrative data, social media data, etc.) Please provide details about how you controlled for confounding variables in your analyses.* |
| Population characteristics | *Describe the covariate-relevant population characteristics of the human research participants (e.g. age, genotypic information, past and current diagnosis and treatment categories). If you filled out the behavioural & social sciences study design questions and have nothing to add here, write "See above."* |
| Recruitment | *Describe how participants were recruited. Outline any potential self-selection bias or other biases that may be present and how these are likely to impact results.* |
| Ethics oversight | *Identify the organization(s) that approved the study protocol.* |

Note that full information on the approval of the study protocol must also be provided in the manuscript.

# Field-specific reporting

Please select the one below that is the best fit for your research. If you are not sure, read the appropriate sections before making your selection.

☒ Life sciences         ☐ Behavioural & social sciences         ☐ Ecological, evolutionary & environmental sciences

For a reference copy of the document with all sections, see nature.com/documents/nr-reporting-summary-flat.pdf

# Life sciences study design

All studies must disclose on these points even when the disclosure is negative.

| | |
|---|---|
| Sample size | For microscopy analysis of dividing cells, the totality of mitotic cells found in each sample were imaged/acquired. The numbers varied depending on the abundance of the subsets analyzed, their proliferation potential, and the quality of the preparation prior to imaging. For analysis of mitochondrial geometric parameters, high resolution Z-stack images of >30 cells were analyzed. For functional readouts, 3-5 recipient mice were used per group within each experiment. This number was based on previous experience with adoptive transfers of T cells followed by acute infections (Borsa et al., Science Immunology 2019), the availability of the mice, and feasibility of the experiment. For in vitro assays (e.g. metabolic reliance, survival, proliferation, phenotyping), cells from at least 3 mice were used in independent experiments with technical replicates. Proteomics experiments had cells from at least 6 different mice contributing to the results (minimum of 2 samples, each representing a pool of 2-3 mice). For scRNAseq experiments, cells from 5 mice were used and pooling was done just prior to encapsulation for library preparation. For metabolomics experiments, cells from 3-5 mice were used as independent biological replicates. No statistical test was used to predetermine sample size. |
| Data exclusions | Proteomics data from CD8hi and CD8lo cells had samples excluded following analysis of histone counts. |

| | | |
|---|---|---|
| Replication | In vitro experiments and in vivo readouts represented in main figures were repeated at least 2x. Rare exceptions include screening imaging results or those where alternative methods were used to address the same question. When applicable, this is made clear in figure legends. All replication attempts were successful, but pooling of the data was not always done in the graphs because of differences in cell survival and proliferation potential across experiments or due to variation of fluorescence intensity when Mean Fluorescent Intensity (MFI) was the main readout. | |
| Randomization | Recipient mice were randomly distributed amongst groups at the start of each experiment to avoid age and sex bias. | |
| Blinding | For in vivo experiments it is required by local authorities to state on the cage cards all handling that is done to the mice. Investigators were not blinded during data collection and analysis of freshly isolated tissues or cultured cells. When possible, unbiased sample processing and analysis were performed (proteomics, transcriptomics, metabolomics and part of imaging analysis). | |

# Reporting for specific materials, systems and methods

We require information from authors about some types of materials, experimental systems and methods used in many studies. Here, indicate whether each material, system or method listed is relevant to your study. If you are not sure if a list item applies to your research, read the appropriate section before selecting a response.

## Materials & experimental systems

| n/a | Involved in the study |
|---|---|
| ☐ | ☒ Antibodies |
| ☒ | ☐ Eukaryotic cell lines |
| ☒ | ☐ Palaeontology and archaeology |
| ☐ | ☒ Animals and other organisms |
| ☒ | ☐ Clinical data |
| ☒ | ☐ Dual use research of concern |
| ☒ | ☐ Plants |

## Methods

| n/a | Involved in the study |
|---|---|
| ☒ | ☐ ChIP-seq |
| ☐ | ☒ Flow cytometry |
| ☒ | ☐ MRI-based neuroimaging |

## Antibodies

| | |
|---|---|
| Antibodies used | T cell activation:<br>anti-CD3 145-2C11 (BioLegend, 100340)<br>anti-CD28 37.51 (BioLegend, 102116)<br><br>Microscopy:<br>mouse anti-β-tubulin (Sigma-Aldrich, T8328)<br>mouse anti-Tom20 (F-10) (SantaCruz, sc-17764)<br>rabbit anti-MTHFD2 (EPR26938-20) (Abcam, ab307428)<br>rabbit anti-SHMT2 (E7X5B) (Cell Signaling, #93566)anti-CD8 APC (53-6.7) (BioLegend, 100712)<br>anti-mouse IgG AF488 (Abcam, ab150113)<br>anti-rabbit IgG AF594 (Invitrogen, A-11011)<br>anti-LC3B (D11) XP® Rabbit mAb PE (Cell Signaling, #3868)<br>anti-Tomm20 AF405 (EPR15581-54) (Abcam, ab210047)<br><br>Flow Cytometry and FACS:<br>anti-CD11b N418 PE-Cy7 (Biolegend, 117318)<br>anti-CD127 A7R34 BV711 (Biolegend, 135035)<br>anti-CD127 A7R34 BV785 (Biolegend, 135037)<br>anti-CD127 A7R34 PerCPCy5.5 (eBioscience, 45-1271-82)<br>anti-CD25 PC61 AF700 (Biolegend, 102024)<br>anti-CD25 PC61 PECy7 (Biolegend, 102015)<br>anti-CD25 PC61 APC (Biolegend, 102012)<br>anti-CD44 IM7 AF700 (Biolegend, 103026)<br>anti-CD44 IM7 BV785 (Biolegend, 103041)<br>anti-CD44 IM7 PE (Biolegend, 103008)<br>anti-CD44 IM7 PerCPCy5.5 (Biolegend, 103032)<br>anti-CD45.1 A20 BV785 (Biolegend, 110743)<br>anti-CD45.1 A20  FITC (Biolegend, 110706)<br>anti-CD45.1 A20 PB (Biolegend, 110722)<br>anti-CD45.2 104 AF700 (Biolegend, 109822)<br>anti-CD45.2 104 BV711 (Biolegend, 109847)<br>anti-CD45.2 104 FITC (Biolegend, 109806)<br>anti-CD62L MEL-14 FITC (Biolegend, 104406)<br>anti-CD62L MEL-14 eF450 (eBioscience, 48-0621-82)<br>anti-KLRG1 2F1 BV711 (Biolegend, 138427)<br>anti-KLRG1 BV785 (Biolegend, 138429)<br>anti-CD8 53-6.7 BV510 (Biolegend, 100751)<br>anti-CD8 53-6.7 BV605 (Biolegend, 100743) |

anti-CD8 53-6.7 FITC (Biolegend, 100706)
anti-CD8 53-6.7 PE (Biolegend, 100708)
anti-TCRb H57-597 APC-Cy7 (Biolegend, 109220)
anti-TCRb H57-597 PerCPCy5.5 (Biolegend, 109228)
anti-IL2 JES6-5H4 APC (Biolegend, 503810)
anti-IL2 JES6-5H4 PE (Biolegend, 503808)
anti-IFNg XMG1.2 BV421 (Biolegend, 505829)
anti-TNF MP6-XT22 PE-Cy7 (ThermoFischer, 25-7321-82)
anti-Granzyme B QA16A02 APC (Biolegend, 372204)

Western Blot:
rabbit anti-ATG16L1 (EPR15638) (Abcam, ab187671)
mouse anti-GAPDH (6C5) (Sigma-Aldrich, MAB374).
IRDye 680LT Goat anti-Mouse IgG (H+L) (Licor, 926-680-70)
IRDye 800CW Goat anti-Rabbit IgG (H+L) (Licor, 926-322-11).

Identification of viable cells was done by fixable dead cell staining (Life Technologies, L34993 or L34957).
MitoSOX™ Mitochondrial Superoxide Indicator (Invitrogen) was used to detect mitochondrial superoxide
SnapTag labelling was done using cell permeable Snap-Cell substrates (New England Biolabs, NEB) in the following concentrations: 3 μM (Snap-Cell 647-SiR S9102S), 3 μM (Snap-Cell TMR-Star S9105S), 5 μM (Snap-Cell Oregon Green S9104S). SnapTag blocking was performed using 5 μM of unlabelled SnapSubstrate (Snap-Cell Block S9106S).

**Validation**

Antibodies were validated by using known negative and positive cell populations as controls. Some antibodies were previously validated by the manufacturer and information was extracted from their website.

# Animals and other research organisms

Policy information about studies involving animals; ARRIVE guidelines recommended for reporting animal research, and Sex and Gender in Research

**Laboratory animals**

CD45.2 Omp25-SnapTagfl/fl mice were bred with CD45.1 Atg16l1fl/fl Ert2Cre OT-I mice expressing a TCR specific for OVA257–264 SIINFEKL peptide, and maintained as CD45.1 or CD45.1/2 mice in a C57BL/6 background. Host mice in adoptive transfer experiments were either B6.SJL.CD45.1 or C57BL/6 naïve mice. Six-to-sixteen-week-old mice were considered young and >100 week-old mice were considered aged.

**Wild animals**

The study did not involve wild animals.

**Reporting on sex**

For this study we used both male and female mice and did not observe any sex bias in our results.

**Field-collected samples**

The study did not involve samples collected from the field.

**Ethics oversight**

All animal work was reviewed and approved by Oxford Ethical committee and the UK Home office under the project licenses PPL30/3388 and P01275425.

Note that full information on the approval of the study protocol must also be provided in the manuscript.

# Plants

**Seed stocks**

*Report on the source of all seed stocks or other plant material used. If applicable, state the seed stock centre and catalogue number. If plant specimens were collected from the field, describe the collection location, date and sampling procedures.*

**Novel plant genotypes**

*Describe the methods by which all novel plant genotypes were produced. This includes those generated by transgenic approaches, gene editing, chemical/radiation-based mutagenesis and hybridization. For transgenic lines, describe the transformation method, the number of independent lines analyzed and the generation upon which experiments were performed. For gene-edited lines, describe the editor used, the endogenous sequence targeted for editing, the targeting guide RNA sequence (if applicable) and how the editor was applied.*

**Authentication**

*Describe any authentication procedures for each seed stock used or novel genotype generated. Describe any experiments used to assess the effect of a mutation and, where applicable, how potential secondary effects (e.g. second site T-DNA insertions, mosiacism, off-target gene editing) were examined.*

# Flow Cytometry

## Plots

Confirm that:

☒ The axis labels state the marker and fluorochrome used (e.g. CD4-FITC).

☒ The axis scales are clearly visible. Include numbers along axes only for bottom left plot of group (a 'group' is an analysis of identical markers).

☐ All plots are contour plots with outliers or pseudocolor plots.

☒ A numerical value for number of cells or percentage (with statistics) is provided.

## Methodology

| | |
|---|---|
| Sample preparation | Blood samples used for kinetics analysis were obtained from the tail vein. Spleens and inguinal lymph nodes were harvested without perfusion. Single cell splenocytes were prepared by meshing whole spleens through 70 μm strainers using a syringe plunger. Lymph node single cell suspensions were prepared by meshing organs through 40 μm strainers using a syringe plunger. When cytokine production was assessed, CD8+ T cells within splenocytes were stimulated with 1 μg/ml of SIINFEKL peptide in the presence of Brefeldin A (Sigma-Aldrich) for 4-6 hours at 37°C. Identification of viable cells was done by fixable near-IR dead cell staining (Life Technologies). Erythrocytes were lysed by red blood cell lysis buffer treatment 5 min at room temperature. Surface stainings were performed for 20-30 min at 4°C. Specific CD8+ T cells were evaluated by incubation with SIINFEKL257-264-APC-Labeled or SIINFEKL257-264-BV421-Labeled tetramers (NIH Tetramer Core Facility at Emory University). All samples were washed and stored in PBS containing 2% FBS and 5mM of EDTA before acquisition. |
| Instrument | LSR II and Fortessa X20 (BD) |
| Software | BD FACSDiva (acquisition); BD Flowjo 10.10.0 (analysis) |
| Cell population abundance | For adoptive cell transfer experiments and in vitro assays, CD8+ T cells were isolated from spleens and inguinal lymph nodes using the EasySep Mouse CD8+ T cell Isolation Kit (naive or total population, StemCell), following manufacturer's instructions (purity >90%). We obtained 3-10x10^6 cells per mouse (3-10% of total splenic cells). For ex vivo analysis of splenic T cells from adoptive transfer experiments, 10% to 20% of the organ was used per staining. Frequencies of transferred cells within CD8 T cell populations varied across experiments. |
| Gating strategy | For all samples the following gating strategy was used: lymphocytes (SSC-A/FSC-A), exclusion of doublets (FSC-A/FSC-H), live cells (FSC-A/Live/Dead marker). To identify congenitally marked TCR transgenic CD8+ T cells, cells were gated for CD8 and for the congenic marker CD45.1 or CD45.2. First-daughter cells were identified by dilution of Cell Trace Violet. |

☒ Tick this box to confirm that a figure exemplifying the gating strategy is provided in the Supplementary Information.

