## [Peer Review File · Nature Cell Biology]

Autophagy-regulated mitochondrial inheritance controls early CD8⁺ T cell fate commitment

Corresponding Author: Professor Mariana Borsa

Version 0:

Decision Letter:

*Please delete the link to your author homepage if you wish to forward this email to co-authors.

Dear Dr Borsa,

Thank you again for submitting your manuscript, "Inheritance of old mitochondria controls early CD8+ T cell fate commitment and is regulated by autophagy", to Nature Cell Biology and I apologize for the delay in sharing our decision with you. Your manuscript has now been seen by 3 referees, who are experts in immunometabolism, autophagy (Referee #1); T cell development (Referee #2); and immunology; T cells; metabolism (Referee #3). As you will see from their comments (attached below), they found the work of potential interest but have raised substantial concerns, which in our view would need to be addressed with considerable revisions before we can consider publication in Nature Cell Biology.

Nature Cell Biology editors discuss the referee reports in detail within the editorial team, including the chief editor, to identify key referee points that should be addressed with priority, and requests that are overruled as being beyond the scope of the current study. To guide the scope of the revisions, I have listed these points below. Our standard revision period is six months, and we are committed to providing a fair and constructive peer-review process, so please feel free to contact me if you would like to discuss any of the referee comments further.

In particular, it would be essential to dedicate efforts in revision to address the following points:

1-- The reviewers were not yet convincing by the functional impact of the differential inheritance of old and young mitochondria in T cells, and they recommended further work to strengthen the comparative analyses:

Rev#1 points #2, #3, #4

Rev#2 paragraphs "The use of Omp25-SnapTag.."; "The in vitro experiments in Fig 3 .."; "Morphological characterisation of the mitochondria.."; "Fig 5 is a nice analysis of proteomic .."; "Fig 6 uses labelling of both old..."; "Overall, this is a very nice study.."

Rev#3 point #1

2-- Claims related to glycolysis and one-carbon metabolism should be strengthened as suggested by Rev#1 in point #5.

3-- All other referee concerns pertaining to strengthening existing data, providing controls, methodological details, clarifications and textual changes should also be addressed.

4-- Finally, please pay close attention to our guidelines on statistical and methodological reporting (listed below) as failure to do so may delay the reconsideration of the revised manuscript. In particular, please provide:

We would be happy to consider a revised manuscript that would satisfactorily address these points, unless a similar paper is published elsewhere or is accepted for publication in Nature Cell Biology in the meantime.

- ensure that it conforms to our format instructions and publication policies (see below and <https://www.nature.com/nature/for-authors>).
- provide a point-by-point rebuttal to the full referee reports verbatim, as provided at the end of this letter.
- provide the completed Reporting Summary (found here <https://www.nature.com/documents/nr-reporting-summary.pdf>). This is essential for reconsideration of the manuscript will be available to editors and referees in the event of peer review. For more information see <http://www.nature.com/authors/policies/availability.html> or contact me.

Nature Cell Biology is committed to improving transparency in authorship. As part of our efforts in this direction, we are now requesting that all authors identified as 'corresponding author' on published papers create and link their Open Researcher and Contributor Identifier (ORCID) with their account on the Manuscript Tracking System (MTS), prior to acceptance. ORCID helps the scientific community achieve unambiguous attribution of all scholarly contributions. You can create and link your ORCID from the home page of the MTS by clicking on 'Modify my Springer Nature account'. For more information please visit www.springernature.com/orcid.

This journal strongly supports public availability of data. Please place the data used in your paper into a public data repository, or alternatively, present the data as Supplementary Information. If data can only be shared on request, please explain why in your Data Availability Statement, and also in the correspondence with your editor. Please note that for some data types, deposition in a public repository is mandatory - more information on our data deposition policies and available repositories appears below.

Link Redacted

We hope that you will find our referees' comments and editorial guidance helpful. Please do not hesitate to contact me if there is anything you would like to discuss. Thank you again for considering NCB for your work.

Best wishes,

Melina

Melina Casadio, PhD
Senior Editor, Nature Cell Biology
ORCID ID: <https://orcid.org/0000-0003-2389-2243>

Reviewers' Comments:

Reviewer #1:

Remarks to the Author:

The manuscript by Borsa et al., is quite exciting and novel. This paper attempts to mechanistically link inheritance of cell cargoes with fate decisions. Through a series of elegant experiments, the authors show that autophagy regulates the inheritance of mitochondria. These mitochondria have distinct features, notably termed young and old (through SnapTag technology). Moreover, daughter cells possessing old mitochondria are more glycolytic and have lower capacity to form

memory.

Overall, the genetic models and well-executed experiments is compelling. There are several key points I would like to see addressed as crucial to validate the key findings in this paper:

1. Regarding the following comment and the assay to assess old versus young mitos

“SnapTag is a modified DNA repair enzyme that can covalently bind to different cell-permeable substrates linked to fluorophores. Sequential labelling of cells expressing OMP-25-SnapTag+ mitochondria allows the discrimination between old (pre-mitotic) and young (post-mitotic) organelles and the separation of different cell sub-populations based on patterns of organelle inheritance.”

2. The SnapTag system requires additional validation. The stability of OMP-25 may impact the labelling and consequently the demarcation of mitosnap high versus low. The authors should consider controls using SnapTag to a high turnover versus low turnover mitochondrial protein

3. Tagging a post-mitotic Tcm cells that have low mitotic turnover would be inconsistent with labelling Tcm cells as possessing young mitochondria. This could be addressed by evaluating the scRNA-seq data (or flow staining) for conventional Tcm markers in MitoSnap high and low cells. In other words, if autophagy deficient cells retain the mitoSnap high feature, there should be a concomitant loss in Tcm markers of these same cells.

4. It is not entirely clear whether autophagy KO cells lose MitoSnap low feature after in vivo challenge which is the critical experiment to demonstrate the mechanistic connection to memory formation.

5. With respect to the scRNA-seq and proteomic dataset, the experiments to demonstrate the importance of cluster 2 on MitoSnap labelling is strong. However, additional metabolite assessments need to be conducted in autophagy defective vs WT cells (mitoSnap high versus low), to formally show that 1C metabolites are altered and there is indeed a shift to glycolysis and 1C metabolism away from OXPHOS. Ideally, the authors should consider a 13C-labelling experiment. In other words, does the mitoSnap labelling segregate with the metabolic features proposed by the authors?

Reviewer #2:

Remarks to the Author:

Borsa et al describe a comprehensive and interesting series of experiments that shed new light on how mitochondria are regulated by asymmetric cell division in CD8+ T cells, and how this influences the memory versus effector fate decisions. The work is carefully performed, novel and compelling; and appropriated for Nature Cell Biology. However, I have concerns regarding interpretation of the young versus new mitochondria, as detailed below.

The authors first compared high and low CD8 daughter cells for differences in proteome between effector-like and memory-like precursors after the first division. This approach complements previous transcriptomics efforts, but is not substantially explored or discussed (eg how do the proteomic data compare to similar transcriptomic efforts?). The experiments are only performed using anti-CD3, anti-CD28 and Fc-208 ICAM-1 coated plates, and ideally would be assessed on MHC-peptide presentation (perhaps using the OT1 mice to tie in with the in vivo experiments) to confirm physiological relevance. At a minimum the artificial nature of this activation should be discussed. This notwithstanding, the proteomics approach and comparison with different experimental conditions throughout the manuscript provides a nice complement to the single cell studies.

The finding that MitoSOX (superoxide-producing mitochondria) is asymmetric in the first daughters is solid, novel and interesting, as is the elegant approach to ablating autophagy after TCR activation, which prevents this asymmetry. Together, the experiments in Fig 1 provide compelling evidence for autophagy-dependent asymmetry in mitochondrial function.

The use of Omp25-SnapTag mice provides an elegant approach to track aging of mitochondria, and compelling data that old mitochondria are asymmetric at division. Fig. 2E clearly shows more old mitochondria in the knockout cells, but I'm not sure I agree with the interpretation of 'only lost the old mitochondria in 3.6% of all dividing cells, as opposed to 23% in WT conditions'. It isn't clear which (if any) of the cells are undivided, but all CTV populations seem higher for 'old mito' in the knockout, and there is not an obvious inflection point to motivate binarizing the data. Perhaps presenting the data as histograms binned by CTV halving would be clearer, and a more appropriate conclusion that old mitochondria were increased at each division compared with the WT. And surely the same analysis was performed on young mitochondria? Did they show a similar loss over time, and was this also dependent on autophagy? 2F is the more compelling figure demonstrating that autophagy is required for asymmetry of old mitochondria at division. I agree that Fig 2H shows autophagy is required for the loss of old mitochondria, but doesn't this also show autophagy is required for the loss of new mitochondria? Flow cytometric characterisation of the mitochondria themselves adds interesting information, but it should be pointed out that the effect in Figure S2C is subtle, and that qualitative difference (fewer smaller mitochondria) would not be sufficient to see the difference in total mitochondrial content, leaving open the possibility that all mitochondrial sizes are reduced more in wt). Figure 2K is difficult to interpret – few (and similar proportions in the two conditions) show only old mitochondria, similar proportions show both old and new, and the major difference highlighted in the manuscript is the 16.4 % of very bright for old mitochondria (in the KO, 1.71 in WT). This discrete population was not evident in Fig 2I – what is the

difference here? And how do the authors explain that the bulk of the old mitochondria seems untouched by the KO vs WT comparison. I suspect I am missing something here, a schematic describing each experiment would help. Overall, I find the asymmetry of mitochondria and the requirement for autophagy compelling, and the discrimination between old and young mitochondria less so. This becomes an issue in later parts of the manuscript, where claims such as 'MitoSnap^{hi} CD8⁺ T cells inheriting old mitochondria exhibited higher global translation rates than MitoSnap^{lo} cells' (line 353) – would be more accurate if 'inheriting old mitochondria' were deleted. (similar concerns throughout the manuscript eg line 369).

The in vitro experiments in Fig 3 are a nice demonstration of functional differences between MitoSnap^{hi} and ^{lo} cells, with MitoSnap^{lo} cells having better memory potential. A schematic of the timing (eg Fig 2A) and data showing the initial flow cytometry sort would be helpful here to give a feel for the spread in mitochondrial content and the changes with division. And it is not stated, but were young mitochondria also labelled?

Morphological characterisation of the mitochondria in Fig 4 is interesting, carefully performed, and compatible with mitochondrial fusion. If I understand correctly, the initial sort for hi vs lo was made somewhat redundant by the post sort binning of images into hi, medium and low. Is there any difference in intensity between the low sort and the low intensity cells in the high sort? Perhaps to avoid confusion, the difference between these two could be in supplemental data, leaving the more interesting comparison of the three bins from the high sort in the main figure. It was not clear to me how long after the sort the cells were imaged. Importantly, this characterisation was performed on aged mitochondria, but presumably the same image analysis could be performed on young (green) mitochondria. This would be interesting to show. The functional characterisation of cells with high and low mitochondria in the remaining panels of Fig 4 are nicely performed, and appropriately interpreted.

Fig 5 is a nice analysis of proteomic and transcriptomic differences between cells with high and low mitochondria. Again, the distribution of CTV and Snap tag at the time of sorting should be shown, and the cutoffs for the sort. The implication as worded is that this cutoff discriminates one daughter from the other, but if the hi population is primarily a small distinct very bright population as per Fig 2K, it is not clear how this relates to the more subtle asymmetry likely to reflect ACD.

Fig 6 uses labelling of both old and new mitochondria, but as far as I can tell, does not then use the two colours to discriminate between old and new, which leaves me to assume that they were coregulated? It was the cutoff for hi and low discrimination based on green, red or both? As written the inferences about old and new mitochondria in this section do not seem to be based on the labelling, and it is unclear what the labelling of young mitochondria achieved here.

The discussion begins with an appropriate description of the lack of a direct link between molecular asymmetry at division and functional differences for T cells. However, I don't agree that this paper shows 'asymmetric inheritance of an unequivocal pre-mitotic cell cargo causes divergent T cell fate commitment' (line 561). This would only be the case if the mother mitochondria were inherited by one daughter but not the other, but here the interpretation is confounded by both the small differences in mitochondrial inheritance, and the amplification of these differences by differential autophagy. Unfortunately, this study does not provide that sought-after link. However, this does not detract from the value of this study, just does not merit such a claim.

Overall, this is a very nice study, carefully performed and with several novel and important findings. My major concern, raised above but exemplified in line 381 'even in MitoSnap^{hi} cells that inherited their mitochondria from the mother. This implies an all-or-nothing aspect of the mitochondria asymmetry, which is not compatible with the mean value <0.3 in Fig 1E – slightly more mitochondria inherited by one daughter than the other but certainly leaving plenty in the other daughter. This subtle difference, combined with the observation that both red and green mitochondria were lost over time (Fig 2I), and that new mitochondria were not explored in parallel in many of the assays, is not compatible with the framing that old mitochondria behave substantially different to new mitochondria in their regulation and impact. The manuscript would lose very little by reframing away from the old vs new phenomena to a refocus on the mitochondria as a nexus between asymmetric autophagy and subsequent metabolism and function in the daughter cells.

Sarah Russell

Reviewer #3:

Remarks to the Author:

During initial T cell activation, asymmetric cell division has been linked to daughter cell fate, specifically as to whether a daughter cell adopts a short-lived effector cell phenotype or ultimately enters the memory cell pool. Previous studies have linked asymmetric division of metabolic signalling proteins/pathways (eg mTOR) and key transcriptional factors (eg Myc) as being key to this process. In the current study, Borsa and colleagues provide evidence that asymmetric inheritance of "old" mitochondria biases towards short-lived effector cell fate. This is linked to and dependent upon autophagy/mitophagy and is associated with enhancement of one-carbon metabolism and glycolysis. To my knowledge, these findings are very novel whilst the data is convincing. The experimental approaches are cutting edge, including novel mouse models (Mitosnap mice) whilst the manuscript is clearly written and data presented well.

Specific comments:

1. The adoptive cell transfer experiments (Fig 3) show that MitoSnap^{low} T cells have increased persistence after Listeria clearance and better recall responses, consistent with the view that inheritance of "old" mitochondria biases against memory formation. It would be informative to compare the responses of MitoSnap^{low} and high cells during the initial phase of

activation in vivo - presumably the hypothesis is that the Mitosnap high cells are initially more-effector like but do not persist? - minor comment. Fig 3 legend states that 5×10^3 cells were transferred whilst methods say $5-50 \times 10^3$ were transferred. Was the same result achieved when the higher cell number was used? - this is not reported.

2. As discussed in the manuscript, previous studies linked asymmetric inheritance/expression of Myc to T cell fate. Did the authors find evidence for differential Myc expression in the initial proteomics experiments (sorted CD8hi vs CD8lo, Fig 1) or proteomic/transcriptomic analysis of Mitosnap high vs low cells (Fig 5)?

Minor point. Fig 1F microscopy images right panel are mislabelled as "Atg16l1 KO" - should it not be "old"?

Methods should be written concisely, but should contain all elements necessary to allow interpretation and replication of the results. As a guideline, Methods sections typically do not exceed 3,000 words. The Methods should be divided into subsections listing reagents and techniques. When citing previous methods, accurate references should be provided and any alterations should be noted. Information must be provided about: antibody dilutions, company names, catalogue numbers and clone numbers for monoclonal antibodies; sequences of RNAi and cDNA probes/primers or company names and catalogue numbers if reagents are commercial; cell line names, sources and information on cell line identity and authentication. Animal studies and experiments involving human subjects must be reported in detail, identifying the committees approving the protocols. For studies involving human subjects/samples, a statement must be included confirming that informed consent was obtained. Statistical analyses and information on the reproducibility of experimental results should be provided in a section titled "Statistics and Reproducibility".

All Nature Cell Biology manuscripts submitted on or after March 21 2016 must include a Data availability statement as a separate section after Methods but before references, under the heading "Data Availability". For Springer Nature policies on data availability see <http://www.nature.com/authors/policies/availability.html>; for more information on this particular policy see <http://www.nature.com/authors/policies/data/data-availability-statements-data-citations.pdf>. The Data availability statement should include:

- Accession codes for primary datasets (generated during the study under consideration and designated as "primary accessions") and secondary datasets (published datasets reanalysed during the study under consideration, designated as "referenced accessions"). For primary accessions data should be made public to coincide with publication of the manuscript. A list of data types for which submission to community-endorsed public repositories is mandated (including sequence, structure, microarray, deep sequencing data) can be found here <http://www.nature.com/authors/policies/availability.html#data>.
- Unique identifiers (accession codes, DOIs or other unique persistent identifier) and hyperlinks for datasets deposited in an approved repository, but for which data deposition is not mandated (see here for details <http://www.nature.com/sdata/data-policies/repositories>).
- At a minimum, please include a statement confirming that all relevant data are available from the authors, and/or are included with the manuscript (e.g. as source data or supplementary information), listing which data are included (e.g. by figure panels and data types) and mentioning any restrictions on availability.
- If a dataset has a Digital Object Identifier (DOI) as its unique identifier, we strongly encourage including this in the Reference list and citing the dataset in the Methods.

We recommend that you upload the step-by-step protocols used in this manuscript to [protocols.io](http://www.protocols.io). More details can be found at <https://www.protocols.io/help/publish-articles>.

All imaging data should be accompanied by scale bars, which should be defined in the legend.

Cropped images of gels/blots are acceptable, but need to be accompanied by size markers, and to retain visible background signal within the linear range (i.e. should not be saturated). The boundaries of panels with low background have to be demarked with black lines. Splicing of panels should only be considered if unavoidable, and must be clearly marked on the figure, and noted in the legend with a statement on whether the samples were obtained and processed simultaneously. Quantitative comparisons between samples on different gels/blots are discouraged; if this is unavoidable, it should only be performed for samples derived from the same experiment with gels/blots were processed in parallel, which needs to be stated in the legend.

The total number of Supplementary Figures (not including the "unprocessed scans" Supplementary Figure) should not exceed the number of main display items (figures and/or tables (see our Guide to Authors and March 2012 editorial <http://www.nature.com/ncb/authors/submit/index.html#suppinfo>; <http://www.nature.com/ncb/journal/v14/n3/index.html#ed>). No restrictions apply to Supplementary Tables or Videos, but we advise authors to be selective in including supplemental data.

GUIDELINES FOR EXPERIMENTAL AND STATISTICAL REPORTING

REPORTING REQUIREMENTS – We are trying to improve the quality of methods and statistics reporting in our papers. To that end, we are now asking authors to complete a reporting summary that collects information on experimental design and reagents. The Reporting Summary can be found here <https://www.nature.com/documents/nr-reporting-summary.pdf>. If you would like to reference the guidance

text as you complete the template, please access these flattened versions at http://www.nature.com/authors/policies/availability.html.

Version 1:

Decision Letter:

Our ref: NCB-A54956A

26th June 2025

Dear Dr. Borsa,

Thank you for submitting your revised manuscript "Autophagy-regulated mitochondrial inheritance controls early CD8⁺ T cell fate commitment" (NCB-A54956A). It has now been seen by the original referees and their comments are below. The reviewers find that the paper has improved in revision, and therefore we'll be happy in principle to publish it in Nature Cell Biology, pending minor revisions to satisfy the referees' final requests and to comply with our editorial and formatting guidelines.

Thank you again for your interest in Nature Cell Biology Please do not hesitate to contact me if you have any questions.

Sincerely,

Angela R Parrish, PhD
Locum Senior Editor
Nature Cell Biology

Reviewer #1 (Remarks to the Author):

The authors have adequately responded to my original queries. For (P3) 4, I do not disagree with the inherent challenge of these experiments. However, it is central to the mechanistic connection to memory formation. I would appreciate some commentary in the discussion on this point.

Reviewer #2 (Remarks to the Author):

Congratulations to the authors on such a comprehensive revision of what was already a very interesting paper. I am pleased

that all concerns have been alleviated, and I feel the paper is highly suitable for NCB.

Sarah Russell

Reviewer #3 (Remarks to the Author):

The authors have addressed my questions in full. Congratulations to them on their interesting study.

Version 2:

Decision Letter:

Dear Dr. Borsa,

I am writing on behalf of my colleague, Dr. Angela Parrish, who is out of the office.

I am pleased to inform you that your manuscript, "Autophagy-regulated mitochondrial inheritance controls early CD8⁺ T cell fate commitment", has now been accepted for publication in *Nature Cell Biology*.

Over the next few weeks, your paper will be copyedited to ensure that it conforms to *Nature Cell Biology* style. Once your paper is typeset, you will receive an email with a link to choose the appropriate publishing options for your paper and our Author Services team will be in touch regarding any additional information that may be required.

Publication is conditional on the manuscript not being published elsewhere and on there being no announcement of this work to any media outlet until the online publication date in *Nature Cell Biology*.

Please note that *Nature Cell Biology* is a Transformative Journal (TJ). Authors may publish their research with us through the traditional subscription access route or make their paper immediately open access through payment of an article-processing charge (APC). Authors will not be required to make a final decision about access to their article until it has been accepted. <https://www.springernature.com/gp/open-research/transformative-journals> Find out more about Transformative Journals

Authors may need to take specific actions to achieve compliance with funder and institutional open access mandates. If your research is supported by a funder that requires immediate open access (e.g. according to a

<https://www.springernature.com/gp/open-science/plan-s-compliance>> Plan S principles or the https://www.springernature.com/gp/open-science/us-federal-agency-compliance NIH public access policy) then you should select the gold OA route, and we will direct you to the compliant route where possible. Because authors warrant under our subscription licensing terms that they haven't committed to licensing any version of their article under a licence inconsistent with the terms of our agreement – including the applicable embargo period – publication under the subscription model isn't suitable for authors whose funders require no embargo.

If you have not already done so, we strongly recommend that you upload the step-by-step protocols used in this manuscript to protocols.io (<https://protocols.io>), an open online resource that allows researchers to share their detailed experimental know-how. All uploaded protocols are made freely available and are assigned DOIs for ease of citation. Protocols and Nature Portfolio journal papers in which they are used can be linked to one another, and this link is clearly and prominently visible in the online versions of both. Authors who performed the specific experiments can act as primary authors for the Protocol as they will be best placed to share the methodology details, but the Corresponding Author of the present research paper should be included as one of the authors. By uploading your Protocols onto protocols.io, you are enabling researchers to more readily reproduce or adapt the methodology you use, as well as increasing the visibility of your protocols and papers. You can also establish a dedicated workspace to collect your lab Protocols. Further information can be found at <https://www.protocols.io/help/publish-articles>.

Nature Cell Biology encourages authors presenting evidence for cell, biological, molecular, and genetic interactions to consider communicating these findings using Biofactoid (<https://biofactoid.org/>). This tool helps users share a searchable representation of interactions (e.g. binding, gene expression, post-translational modification) between genes, gene products, or chemicals. Information added to Biofactoid, with author attribution, is shared on social media and public databases, such as Pathway Commons, where it can be discovered and analyzed in the context of a large and growing corpus of knowledge.

All the best,

Christina

Christina Kary, PhD
Chief Editor
Nature Cell Biology
1 New York Plaza

** Visit the Springer Nature Editorial and Publishing website at http://editorial-jobs.springernature.com?utm_source=ejp_NCB_email&utm_medium=ejp_NCB_email&utm_campaign=ejp_NCB for more information about our career opportunities. If you have any questions please click here.

Dear colleagues,

We sincerely appreciate your constructive feedback. Your insightful comments and suggestions have helped us improve the manuscript and strengthen its conclusions. Below, you will find our detailed responses to each of your points. We realized we were not clear when writing about some of our findings, which we have now amended. Regarding the points that required additional experiments, in spite of a few biological and structural/logistics challenges that delayed our progress (e.g. it took a very long time to identify a suitable metabolomics facility and to obtain the results), we believe we have successfully addressed all concerns. We are therefore hopeful that the revised manuscript now meets your standards in terms of novelty and scientific rigor. To facilitate your assessment, we have numbered the highlighted revisions and annotated your corresponding points as Pn in the right margins to facilitate their identification in the manuscript file.

Kind regards,
Mariana

Reviewers' Comments:

Reviewer #1:

Remarks to the Author:

The manuscript by Borsa et al., is quite exciting and novel. This paper attempts to mechanistically link inheritance of cell cargoes with fate decisions. Through a series of elegant experiments, the authors show that autophagy regulates the inheritance of mitochondria. These mitochondria have distinct features, notably termed young and old (through SnapTag technology). Moreover, daughter cells possessing old mitochondria are more glycolytic and have lower capacity to form memory.

Overall, the genetic models and well-executed experiments is compelling. There are several key points I would like to see addressed as crucial to validate the key findings in this paper:

1. Regarding the following comment and the assay to assess old versus young mitos

“SnapTag is a modified DNA repair enzyme that can covalently bind to different cell-permeable substrates linked to fluorophores. Sequential labelling of cells expressing OMP-25-SnapTag+ mitochondria allows the discrimination between old (pre-mitotic) and young (post-mitotic) organelles and the separation of different cell sub-populations based on patterns of organelle inheritance.”

(P1) *2. The SnapTag system requires additional validation. The stability of OMP-25 may impact the labelling and consequently the demarcation of mitosnap high versus low. The authors should consider controls using SnapTag to a high turnover versus low turnover mitochondrial protein.*

This is a fair point. OMP-25-SnapTag is being used here as a tool to visualize mitochondria and assess its degradation/segregation and new biogenesis. The turnover of OMP25 could impact the reliability of our readouts if not preserved for the timeframe of SnapTag inheritance assessment (1st mitosis). However, we believe that the maintenance of OMP-25-SnapTag labelling over several days is enough to address the questions answered in this study, as all our functional readouts are based on the comparison of first-daughter MitoSnap^{hi} and MitoSnap^{lo} cells. Furthermore, because most cells preserve **pre-mitotic (preM)** old OMP-25 labelling at the timepoint of first mitosis (new Fig. 2D) and autophagy-deficient cells keep it for several days (new Fig. 3B), one can conclude that labelling within the time frame we are measuring is subject to loss through autophagy-dependent degradation or segregation of mitochondria. Importantly, asymmetric segregation happens in a short timeframe during mitosis and is not influenced by prior turnover of OMP-25 protein and the remaining volume of MitoSnap labelled organelles. Thus, these data allow us to conclude that OMP-25-SnapTag turnover correlates with mitochondrial turnover.

To further address the reviewer's point and confirm that loss of OMP-25-SnapTag is linked to autophagy/mitophagy and not lack of stability, we have performed a similar experiment to the one shown in new Fig. 3C (former Fig. 2J), where we evaluated the loss of young-OMP-25-SnapTag in presence or not of the autophagy inhibitor BafA. In this new experiment, we sorted MitoSnap^{hi} and MitoSnap^{lo} daughter cells to be certain about whether loss of OMP-25-SnapTag labelling could be linked to mitophagy in both subsets. As one can see below, both biogenesis of young mitochondria (Figure R1A) and induction of mitophagy-induced (CCCP) loss of young-OMP-25-SnapTag happen in both subsets (Figure R1B). These results confirm the contribution of mitochondrial degradation in the rise of MitoSnap^{lo} cells and are now part of new Figure 3 D-G.

Figure R1: Mitophagy results in loss of OMP-25-Snap staining. (A) Experimental layout: CTV labelled naïve MitoSnap CD8⁺ T cells (WT-*Atg16l1*^{fl/fl} *Omp25*^{fl/fl} *Ert2*^{Cre}) were activated on anti-CD3, anti-CD28 and Fc-ICAM-1 coated plates for 40 h. Cells were cultured in T cell medium containing 500 nM Z-4-Hydroxytamoxifen (4OHT). 16 h post-activation, cells were harvested and labelled with Snap-Cell 647-SiR to tag pre-mitotic (preM)/old mitochondria and cultured for a further 24 h, when Snap-Cell Block and Snap-Cell Oregon Green incubations allowed young organelle labelling. Efficiency of young labelling to confirm recent mitochondrial biogenesis was performed (right panel). 1st daughter-cells were sorted (as in main Figure 2D) and put back in culture for 2 h in presence or not of CCCP. (B) Representative histograms exhibiting young MitoSnap labelling in MitoSnap^{hi} and MitoSnap^{lo} cells subjected or not mitophagy induction by CCCP (left panel). Cell frequencies of cells keeping or not young-OMP-25 labelling after 2h were quantified (right panel). Statistical analysis was performed using 2-way ANOVA and Tukey's multiple comparisons test.

We do not disagree that a second SnapTag model using a different mitochondrial protein would be of potential interest, however, we would like to highlight that we spent few years optimizing the transgenic OMP-25-SnapTag mouse model to be used in the context of T cells. Considering this was the case even though the lab of Prof. Pekka Katajisto had validated this model and target protein of choice in another cell type, we believe it would go beyond the scope of this manuscript to generate a new mouse model. We hope the reviewer can understand our position.

(P2) 3. Tagging a post-mitotic Tcm cells that have low mitotic turnover would be inconsistent with labelling Tcm cells as possessing young mitochondria. This could be addressed by evaluating the scRNA-seq data (or flow staining) for conventional Tcm markers in MitoSnap high and low cells. In other words, if autophagy deficient cells retain the MitoSnap high feature, there should be a concomitant loss in Tcm markers of these same cells.

Concerning the first comment in this point, our interpretation is that the reviewer refers to MitoSnap^{lo} cells as Tcm (to-be) cells. As already addressed in **P1**, our results suggest that post-mitotic "memory-

like" daughter cells (MitoSnap^{lo}) are the ones endowed with higher mitochondrial turnover rates, as loss of OMP-25-SnapTag labelling relies on autophagy. Because virtually all cells are tagged upon young OMP-25-SnapTag labelling (Figure R1A above), which occurs after 1st mitosis but prior to a second division, the loss of young mitochondrial structures within few hours (time between labelling and readouts) in cells that had previously cleared pre-mitotic old OMP-25-SnapTag (by degradation or segregation, Fig. 2D in the manuscript) can only be interpreted as the **result of degradation**. Although we are the first to show that fate divergency can be influenced by mitochondrial inheritance and different mitochondrial turnover rates as early as the first mitosis following CD8⁺ T cell activation, the role of autophagy in maintenance/formation of memory CD8⁺ T cells has been previously investigated (Sinclair et al., 2025, Nature Immunology; Franco et al., 2023, Science Immunology; Puleston et al., 2014, eLife; Xu et al.; 2014, Nature Immunology).

Concerning the raised point about evaluating whether MitoSnap^{lo} cells exhibit Tcm features, this has already been addressed by us in original figure S5A. We have now moved it to main Figure 6F. We also observed memory-features in MitoSnap^{lo} cells and effector-features in MitoSnap^{hi} cells (first-daughter cells) – these data is shown later in this document in response to points **P6** and **P18** (Figures R3 and R7 in this document; new Figure 4A-D).

The aim of the current study was to identify mother cell cargoes that play a role as fate determinants (pre-mitotic older mitochondria) and addressing whether the rise of daughter cells exhibiting unequal inheritance relies on autophagy, which we could determine by evaluating the impact of autophagy-loss in early-fate divergence/ACD. We did not perform scRNAseq of autophagy-deficient cells because the role of autophagy in T cell differentiation has been reported previously by our lab and others (Sinclair et al., 2025, Nature Immunology; Franco et al., 2023, Science Immunology; Puleston et al., 2014, eLife; Xu et al.; 2014, Nature Immunology).

(P3) 4. *It is not entirely clear whether autophagy KO cells lose MitoSnap low feature after in vivo challenge which is the critical experiment to demonstrate the mechanistic connection to memory formation.*

The reviewer raises a relevant point, as our analyses were limited to assessing the phenotype and function of MitoSnap^{hi} and MitoSnap^{lo} autophagy-deficient progenies *in vitro*, when we rarely observed significant differences between these two populations – opposite to what we observed in their wild type counterparts. We indeed attempted transferring autophagy KO CD8⁺ T cells (either prior to mitosis or MitoSnap^{hi} vs MitoSnap^{lo} progenies) before the *in vivo* challenge several times over the last few years. We agree that this would be an interesting readout to address whether progeny diversity (effector and memory) relies on autophagy. However, we could never find these cells following adoptive transfer in enough numbers that would allow assessment of their phenotype and function. We link this to the fact that autophagy-deficient cells exhibit poor survival even in nutrient-rich conditions and in presence of cytokines that can support cell survival. We show this in new Fig. 5H. Furthermore, as mentioned above, the supportive role of autophagy for memory CD8⁺ T cells has already been described previously. Indeed, the inability to find autophagy-KO MitoSnap^{hi} vs MitoSnap^{lo} cells after adoptive transfers corroborates the idea that these cells do not live-long term and do not exhibit divergent fates, which happens independently of early mitochondrial inheritance patterns. Given this biological limitation, we are not able to directly address the inheritance pattern of mitochondria in autophagy-deficient cells *in vivo* and link this to their fate trajectory. We believe the reviewer will appreciate the challenges that make this experiment unfeasible.

(P4) 5. *With respect to the scRNA-seq and proteomic dataset, the experiments to demonstrate the importance of cluster 2 on MitoSnap labelling is strong. However, additional metabolite assessments need to be conducted in autophagy defective vs WT cells (mitoSnap high versus low), to formally show that 1C metabolites are altered and there is indeed is a shift to glycolysis and 1C metabolism away from OXPHOS. Ideally, the authors should consider a 13C-labelling experiment. In other words, does the mitoSnap labelling segregate with the metabolic features proposed by the authors?*

We fully agree with the reviewer and we have now made considerable efforts to answer this question appropriately. The core metabolomics facilities to which we have access were not able to perform this experiment in a reasonable timeframe. Thus, we have outsourced this experiment, which also allowed

us to have access to a very experienced and reputable facility (VIB, Leuven, Belgium). Nevertheless, MitoSnap^{lo} CD8⁺ T cells are not abundant following first mitosis (Fig. 2D). Together with the need to expand our breeding to obtain enough MitoSnap mice with the right genotype, this resulted in a significant time delay. We performed experiments following 24 h cell incubation with labelled glucose or serine to evaluate whether MitoSnap^{hi} cells would indeed exhibit more active 1C metabolism when compared to MitoSnap^{lo} cells. We could confirm the metabolic reliance of MitoSnap^{hi} cells on this metabolic pathway: WT MitoSnap^{hi} cells have a significant higher abundance and/or fraction contribution of ¹³C labelled isotopes in metabolites that are essential to 1C metabolism, such as serine and glycine. For these amino acids, autophagy loss frequently resulted in a phenotype similar to the one found in WT MitoSnap^{hi} cells. These results are now part of manuscript (Fig. 8, Fig. S9, S10), but we also add them here to facilitate the reviewer's assessment (Figure R2).

Figure R2: Serine and glycine abundance and ¹³C incorporation (metabolic flux) are influenced by mitochondrial inheritance and autophagy status. (A) CTV labelled naïve MitoSnap CD8⁺ T cells (WT-*Atg16l1*^{fl/+} *Omp25*^{fl/+} *Ert2*^{Cre} or KO-*Atg16l1*^{fl/fl} *Omp25*^{fl/+} *Ert2*^{Cre}) were activated on anti-CD3, anti-CD28 and Fc-ICAM-1 coated plates for 40 h. Cells were cultured in T cell medium containing 500 nM Z-4-Hydroxytamoxifen (4OHT). 16 h post-activation, cells were harvested and labelled with Snap-Cell 647-SiR (preM/old mitochondria) and cultured for further 24 h in presence of ¹³C Glucose (2g/L) or ¹³C Serine (30mg/L). Mito^{lo} and Mito^{hi} cells were sorted and had their metabolome evaluated. (B) C1 metabolism summary. Labelled glucose or serine was used to evaluate metabolic flux. (C) Pie Charts were generated using Travis Pies v1.31. Diameter represents metabolite abundance and red fractions represent the contribution of labelled carbon units originating from ¹³C-Glucose (left-half) or ¹³C-Serine (right-half). WT: 5 biological replicates; KO: 3 biological replicates.

Reviewer #2:

Remarks to the Author:

Borsa et al describe a comprehensive and interesting series of experiments that shed new light on how mitochondria are regulated by asymmetric cell division in CD8+ T cells, and how this influences the memory versus effector fate decisions. The work is carefully performed, novel and compelling; and appropriated for Nature Cell Biology. However, I have concerns regarding interpretation of the young versus new mitochondria, as detailed below.

We would like to thank the reviewer for the extensive, insightful and constructive feedback. As mentioned above, aiming to facilitate your assessment and link between comments and changes in the manuscript file, we have numbered them.

(P5). The authors first compared high and low CD8 daughter cells for differences in proteome between effector-like and memory-like precursors after the first division. This approach complements previous transcriptomics efforts, but is not substantially explored or discussed (eg how do the proteomic data compare to similar transcriptomic efforts?).

The reviewer is correct. The first author of this manuscript has previously performed transcriptomics analysis of CD8^{hi} and CD8^{lo} first-daughter-cells using identical sorting strategies and naïve CD8⁺ T cell activation protocols (Borsa et al., Science Immunology, 2019). We decided to evaluate the proteome of the same populations to identify new asymmetrically inherited cells cargoes in an unbiased manner, which would also allow us to determine if any of them could be autophagy targets. Amongst the unequally inherited proteins, we did not observe those encoded by genes amongst those exhibiting the highest differential expression in our previous work. We can't exclude that the lower depth and higher output variability in our proteomics dataset in comparison to the previous transcriptomics dataset might be at the root of this observation, but it suggests that either i) the inheritance of mRNA preceded any detectable differences in protein expression or ii) transcription events are not always followed by translation. Nevertheless, this step of the work was exploratory, and initiated our interest in studying mitochondrial inheritance due to differential profile of mitochondria-related proteins when CD8^{hi} and CD8^{lo} cells were compared.

(P6). The experiments are only performed using anti-CD3, anti-CD28 and Fc-ICAM-1 coated plates, and ideally would be assessed on MHC-peptide presentation (perhaps using the OT1 mice to tie in with the *in vivo* experiments) to confirm physiological relevance. At a minimum the artificial nature of this activation should be discussed. This notwithstanding, the proteomics approach and comparison with different experimental conditions throughout the manuscript provides a nice complement to the single cell studies.

This is a valid point. To evaluate whether the inheritance pattern of mitochondria across cell divisions would be influenced by the nature of TCR-stimulation, we have performed experiments using peptide-loaded dendritic cells (DC1940 cell line as used in Borsa et al., 2019, Science Immunology). Similar to what we observed in progenies emerging from coated-plate stimulation (Fig. 2E), results suggest that the emergence of MitoSnap^{lo} derives from both degradation and segregation events throughout cell divisions (Figure R5). In response to point P18 (1 from Reviewer #3) we have also analysed the inheritance of mitochondria *in vivo* with similar outcomes.

Figure R3: Inheritance of pre/mitotic old mitochondria following stimulation of OT-I CD8⁺ T cells in a co-culture with peptide-loaded dendritic cells (DCs) correlates with an effector-like signature. Left: Isolated OT-I CD8⁺ T cells were co-cultured with SIINFEKL-loaded DC1940 dendritic cells for 40h. **Middle:** Co-culture of OT-I cells with unloaded DCs was used as a control to determine undivided CD8⁺ T cells and thus, 1st division gates. **Right:** MitoSnap^{lo} cells exhibited a stronger memory-like phenotype (CD44⁺ CD62L⁺), while MitoSnap^{hi} cells exhibited a stronger effector-like phenotype (CD44⁺ CD25⁺).

(P7) The finding that MitoSOX (superoxide-producing mitochondria) is asymmetric in the first daughters is solid, novel and interesting, as is the elegant approach to ablating autophagy after TCR activation, which prevents this asymmetry. Together, the experiments in Fig 1 provide compelling evidence for autophagy-dependent asymmetry in mitochondrial function. The use of *Omp25-SnapTag* mice provides an elegant approach to track aging of mitochondria, and compelling data that old mitochondria are asymmetric at division. Fig. 2E clearly shows more old mitochondria in the knockout cells, but I'm not sure I agree with the interpretation of 'only lost the old mitochondria in 3.6% of all dividing cells, as opposed to 23% in WT conditions' It isn't clear which (if any) of the cells are undivided, but all CTV populations seem higher for 'old mito' in the knockout, and there is not an obvious inflection point to motivate binarizing the data. Perhaps presenting the data as histograms binned by CTV halving would be clearer, and a more appropriate conclusion that old mitochondria were increased at each division compared with the WT. And surely the same analysis was performed on young mitochondria? Did they show a similar loss over time, and was this also dependent on autophagy?

The reviewer is correct about the difficulty of binarizing the data and using similar gating strategies to determine MitoSnap^{lo} cells in autophagy-sufficient and -deficient cells, as the latter exhibit brighter labelling due to the higher abundance of MitoSnap⁺ structures. That is also the reason we decided to refer to the emerging T cell progenies as MitoSnap^{hi} and MitoSnap^{lo} instead of MitoSnap⁺ vs. MitoSnap⁻. However, we would like to highlight that the loss of MitoSnap labelling is a consequence of both segregation and degradation events. Thus, a cell that divided several times might become MitoSnap^{lo} even if autophagy-deficient and an undivided cell that is autophagy-sufficient might reach the same arbitrary inflection point without segregation. Beyond determining what is the exact frequency of cells that would end up being cleared of pre-mitotic (preM) old mitochondria, the data represented in previous Fig. 2E aimed to show that these 2 mechanisms might contribute to the resulting volume of older mitochondria kept by a cell. For KO cells it is more difficult to become MitoSnap^{lo}, as they mostly rely on segregation, which to a large extent happens symmetrically in these cells. **By analysing the data per "CTV halving" we would lose the "dilution effect" of cell division events.** As suggested by the reviewer, we have added data showing the inheritance pattern of younger and older (pre-mitotic) mitochondria per cell division throughout time in the manuscript (new Fig. 2F) and this new data is also represented below (Figure R6). Importantly, at the timepoint when cells were analyzed (several days after activation), what was considered "young" is older than "preM/old mitochondria" at the time point of first cell division, which was used for most functional readouts *in vitro* and *in vivo*. Threshold bars were set based on unlabelled cells, and similar to what has been shown in original Fig 2H (new Fig. 3B), wild type cells clear up both preM/old and young MitoSnap labelling, while autophagy-deficient cells retain them.

Figure R4: Autophagy-deficient results in maintenance of both pre-mitotic old(er) and post-mitotic young(er) MitoSnap labelled organelles. (A) Experimental layout: CTV labelled naïve MitoSnap CD8⁺ T cells (WT-*Atg16l1*^{fl/fl}-*Omp25*^{fl/-}-*Ert2*^{Cre} or KO-*Atg16l1*^{fl/fl}-*Omp25*^{fl/-}-*Ert2*^{Cre}) were activated on anti-CD3, anti-CD28 and Fc-ICAM-1 coated plates for 36-40 h. Cells were cultured in T cell medium containing 500 nM Z-4-Hydroxytamoxifen (4OHT). 16 h post-activation, cells were harvested and labelled with Snap-Cell TMR-Star to tag preM/old mitochondria and cultured for a further 24 h, when Snap-Cell Block and Snap-Cell Oregon Green incubations allowed young organelle labelling. Downstream analysis was done 3 days later. **(B)** PreM/old (left) and young (right) mitochondria inheritance pattern in autophagy-sufficient (WT) and autophagy-deficient (KO) cells.

(P8). *2F is the more compelling figure demonstrating that autophagy is required for asymmetry of old mitochondria at division. I agree that Fig 2H shows autophagy is required for the loss of old mitochondria, but doesn't this also show autophagy is required for the loss of new mitochondria? Flow cytometric characterisation of the mitochondria themselves adds interesting information, but it should be pointed out that the effect in Figure S2C is subtle, and that qualitative difference (fewer smaller mitochondria) would not be sufficient to see the difference in total mitochondrial content, leaving open the possibility that all mitochondrial sizes are reduced more in wt).*

This a relevant point, partially addressed in **P7** (young mitochondria). We corrected the text so that it more accurately describes our results, clarifying that the MitoFlow experiment addresses aspects of mitochondrial architecture and not total mitochondrial content.

(P9). *Figure 2K is difficult to interpret – few (and similar proportions in the two conditions) show only old mitochondria, similar proportions show both old and new, and the major difference highlighted in the manuscript is the 16.4 % of very bright for old mitochondria (in the KO, 1.71 in WT). This discrete population was not evident in Fig 2I – what is the difference here?*

The characterization of mitochondria by flow cytometry does not allow us to directly link the data from preM/old Fig. 2K (new Fig. 3H) to data where MitoSnap fluorescence is measured at the cellular level. Concerning the higher abundance of very bright mitochondrial structures that are positive for both old and young OMP-25, but indeed very bright just for old mitochondria, our interpretation is as following:

1. Upon autophagy-deletion, a larger proportion of mitochondria across different cells retained high/bright OMP-25 labelling.
2. The absence of young-only mitochondrial entities corroborates our microscopy data showing that younger mitochondria structures are generated on the existing mitochondrial network (Fig. 2B). Importantly, the absence of cell populations with the same feature (young-only), could be the result of cells containing a mix of old-only, young-only and mixed mitochondrial structures, and the analysis of MitoSnap labelling at the organelle level allows us to suggest that this is likely a reflection of how mitochondria biogenesis occurs in T cells, where young-only structures are absent or at least very rare.
3. We are not able to see a population of cells that is very bright for old mitochondria because at the cellular level these structures are just a fraction of the whole mitochondrial content, whereas by flow cytometry the mitochondria were fractionated, and analyzed as individual entities.
4. The absence of mitochondria that are “very-bright” for younger structures suggests that their biogenesis occurs at similar levels in autophagy-sufficient and deficient-cells post activation.

We incorporated this interpretation in the text to make our results clearer to any reader.

(P10). *And how do the authors explain that the bulk of the old mitochondria seems untouched by the KO vs WT comparison. I suspect I am missing something here, a schematic describing each experiment would help.*

Indeed, during the first mitosis, even under autophagy-competent conditions, most cells retain older mitochondrial structures, with MitoSnap^{lo} cells being less frequent than MitoSnap^{hi} cells (Fig. 2D). The substantial population of mitochondria labelled with both young and preM/old OMP-25 reflects this retention. As a result, at the time point of first mitosis (40h post-activation), when mitochondrial fractions were purified, differences between WT and KO cells are not pronounced. We have now clarified this point in the text.

(P11). *Overall, I find the asymmetry of mitochondria and the requirement for autophagy compelling, and the discrimination between old and young mitochondria less so. This becomes an issue in later parts of the manuscript, where claims such as ‘MitoSnap^{hi} CD8+ T cells inheriting old mitochondria exhibited higher global translation rates than MitoSnap^{lo} cells’ (line 353) – would be more accurate if ‘inheriting old mitochondria’ were deleted. (similar concerns throughout the manuscript eg line 369).*

The reviewer raises a valid point. Overall, we observed that cells retaining mitochondrial structures for a longer period tend to adopt an effector or short-lived fate. However, we would like to clarify that what we refer to as “old” mitochondria at the first division were generated in the mother cell approximately one day prior to the inheritance readouts. Therefore, “young” mitochondria can effectively become “old” when preserved over a similar timeframe. This is particularly relevant because OMP-25 biogenesis in T cells appears to occur within pre-existing mitochondrial structures, resulting in co-inheritance. For this reason, we did not perform “young” staining in some of the experiments.

Moreover, in some of our findings, such as when measuring ATP production (new Fig. 5F) and when evaluating the metabolomic profiling of MitoSnap^{hi} and MitoSnap^{lo} cells (new Fig. 8, S9, S10), we observe that even under autophagy-deficient conditions, where mitochondrial degradation is impaired, inheritance of pre-mitotic mitochondria can still result in phenotypic differences between MitoSnap^{hi} and MitoSnap^{lo} KO cells, which can influence future T cell fate trajectories. We have nevertheless deleted claims linking inheritance of preM/old mitochondria to any functional outcomes when this would lead to ambiguous or misleading interpretations/conclusions.

(P12). The in vitro experiments in Fig 3 are a nice demonstration of functional differences between MitoSnap^{hi} and lo cells, with MitoSnap^{Lo} cells having better memory potential. A schematic of the timing (eg Fig 2A) and data showing the initial flow cytometry sort would be helpful here to give a feel for the spread in mitochondrial content and the changes with division. And it is not stated, but were young mitochondria also labelled?

We added new information in the figure as suggested by the reviewer. We also added our gating strategy to sort MitoSnap^{hi} and MitoSnap^{lo} cells (which was consistent across experiments) as a figure in the supplementary material (new Fig. S3A). The reason for not using young labelling in all experiments (co-inheritance) has been explained above.

(P13). Morphological characterisation of the mitochondria in Fig 4 is interesting, carefully performed, and compatible with mitochondrial fusion. If I understand correctly, the initial sort for hi vs lo was made somewhat redundant by the post sort binning of images into hi, medium and low. Is there any difference in intensity between the low sort and the low intensity cells in the high sort? Perhaps to avoid confusion, the difference between these two could be in supplemental data, leaving the more interesting comparison of the three bins from the high sort in the main figure. It was not clear to me how long after the sort the cells were imaged. Importantly, this characterisation was performed on aged mitochondria, but presumably the same image analysis could be performed on young (green) mitochondria). This would be interesting to show. The functional characterisation of cells with high and low mitochondria in the remaining panels of Fig 4 are nicely performed, and appropriately interpreted.

We added data concerning MitoSnap intensity in low, mid and high mitochondrial populations found in MitoSnap^{hi} and MitoSnap^{lo} cells both in the manuscript (new Fig. 5B) and here (Figure R5A). For the experiments represented in Figure 5 in the manuscript, we analysed the expression of 1. A membrane or nuclear marker to identify cells, 2. Total mitochondria for identification of individual organelle structures (3D rendering), 3. preM/old mitochondria and 4. 1C metabolism enzymes (Fig. R5B, Fig. S8), which brought us to the limit of colours that can be simultaneously analyzed, reason why we haven't acquired data on young mitochondria. As observed in Fig. 2B and discussed in response to previous points raised by the reviewer, that old and young mitochondria are co-inherited and the expression of young structures is not different between WT and KO cells, we prioritized the characterization of old MitoSnap structures.

Figure R5: Expression of SHMT2 is higher in bright MitoSnap⁺ mitochondria found in MitoSnap^{hi} cells (hi-hi). FACS-purified MitoSnap^{lo} and MitoSnap^{hi} CD8⁺ T cells (based on preM/old mitochondria expression) were imaged by confocal microscopy and had their mitochondrial architecture analysed. This was accompanied by quantification of MitoSnap labelling (left) and SHMT2 expression (right) per individual mitochondria (n=1961). Data are represented as mean ± SEM. Statistical analysis was performed using a non-parametric Dunn test with Bonferroni correction. Exact P values are depicted in the figure. Pooled data from 2 independent experiments.

To directly address the reviewer's point concerning young mitochondria, we nevertheless did one experiment when MitoSnap^{hi} cells were sorted and imaged to determine whether young⁺ old⁺ (double-positive) mitochondrial structures would differ from those young⁻ old⁺ (old only). Our aim was to identify whether sites of young-OMP-25 maintenance would have a preferential location in the mitochondrial network. The results, which can be found below (Figure R6; new Figure S5), indicate that i) brighter old MitoSnap⁺ mitochondria are the ones exhibiting higher young MitoSnap staining (co-inheritance at organelle level as previously observed by MitoFlow and ii) that mitochondrial structures retaining young OMP-25 are found amongst more complex and less compact (fused/tubular) organelles, corroborating the idea that fused mitochondrial networks prevent mitophagy (Rambold et al., PNAS, 2011). We added this data as new Fig. S5 in the manuscript.

Figure R6: Maintenance of young OMP-25 preferentially occurs in less compact mitochondria structures. FACS-purified MitoSnap^{hi} CD8⁺ T cells (based on preM/old and young MitoSnap expression) were imaged by confocal microscopy and had the mitochondrial architecture of young⁺ and young⁻ networks compared (n=1262). This was accompanied by quantification of MitoSnap labelling (young and old OMP-25-SnapTag) intensities in each individual mitochondrion. Statistical analysis was performed using a non-parametric Dunn test with Bonferroni correction. Exact P values are depicted in the figure.

(P14). Fig 5 is a nice analysis of proteomic and transcriptomic differences between cells with high and low mitochondria. Again, the distribution of CTV and Snap tag at the time of sorting should be shown, and the cutoffs for the sort. The implication as worded is that this cutoff discriminates one daughter from the other, but if the hi population is primarily a small distinct very bright population as per Fig 2K, it is not clear how this relates to the more subtle asymmetry likely to reflect ACD.

We added the sorting gating strategy used for MitoSnap^{hi} and MitoSnap^{lo} cell sorting in Fig. S3A.

(P15). Fig 6 uses labelling of both old and new mitochondria, but as far as I can tell, does not then use

the two colours to discriminate between old and new, which leaves me to assume that they were co-regulated? Is the cutoff for hi and low discrimination based on green, red or both? As written the inferences about old and new mitochondria in this section do not seem to be based on the labelling, and it is unclear what the labelling of young mitochondria achieved here.

As explained in response to previous points, indeed preM/old and young mitochondria are co-inherited/regulated, thus MitoSnap^{lo} cells are those that can clear mitochondria in higher rates. For old mitochondria, this results from degradation (autophagy) and/or segregation (ACD), while for young mitochondria, this results only from degradation at the timepoint of sorting, as the timeframe between labelling and any analysis would only allow neglectable frequencies of cell division. We now make clear in the text that, based on our data, first-daughter-cells inheriting pre-mitotic old mitochondria or not clearing young mitochondria in this short timeframe are the same population and being used interchangeably.

(P16). *The discussion begins with an appropriate description of the lack of a direct link between molecular asymmetry at division and functional differences for T cells. However, I don't agree that this paper shows 'asymmetric inheritance of an unequivocal pre-mitotic cell cargo causes divergent T cell fate commitment' (line 561). This would only be the case if the mother mitochondria were inherited by one daughter but not the other, but here the interpretation is confounded by both the small differences in mitochondrial inheritance, and the amplification of these differences by differential autophagy. Unfortunately, this study does not provide that sought-after link. However, this does not detract from the value of this study, just does not merit such a claim.*

We agree with the reviewer that autophagy seems to be important to amplify fate divergence when daughter cells inheriting or not preM/old mitochondria are compared. However, we would like to emphasize that both MitoSnap^{hi} and MitoSnap^{lo} cells are autophagy-competent. This is clearly shown throughout new Fig. 3. Thus, the maintenance of SnapTag-labelled mitochondria (*higher in MitoSnap^{hi} cells*) for a short time frame is enough to impact on cell fate decisions: an early event of inheritance that has long-term consequences, **which prevail over the degradation of these organelles by autophagy in a later timepoint** by MitoSnap^{hi} first-daughter-cells. We sincerely appreciate the reviewer's insightful and constructive comments, as well as their interpretation of our data, which have significantly contributed to clarifying and improving our manuscript. We have highlighted in the new manuscript file the sentences modified based on the reviewer's suggestion.

(P17). *Overall, this is a very nice study, carefully performed and with several novel and important findings. My major concern, raised above but exemplified in line 381 'even in MitoSnap^{hi} cells that inherited their mitochondria from the mother. This implies an all-or-nothing aspect of the mitochondria asymmetry, which is not compatible with the mean value <0.3 in Fig 1E – slightly more mitochondria inherited by one daughter than the other but certainly leaving plenty in the other daughter. This subtle difference, combined with the observation that both red and green mitochondria were lost over time (Fig 2I), and that new mitochondria were not explored in parallel in many of the assays, is not compatible with the framing that old mitochondria behave substantially different to new mitochondria in their regulation and impact. The manuscript would lose very little by reframing away from the old vs new phenomena to a refocus on the mitochondria as a nexus between asymmetric autophagy and subsequent metabolism and function in the daughter cells.*

Sarah Russell

We would like to thank the reviewer for the attention to detail and thorough interpretation of our results, which helped us to strengthen the manuscript's conclusions and address unclear points – a perfect example of how peer review can enhance scientific discoveries. It is now obvious to us that the co-inheritance of pre-mitotic old and young OMP-25⁺ structures was not clearly addressed, but it was the reason not to perform young and old staining MitoSnap labelling in all experiments prior MitoSnap^{hi} and MitoSnap^{lo} characterization. Importantly, the reviewer highlights something we haven't previously done:

autophagy plays a central role in the emergence of cells inheriting mitochondrial content and committing to divergent fates. We have enforced this throughout the text now (highlights referring to P15-17).

Reviewer #3:

Remarks to the Author:

During initial T cell activation, asymmetric cell division has been linked to daughter cell fate, specifically as to whether a daughter cell adopts a short-lived effector cell phenotype or ultimately enters the memory cell pool. Previous studies have linked asymmetric division of metabolic signalling proteins/pathways (eg mTOR) and key transcriptional factors (eg Myc) as being key to this process. In the current study, Borsa and colleagues provide evidence that asymmetric inheritance of "old" mitochondria biases towards short-lived effector cell fate. This is linked to and dependent upon autophagy/mitophagy and is associated with enhancement of one-carbon metabolism and glycolysis. To my knowledge, these findings are very novel whilst the data is convincing. The experimental approaches are cutting edge, including novel mouse models (Mitosnap mice) whilst the manuscript is clearly written and data presented well.

Specific comments:

(P18) *1. The adoptive cell transfer experiments (Fig 3) show that Mitosnap low T cells have increased persistence after Listeria clearance and better recall responses, consistent with the view that inheritance of "old" mitochondria biases against memory formation. It would be informative to compare the responses of Mitosnap low and high cells during the initial phase of activation in vivo - presumably the hypothesis is that the Mitosnap high cells are initially more-effector like but do not persist?*

Following the reviewer's suggestion, we analyzed the phenotype of *in vivo* generated MitoSnap^{hi} vs MitoSnap^{lo} cells. In short, we induced the expression of SnapTag *in vitro* in naïve OT-I cells prior to adoptive transfer. The following day, recipient mice were challenged with LM-OVA. We analysed the phenotype of first-daughter-cells 2 days post-infection. We could observe that very early on, and corroborating our transcriptomics data, MitoSnap^{lo} cells exhibit a stronger memory-precursor phenotype and MitoSnap^{hi} cells show features of cytotoxic CD8⁺ T cells (new Fig. 4B-D; Figure R7).

Figure R7: Mitochondrial inheritance promotes fate divergence *in vivo*. (A) FACS-purified naïve MitoSnap⁺ OT-I CD8⁺ T cells (5×10^5) were transferred to new hosts, followed by infection with *Listeria monocytogenes* expressing OVA. (B) First-daughter-cells were identified based on CTV expression. CD127⁺ populations were quantified in MitoSnap^{hi} and MitoSnap^{lo} progenies (based on pre-mitotic old mitochondria expression). Statistical analysis was performed using unpaired two-tailed Student's *t* test. (C) Frequencies of cells expressing Granzyme B (ex vivo) in MitoSnap^{hi} and MitoSnap^{lo} progenies. Statistical analysis was performed using unpaired two-tailed Student's *t* test.

[P19] - minor comment. Fig 3 legend states that 5×10^3 cells were transferred whilst methods say $5 - 50 \times 10^3$ were transferred. Was the same result achieved when the higher cell number was used? - this is not reported.

We would like to thank the reviewer for the thorough evaluation of our manuscript. We indeed used different cells numbers in different adoptive transfer experiments as the number of MitoSnap^{lo} cells was a limitation and varied across experiments. We added a representative experiment in Figure 3 exactly because of that. The higher memory potential of MitoSnap^{lo} cells was preserved throughout experiments, as represented by numbers and frequencies of transferred OT-I cells in the spleens of host mice when all experiments are pooled together (Fig. R8; new Fig. S3B in the manuscript).

Figure R8: Progenies of MitoSnap^{hi} and MitoSnap^{lo} cells from different adoptive transfer experiments exhibit similar fates. CTV labelled naïve OT-I MitoSnap CD8⁺ T cells (*Atg16l1^{fl/+} Omp25^{fl/+} Ert2^{Cre}*) were activated on anti-CD3, anti-CD28 and Fc-ICAM-1 coated plates for 36-40 h. Cells were cultured in T cell medium containing 500 nM Z-4-Hydroxytamoxifen (4OHT). 16 h post-activation, cells were harvested and labelled with Snap-Cell 647-SiR to tag preM/old mitochondria and put back in culture. 24 h later cells were sorted into MitoSnap^{hi} and MitoSnap^{lo} cells. $5 \times 10^{3-4}$ cells were transferred to new hosts (CD45.1 and CD45.2 congenic markers were used to trace transferred cells). >30 days following adoptive cell transfer, host mice were infected with 2000 colony forming units (CFU) of *Listeria monocytogenes* expressing ovalbumin (OVA) (LM-OVA). Frequency within CD8⁺ T cells and numbers of adoptively transferred OT-I cell progenies in the spleens of recipient mice. Statistical analysis was performed using paired two-tailed Student's *t* test.

(P20) 2. As discussed in the manuscript, previous studies linked asymmetric inheritance/expression of *Myc* to T cell fate. Did the authors find evidence for differential *Myc* expression in the initial proteomics experiments (sorted CD8^{hi} vs CD8^{lo}, Fig 1) or proteomic/transcriptomic analysis of MitoSnap high vs low cells (Fig 5)?

We did not find any evidence of asymmetric inheritance of *Myc* when analyzing the proteomics of CD8^{hi} vs. CD8^{lo} cells (different biological replicates from different experiments). Regarding MitoSnap^{hi} and MitoSnap^{lo} daughter cells, neither *Myc* mRNA nor *Myc* protein were identified as unequally inherited by transcriptomics and proteomics, respectively.

(P21) Minor point. Fig 1F microscopy images right panel are mislabelled as "Atg16l1 KO" - should it not be "old"?

We would like to thank the reviewer for the thorough evaluation of our manuscript. This has now been corrected.